# Towards Principled Dataset Distillation: A Spectral Distribution Perspective

## Abstract

Dataset distillation (DD) aims to compress large-scale datasets into compact synthetic counterparts for efficient model training. However, existing DD methods exhibit substantial performance degradation on long-tailed datasets. We identify two fundamental challenges: *heuristic design choices for distribution discrepancy measure* and *uniform treatment of imbalanced classes*. To address these limitations, we propose **C**lass-**A**ware **S**pectral **D**istribution **M**atching (CSDM), which reformulates distribution alignment via the spectrum of a well-behaved kernel function. This technique maps the original samples into frequency space, resulting in the Spectral Distribution Distance (SDD). To mitigate class imbalance, we exploit the unified form of SDD to perform amplitude-phase decomposition, which adaptively prioritizes the realism in tail classes. On CIFAR-10-LT, with 10 images per class, CSDM achieves a 14.0% improvement over state-of-the-art DD methods, with only a 5.7% performance drop when the number of images in tail classes decreases from 500 to 25, demonstrating strong stability on long-tailed data.

(a) Previous metric only aligned the first-order moment (mean value).

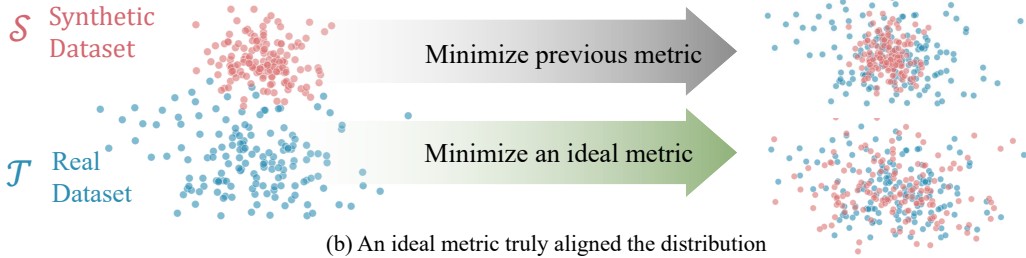

$\mathcal{S}$ Synthetic Dataset

Minimize previous metric

$\mathcal{T}$ Real Dataset

Minimize an ideal metric

(b) An ideal metric truly aligned the distribution

Figure 1: Comparison of different matching methods. (a) Previous works using so-called "MSE" or "linear MMD" metrics essentially align only the first-order moment (mean). However, matching the mean is insufficient to make distributions similar. (b) An ideal metric should align the entire distributions, not just specific moments.

## 1 Introduction

Dataset distillation (DD) synthesizes a small, representative dataset from a large one to accelerate model training in applications like continual learning and federated learning (Wang et al., 2018; Zhao et al., 2020; Zhao & Bilen, 2022). A prominent paradigm, Distribution Matching (DM) (Zhao & Bilen, 2022; Wang et al., 2022), bypasses the bi-level optimization of earlier methods (Zhao et al., 2020; Cazenavette et al., 2022) by directly aligning statistical features of the real and synthetic data.

However, real-world datasets often exhibit a long-tailed distribution (Zhang et al., 2023), posing significant challenges to existing distillation methods, including those based on distribution matching. Notably, when the degree of imbalance is severe, methods designed under the balanced data assumption can even underperform random selection. For example, on CIFAR-10-LT (Zhao et al., 2024) with an imbalance factor of 200, MTT (Cazenavette et al., 2022) achieves only 23.9% accuracy, whereas DM (Zhao & Bilen, 2022) attains 49.9%, merely matching random selection. We attribute the drawbacks of previous DD methods to two aspects:

Figure 2: **Dataset Distillation with Class-Aware Spectral Distribution Matching (CSDM).** (a) *Kernel Embedding.* A feature network extracts features from real and synthetic data. Unlike previous methods using a linear kernel that maps to a suboptimal RKHS, we design a universal kernel that enables accurate measurement of distribution discrepancies. (b) *Class-Aware Spectral Distribution Matching.* We quantify distribution discrepancies using the Spectral Distribution Distance (SDD). By decomposing SDD into amplitude and phase and applying class-specific weights, CSDM dynamically adapts to class imbalance.

**Problem 1: Inadequate Alignment of Distributions.** Early approaches (Zhao & Bilen, 2022; Zhao et al., 2023; Wang et al., 2022; Deng et al., 2024; Sajedi et al., 2023) adopted Mean Squared Error (MSE) or Maximum Mean Discrepancy (MMD) losses to align distributions. However, as shown in Figure 1, they essentially align only the first moment, losing discriminative power when distributions share the same mean—a problem exacerbated on long-tailed datasets. From the kernel perspective (Figure 6), such methods correspond to MMD with a linear kernel, whose mean embeddings cannot guarantee distinctness across distributions. Recent work like NCFM (Wang et al., 2025) leverages characteristic functions but still lacks theoretical grounding. These issues call for a more principled metric that can faithfully measure distribution discrepancies.

**Problem 2: Uniform Treatment of Head and Tail Classes.** Long-tailed datasets exhibit severe imbalance: e.g., CIFAR-10-LT with imbalance factor 200 has head classes with 5000 images but tail classes with only 25. Distilled data must therefore convey richer information for tail classes to offset data scarcity. Yet existing distillation strategies treat all classes uniformly, leading to unrealistic synthesis for underrepresented categories.

To address these problems, we propose **C**lass-**A**ware **S**pectral **D**istribution **M**atching (CSDM). To solve **Problem 1**, we design a principled metric. We show that MMD with a universal kernel can match full distributions. By applying Bochner's theorem, we reformulate this in the Fourier domain, yielding the **S**pectral **D**istribution **D**istance (SDD), an efficient and theoretically grounded metric for distribution matching. To solve **Problem 2**, we decompose the SDD into amplitude and phase components, which we link to data diversity and realism, respectively. CSDM applies class-aware weights, prioritizing diversity for data-rich head classes and realism for data-scarce tail classes. This adaptive strategy preserves crucial information in tail categories. Our main contributions are:

- **Theoretical Framework of Principled DM.** We theoretically reveal the inherent limitations of traditional approaches—such as their restriction to low-order alignment and the entanglement of diversity and realism information. This leads to the reformulation of metric design as the task of designing an optimal kernel.

- **Class-Aware Spectral Distribution Matching (CSDM).** We introduce *Spectral Distribution Distance* (SDD), a discriminative metric grounded in Bochner's theorem and kernel methods. Based on SDD, we propose *Class-Aware Spectral Distribution Matching* (CSDM). By decomposing characteristic function differences into amplitude and phase components, and assigning class-specific weights, CSDM dynamically adapts to class imbalance.

- **Experimental Results.** Our proposed CSDM significantly outperforms all methods based on trivial metrics on both CIFAR-10/100-LT and ImageNet-subset-LT. Notably, CSDM surpasses the *State-of-the-art* method LAD (Zhao et al., 2024), achieving improvements of 12.9% and 14.3% on CIFAR-10-LT and CIFAR-100-LT with 10 images per class (IPC).

## 2 RELATED WORKS

### 2.1 DATASET DISTILLATION WITH DISTRIBUTION MATCHING

Early dataset distillation methods, such as trajectory matching (Cazenavette et al., 2022; Cui et al., 2023; Du et al., 2023; Guo et al., 2023) and gradient matching (Zhao et al., 2020; Lee et al., 2022; Zhao & Bilen, 2021), rely on computationally expensive bi-level optimization to align training dynamics between real and synthetic data. To address this, a more efficient paradigm—distribution matching (DM)—was proposed by (Zhao & Bilen, 2022), which bypasses bilevel optimization by directly aligning real and synthetic data in feature space. However, most of the existing methods pay little attention to the design of the matching metric. For instance, (Zhao & Bilen, 2022; Zhao et al., 2023; Wang et al., 2022; Deng et al., 2024; Sajedi et al., 2023) adopt linear-kernel MMD, which directly compares the means of real and synthetic data, capturing only first-order moment discrepancies. M3D (Zhang et al., 2024) was the first to propose a non-trivial matching metric, but due to a lack of deeper analysis, its empirical performance is nearly indistinguishable from linear-kernel metrics. NCFM (Wang et al., 2025) is the first work in dataset distillation to design the matching metric from the perspective of characteristic functions; however, its neural sampling network remains heuristic.

### 2.2 DATASET DISTILLATION WITH FREQUENCY

Only a few works investigate dataset distillation from a frequency-domain perspective. FreD (Shin et al., 2023) converts images into the Fourier domain and directly optimizes a sparse set of frequency coefficients selected by an explained-variance-based mask, while NSD (Yang et al., 2024) parameterizes the synthetic set as a high-order tensor and applies neural spectral decomposition to share spectral bases across samples. These methods primarily innovate on the parameterization of synthetic data in the frequency domain, typically reusing existing trajectory-, gradient-, or distribution-matching objectives, and pay little attention to how frequency-domain structure could guide the design of the matching metric itself.

### 2.3 LONG-TAILED DATASET CLASSIFICATION

Long-tailed datasets are characterized by an imbalanced distribution of class frequencies, where a few head classes have abundant data and most tail classes have very few samples. A direct approach to addressing this issue is to transform the data into a balanced distribution, with common techniques including oversampling (Haixiang et al., 2017; Janowczyk & Madabhushi, 2016; Peng et al., 2020), undersampling (Buda et al., 2018; Haixiang et al., 2017), and data augmentation (Chou et al., 2020; Gidaris & Komodakis, 2018). Beyond these, LAD (Zhao et al., 2024) adopts decoupling methods (Kang et al., 2019) to enhance the pre-training process, improving the model's ability to handle long-tailed data by separating the backbone learning from classifier fine-tuning. However, existing work has primarily focused on training strategy design, with no studies exploring the development of specialized matching metrics tailored for imbalanced data.

## 3 PRELIMINARIES: DISTRIBUTION MATCHING

Dataset distillation originally relied on gradient (Zhao et al., 2020; Kim et al., 2022; Lee et al., 2022; Zhao & Bilen, 2021) or trajectory (Du et al., 2023; Guo et al., 2023), which incurs high computational costs due to bilevel optimization. As an alternative paradigm, Distribution Matching (DM) (Zhao & Bilen, 2022) embeds real and synthetic data into a feature space and aligns their distributions, yielding improved efficiency and interpretability.

Formally, let $\mathcal{T}$ denote the real dataset and $\mathcal{S}$ the synthetic set. We consider a differentiable siamese augmentation operator $\mathcal{A}(\cdot, \omega)$ (Zhao & Bilen, 2021) with randomness $\omega$, and a feature extractor $f_\vartheta$ parameterized by network weights $\vartheta \sim P_\vartheta$. DM seeks

$$\mathcal{S}^* = \arg\min_{\mathcal{S}} \mathbb{E}_{\vartheta \sim P_\vartheta} \big[ d\big( f_\vartheta(\mathcal{A}(\mathcal{T}, \omega)), f_\vartheta(\mathcal{A}(\mathcal{S}, \omega)) \big) \big]. \tag{1}$$

where $d(\cdot, \cdot)$ measures distributional discrepancy. For the sake of notational convenience, we define $F_{\mathcal{T}} = f_\vartheta(\mathcal{A}(\mathcal{T}, \omega))$ and $F_{\mathcal{S}} = f_\vartheta(\mathcal{A}(\mathcal{S}, \omega))$ and thereby simplify the problem to

$$\mathcal{S}^* = \arg\min_{\mathcal{S}} \mathbb{E}_{\vartheta \sim P_\vartheta}\big[d(F_{\mathcal{T}}, F_{\mathcal{S}})\big]. \tag{2}$$

**Misnomer of Mean Square Error in DM.** Most prior works (Wang et al., 2022; Zhao & Bilen, 2022; Zhao et al., 2023; Sajedi et al., 2023; Deng et al., 2024) choose $d$ as linear-kernel MMD, which is often inappropriately referred to as "MSE":

$$d_{\text{``MSE''}}(F_{\mathcal{T}}, F_{\mathcal{S}}) = d_{\text{linearMMD}}(F_{\mathcal{T}}, F_{\mathcal{S}}) = \big\|\mathbb{E}_{x \sim F_{\mathcal{T}}}[x] - \mathbb{E}_{y \sim F_{\mathcal{S}}}[y]\big\|^2.$$

Mean Square Error (MSE) is defined for paired observations $\mathcal{Y} = \{y_i\}_{i=1}^n$ and $\hat{\mathcal{Y}} = \{\hat{y}_i\}_{i=1}^n$ as

$$\text{MSE}(\mathcal{Y}, \hat{\mathcal{Y}}) = \frac{1}{n}\sum_{i=1}^n \|y_i - \hat{y}_i\|^2,$$

where $\|\cdot\|$ denotes the Euclidean norm. While widely used in regression (Theobald, 1974; Ren et al., 2022; Thompson, 1990), MSE presumes a one-to-one correspondence between real and predicted values. In distributional contexts, where no such pairing exists, MSE is not designed to capture distributional differences. Clearly, the previously used $d_{\text{linearMMD}}$ is not MSE, since it does not require point-wise matching.

## 4 METHODOLOGY

### 4.1 RETHINKING PREVIOUS METHODS FROM A KERNEL PERSPECTIVE

The core of our work is to design a better metric for matching two distributions, $P$ and $Q$, given empirical samples. This is a classic two-sample testing problem. A powerful non-parametric tool for this task is the Maximum Mean Discrepancy (MMD), which measures the distance between distributions as the norm difference of their mean embeddings in a Reproducing Kernel Hilbert Space (RKHS) $\mathcal{H}$:

$$\text{MMD}^2[P, Q] = \big\|\mu_P - \mu_Q\big\|_{\mathcal{H}}^2, \quad \text{where} \quad \mu_P = \mathbb{E}_{x \sim P}[\phi(x)]. \tag{3}$$

Here, $\phi : \mathcal{X} \to \mathcal{H}$ is the feature map associated with a kernel $k$. The choice of this kernel is paramount as it defines the richness of the function class used to distinguish the distributions.

A critical property for a kernel is to be **universal** (Steinwart, 2001). A universal kernel ensures its corresponding RKHS is rich enough to uniquely identify any distribution. This guarantees that $\text{MMD}[P, Q] = 0$ if and only if $P = Q$, making MMD a valid metric. This property is crucial for robust distribution matching.

However, many prior works implicitly adopt a simple **linear kernel**, $k(x, y) = x^\top y$. This choice, while computationally convenient, is **not universal**. Its application reduces the MMD to merely matching the first-order moments:

$$\text{MMD}^2_{\text{linear}}[P, Q] = \big\|\mathbb{E}_{x \sim P}[x] - \mathbb{E}_{y \sim Q}[y]\big\|^2. \tag{4}$$

This formulation is insensitive to higher-order statistical differences, such as variance and skewness, which are vital for capturing the full complexity of data distributions.

### 4.2 CLASS-AWARE SPECTRAL DISTRIBUTION MATCHING

#### 4.2.1 EFFICIENT SPECTRAL DISTRIBUTION METRIC DESIGN VIA UNIVERSAL RKHS

The preceding analysis reveals that overcoming the limitations of linear-kernel methods requires employing a **universal kernel**. Such kernels, like the Gaussian RBF or Laplace kernel, provide a theoretical guarantee for MMD to serve as a true metric between distributions, as formalized below.

**Theorem 1** (Property of Universal MMD (Gretton et al., 2008))**.** *Let $\mathcal{F}$ be the unit ball in a universal RKHS $\mathcal{H}$ defined on a compact metric space $\mathcal{X}$, with kernel $k(\cdot, \cdot)$. Then:*

$$\text{MMD}[\mathcal{F}, P, Q] = 0 \quad \text{iff} \quad P = Q.$$

While Theorem 1 establishes MMD with a universal kernel as a principled metric, its standard computation via kernel matrix evaluations faces significant practical hurdles. Specifically, it suffers from a quadratic computational complexity with respect to the sample size, making it inefficient for large datasets. Furthermore, it entangles diversity and realism into a single scalar value, offering limited interpretability and control. These challenges motivate seeking an alternative representation of the universal MMD that is both efficient and more expressive.

### 4.2.2 SPECTRAL DISTRIBUTION DISTANCE

To address the aforementioned limitations of universal MMD, we adopt a Fourier-transform perspective and revisit the formulation in the spectral domain.

**Theorem 2** (Bochner's Theorem (Bochner, 1932)). *Every bounded, continuous, positive-definite function is the Fourier transform of a non-negative finite Borel measure. Specifically, for any bounded shift-invariant kernel $k(\mathbf{x}, \mathbf{y}) = k(\mathbf{x} - \mathbf{y})$, there exists a non-negative finite Borel measure $\mu$ such that:*

$$k(\mathbf{x}, \mathbf{y}) = \int_{\mathbb{R}^d} e^{j\,\mathbf{t}^\top(\mathbf{x}-\mathbf{y})}\,\mathrm{d}\mu(\mathbf{t}), \tag{5}$$

*where $\mu(\mathbf{t})$ is the Fourier transform of $k(\mathbf{x} - \mathbf{y})$, i.e., $\mu(\mathbf{t}) = \mathcal{F}\{k\}(\mathbf{t})$, and the normalized version $p(\mathbf{t}) = \mu(\mathbf{t})/\int \mathrm{d}\mu(\mathbf{t})$ defines a probability distribution. Here $j = \sqrt{-1}$ denotes the imaginary unit. The measure $\mu$ is referred to as the* spectral measure *of the kernel.*

Invoking Theorem 2, any translation-invariant kernel $k(\mathbf{x}, \mathbf{y})$ admits a unique spectral decomposition. To proceed, recall that the characteristic function of a distribution $P$ is defined by $\phi_P(\mathbf{t}) = \mathbb{E}_{\mathbf{x}\sim P}\big[e^{j\,\mathbf{t}^\top \mathbf{x}}\big]$. Exploiting the independence of two samples $\mathbf{x}, \mathbf{x}' \sim P$ and the conjugation identity $\phi_P(-\mathbf{t}) = \overline{\phi}_P(\mathbf{t})$, we obtain $\mathbb{E}_{\mathbf{x},\mathbf{x}'\sim P}\big[e^{j\,\mathbf{t}^\top(\mathbf{x}-\mathbf{x}')}\big] = \phi_P(\mathbf{t})\,\phi_P(-\mathbf{t}) = \big|\phi_P(\mathbf{t})\big|^2$. A straightforward rearrangement of terms then leads directly to the spectral domain expression of the squared MMD (Gretton et al., 2008).

**Theorem 3.** *For any shift-invariant kernel $k(\mathbf{x}, \mathbf{y})$ with spectral measure $\mu(\mathbf{t})$, the squared Maximum Mean Discrepancy (MMD) can be expressed in the frequency domain as:*

$$\mathrm{MMD}^2[\mathcal{F}, P, Q] = \int_{\mathbb{R}^d} |\phi_P(\mathbf{t}) - \phi_Q(\mathbf{t})|^2 \, \mathrm{d}\mu(\mathbf{t}),$$

*where $\phi_P(\mathbf{t}) = \mathbb{E}_{\mathbf{x}\sim P}[e^{j\,\mathbf{t}^\top \mathbf{x}}]$ and $\phi_Q(\mathbf{t}) = \mathbb{E}_{\mathbf{y}\sim Q}[e^{j\,\mathbf{t}^\top \mathbf{y}}]$ are the characteristic functions of distributions $P$ and $Q$, respectively.*

Based on Theorem 3, we define Spectral Distribution Distance (SDD) as follows:

$$\mathrm{SDD}(F_\mathcal{T}, F_\mathcal{S}) = \int_{\mathbb{R}^d} |\phi_{F_\mathcal{T}}(\mathbf{t}) - \phi_{F_\mathcal{S}}(\mathbf{t})|^2 \, \mathrm{d}\mu(\mathbf{t}). \tag{6}$$

In practice, we approximate the integral using Monte-Carlo sampling (Hammersley, 2013): $\mathrm{SDD}(F_\mathcal{T}, F_\mathcal{S}) \approx \frac{1}{m}\sum_{i=1}^{L} |\phi_{F_\mathcal{T}}(\mathbf{t}_i) - \phi_{F_\mathcal{S}}(\mathbf{t}_i)|^2$. Here, $\{\mathbf{t}_i\}_{i=1}^{L}$ are frequency points independently and identically sampled from the spectral distribution $p(\mathbf{t})$ of the kernel. Based on the law of large numbers, as $L \to \infty$, the estimated value converges in probability to the true integral value (Hammersley, 2013).

Remarkably, the proposed SDD enjoys a linear computational complexity with respect to the dataset size. To evaluate the empirical characteristic function $\phi(\cdot)$ on a dataset with $N$ samples of dimension $D$, we compute the dot product between each sample and a frequency vector, requiring $O(ND)$ operations. Since the approximation sums over $L$ sampled frequency points, the total time complexity of calculating SDD is $O(LND)$. Compared to traditional kernel-based metrics that typically scale quadratically (i.e., $O(N^2)$), the linear complexity of SDD ensures significantly better scalability for large-scale datasets.

### 4.2.3 DETERMINING SPECTRAL DISTRIBUTION OF SDD

We recall our expectations for the kernel function: to satisfy Theorem 1, the kernel should be *universal*; to satisfy Bochner's Theorem 2, the kernel should be *shift-invariant*. Therefore, we summarize several commonly used kernels along with their corresponding properties in Table 1.

Table 1: Comparison of commonly used kernels.

| Kernel | Universal | Shift-Invariant | $K(\mathbf{x}, \mathbf{y})$ | $p(\mathbf{t})$ |
|--------|-----------|-----------------|------------------------------|------------------|
| RBF | ✓ | ✓ | $\exp(-\boldsymbol{\gamma}\|\mathbf{x} - \mathbf{y}\|^2)$ | $\propto \exp\left(-\|\mathbf{t}\|^2/(4\boldsymbol{\gamma})\right)$ |
| Laplace | ✓ | ✓ | $\exp(-\boldsymbol{\gamma}\|\mathbf{x} - \mathbf{y}\|)$ | $\propto \prod_{i=1}^{d} \dfrac{\boldsymbol{\gamma}}{\gamma^2 + t_i^2}$ |
| Linear | × | × | $\mathbf{x}^\top \mathbf{y}$ | – |
| Polynomial | × | × | $(\mathbf{x}^\top \mathbf{y} + c)^d$ | – |
| Sigmoid | × | × | $\tanh(\alpha\, \mathbf{x}^\top \mathbf{y} + c)$ | – |
| Cosine | × | ✓ | $\cos(\omega\|\mathbf{x} - \mathbf{y}\|)$ | $\propto \delta(\|\mathbf{t}\| - \omega)$ |

As shown in the table, only the RBF and Laplace kernels are both universal and shift-invariant. In particular, the spectral distribution of the RBF kernel is Gaussian, namely

$$p_{\mathrm{RBF}}(\mathbf{t}) \propto \exp\left(-\frac{\|\mathbf{t}\|^2}{4\boldsymbol{\gamma}}\right) \implies \mathbf{t} \sim \mathcal{N}\left(\mathbf{0},\, 2\boldsymbol{\gamma}\,\mathbf{I}_d\right).$$

#### 4.2.4 CLASS-AWARE DECOMPOSITION

To mitigate the lack of realism in tail classes, we follow a widely-used decomposition strategy from signal processing (Oppenheim & Lim, 1981; Mandic & Constantinides, 2009), generative modeling (Li et al., 2020) and domain generalization (Lee et al., 2023), and decompose the discrepancy between characteristic functions into amplitude and phase components:

$$
\begin{aligned}
|\phi_{F_\mathcal{T}}(t) - \phi_{F_\mathcal{S}}(t)|^2 &= |\phi_{F_\mathcal{T}}(t)|^2 + |\phi_{F_\mathcal{S}}(t)|^2 - 2\,|\phi_{F_\mathcal{T}}(t)|\,|\phi_{F_\mathcal{S}}(t)| \cos\left(\theta_{F_\mathcal{T}}(t) - \theta_{F_\mathcal{S}}(t)\right) \\
&= \underbrace{\left||\phi_{F_\mathcal{T}}(t)| - |\phi_{F_\mathcal{S}}(t)|\right|^2}_{\text{amplitude difference}} + 2\,|\phi_{F_\mathcal{T}}(t)|\,|\phi_{F_\mathcal{S}}(t)|\underbrace{\left(1 - \cos\left(\theta_{F_\mathcal{T}}(t) - \theta_{F_\mathcal{S}}(t)\right)\right)}_{\text{phase difference}},
\end{aligned}
$$

The amplitude difference quantifies the discrepancy in the spectral magnitude, which reflects the diversity of the feature distribution. Phase Difference, on the other hand, captures misalignment in the phase component, which is indicative of the realism of the data (Wang et al., 2025).

In long-tailed settings, tail classes demand more realism to prevent mode collapse while head classes benefit from greater diversity. To achieve this asymmetric balance, we propose **Class-Aware Spectral Distribution Matching** (CSDM). Our method introduces a class-specific coefficient $\alpha(c) \in [0, 1]$ to adaptively weight the amplitude and phase terms, defining the class-wise discrepancy $d_c$ as:

$$
\begin{aligned}
d_c(x, y)_{x \sim F_\mathcal{T},\, y \sim F_\mathcal{S}} = \int_{\mathbb{R}^d} &\alpha(c)\left||\phi_{F_{\mathcal{T}_c}}(t)| - |\phi_{F_{\mathcal{S}_c}}(t)|\right|^2 \\
&+ 2(1 - \alpha(c))\left|\phi_{F_{\mathcal{T}_c}}(t)\right|\left|\phi_{F_{\mathcal{S}_c}}(t)\right|\left(1 - \cos\left(\theta_{F_{\mathcal{T}_c}}(t) - \theta_{F_{\mathcal{S}_c}}(t)\right)\right) \mathrm{d}\mu(t),
\end{aligned}
$$

where $F_{\mathcal{T}_c}$ and $F_{\mathcal{S}_c}$ denote the normalized real and synthetic feature distributions for class $c$, respectively. The overall distance aggregates class-wise discrepancies:

$$d(x, y)_{x \sim F_\mathcal{T},\, y \sim F_\mathcal{S}} = \sum_c d_c(x, y)_{x \sim F_{\mathcal{T}_c},\, y \sim F_{\mathcal{S}_c}}.$$

This formulation supports class-dependent prioritization of either realism (via phase) or diversity (via amplitude), and naturally handles the challenges of class imbalance in long-tailed data synthesis.

## 5 EXPERIMENTS

### 5.1 SETUP

**Datasets.** Our evaluations were conducted on widely-used datasets: CIFAR-10-LT, CIFAR-100-LT (Krizhevsky et al., 2009) (32×32), and long-tailed ImageNet subsets (Howard & Gugger, 2020)

Table 2: Results on CIFAR-10-LT and CIFAR-100-LT.

| | CIFAR-10-LT | | | | | | CIFAR-100-LT | | |
|---|---|---|---|---|---|---|---|---|---|
| Imbalance Factor | 10 | | | 50 | | | 10 | | |
| IPC | 10 | 20 | 50 | 10 | 20 | 50 | 10 | 20 | 50 |
| Random | 32.5±2.2 | 39.6±0.9 | 51.9±1.5 | 33.2±0.4 | 42.0±1.3 | 51.6±1.3 | 14.2±0.6 | 21.7±0.6 | 32.1±0.6 |
| K-Center | 21.9±0.8 | 24.2±0.8 | 31.7±0.9 | 17.8±0.2 | 20.8±0.5 | 26.1±0.2 | 10.7±0.9 | 15.9±1.0 | 24.8±0.2 |
| Graph-Cut | 28.7±0.9 | 34.2±1.0 | 40.6±1.0 | 24.2±0.7 | 28.6±0.8 | 33.9±0.4 | 16.9±0.3 | 22.2±0.4 | 29.9±0.4 |
| DC | 37.9±0.9 | 38.5±0.9 | 37.4±1.4 | 37.3±0.9 | 37.3±0.9 | 35.8±1.2 | 24.0±0.3 | 27.4±0.3 | 27.4±0.3 |
| MTT | 58.0±0.8 | 59.5±0.4 | 62.0±0.9 | 45.8±1.4 | 49.9±0.8 | 53.6±0.5 | 14.3±0.1 | 16.7±0.2 | 13.8±0.2 |
| DREAM | 34.6±0.6 | 42.2±1.5 | 50.5±0.7 | 30.8±0.6 | 38.4±0.3 | 45.5±0.9 | 10.1±0.4 | 12.0±1.0 | 13.1±0.4 |
| IDM | 54.8±0.4 | 57.1±0.3 | 60.1±0.3 | 51.9±0.7 | 53.3±0.6 | 56.1±0.4 | – | – | – |
| DATM | 57.2±0.4 | 60.4±0.2 | 66.7±0.6 | 41.6±0.2 | 43.4±0.3 | 50.3±0.2 | 28.2±0.4 | 34.1±0.2 | 31.6±0.1 |
| LAD | 58.1±0.3 | 63.0±1.0 | 70.5±0.4 | 54.2±1.0 | 59.4±0.7 | 65.8±0.2 | 31.5±0.2 | 37.5±0.4 | 40.0±0.1 |
| RDED | 47.2±0.6 | 55.0±1.0 | 63.0±1.2 | 43.5±0.2 | 46.8±0.8 | 52.9±1.2 | 40.0±0.4 | 41.7±0.2 | 43.4±0.1 |
| NCFM | 70.2±0.3 | 71.2±0.4 | 75.6±0.2 | 68.8±0.3 | 71.3±0.4 | 72.2±0.3 | 44.7±0.2 | 46.9±0.3 | 48.0±0.3 |
| **CSDM (Ours)** | **71.0±0.4** | **73.9±0.1** | **76.5±0.2** | **69.3±0.3** | **71.6±0.2** | **73.4±0.4** | **45.8±0.2** | **48.1±0.3** | **50.0±0.3** |
| Full Dataset | 76.2±0.3 | | | 61.2±0.5 | | | 44.9±0.4 | | |

| | CIFAR-10-LT | | | | | | CIFAR-100-LT | | |
|---|---|---|---|---|---|---|---|---|---|
| Imbalance Factor | 100 | | | 200 | | | 20 | | |
| IPC | 10 | 20 | 50 | 10 | 20 | 50 | 10 | 20 | 50 |
| Random | 34.4±2.0 | 41.4±0.7 | 52.6±0.5 | 32.5±0.8 | 42.2±1.1 | 49.9±1.4 | 15.0±0.3 | 21.6±0.5 | 30.5±0.5 |
| K-Center | 16.2±0.5 | 19.0±1.0 | 24.2±1.2 | 16.8±0.3 | 17.5±1.4 | 22.6±1.6 | 10.0±0.5 | 15.1±1.6 | 23.8±0.3 |
| Graph-Cut | 22.9±0.9 | 26.0±0.5 | 33.3±1.0 | 22.3±0.9 | 25.2±0.5 | 29.2±0.4 | 16.0±0.5 | 20.7±0.5 | 28.7±0.3 |
| DC | 36.7±0.8 | 38.1±1.0 | 35.3±1.4 | 35.6±0.8 | 35.7±0.9 | 33.3±1.4 | 23.2±0.3 | 26.2±0.3 | 27.4±0.3 |
| MTT | 37.7±0.6 | 41.6±1.1 | 47.8±1.1 | – | 22.6±1.0 | 23.9±0.8 | 12.6±0.3 | 15.0±0.2 | 10.6±0.5 |
| DREAM | 30.8±1.7 | 34.9±0.8 | 42.2±0.8 | 32.7±1.3 | 32.4±0.3 | 38.9±0.4 | 9.4±0.4 | 10.3±0.6 | 12.3±0.3 |
| IDM | 49.8±0.6 | 50.9±0.5 | 53.1±1.4 | 47.0±0.5 | 48.1±1.5 | 49.9±0.3 | – | – | – |
| DATM | 37.3±0.2 | 38.9±0.1 | 44.3±0.1 | – | 34.8±0.1 | 40.1±0.2 | 25.3±0.3 | 27.2±0.1 | 27.1±0.1 |
| LAD | 53.4±0.1 | 58.2±0.6 | 64.0±0.9 | 52.2±0.6 | 56.6±0.4 | 62.3±0.3 | 31.4±0.5 | 35.1±1.4 | 37.0±0.7 |
| RDED | 39.2±0.7 | 43.7±0.4 | 47.6±0.9 | 36.2±0.7 | 40.5±0.4 | 42.4±1.2 | 36.0±0.3 | 37.0±0.3 | 38.1±0.1 |
| NCFM | 60.4±0.6 | 60.1±0.7 | 59.3±0.9 | 57.7±1.0 | 57.0±1.4 | 56.1±1.3 | 42.8±0.6 | 44.7±0.4 | 45.6±0.3 |
| **CSDM (Ours)** | **67.4±0.3** | **69.4±0.3** | **71.0±0.4** | **65.3±0.3** | **66.1±0.5** | **66.8±0.6** | **45.6±0.3** | **48.1±1.4** | **46.6±0.3** |
| Full Dataset | 54.8±0.4 | | | 48.5±0.7 | | | 39.2±0.5 | | |

(128×128). Following (Zhao et al., 2024), these datasets were created by sampling from balanced sets, where the sample size for class $c$, $|\hat{\mathcal{D}}_c|$, follows the exponential decay $|\hat{\mathcal{D}}_c| = |\mathcal{D}_c|\mu^c$ with $\mu^c = \beta^{-(c/C)}$. Here, $C$ is the total number of classes and $\beta = \mathcal{D}_0/\mathcal{D}_C$ is the imbalance factor (Cui et al., 2019), where a larger $\beta$ indicates greater imbalance.

**Networks.** Following prior studies (Zhao et al., 2024; Guo et al., 2023; Wang et al., 2024), we used a 3-layer ConvNet with instance normalization for CIFAR-10/100-LT and a 5-layer ConvNet for Long-tailed ImageNet subsets. We did not modify the network models, focusing solely on adjusting the metric design during distillation.

**Baseline methods.** We compared CSDM against several existing methods. Baselines include classical coreset selection techniques like Random sampling, K-Center Greedy (Sener & Savarese, 2017) and Graph-Cut (Iyer et al., 2021); gradient-matching methods such as DC (Zhao et al., 2020) and MTT (Cazenavette et al., 2022); distribution-matching methods like IDM (Zhao et al., 2023) and DREAM (Liu et al., 2023b); and recent state-of-the-art methods including DATM (Guo et al., 2023), LAD (Zhao et al., 2024), NCFM (Wang et al., 2025) and RDED (Sun et al., 2024).

## 5.2 MAIN RESULTS

We evaluated CSDM on various benchmark datasets across different images-per-class (IPC) and imbalance factor settings.

**CIFAR-10-LT.** The results for CIFAR-10-LT are presented in Table 2. CSDM outperforms all methods initially designed for balanced datasets, as well as LAD, which was specifically designed for long-tailed scenarios. Notably, our method excels particularly under low IPC and high imbalance factor conditions. For example, with an IPC of 10 and an imbalance factor $\beta = 200$, it shows improvements of 29.7%, 18.3%, and 13.1% over DC, IDM, and LAD, respectively.

Table 3: Results on long-tailed ImageNet Subsets.

| Dataset | ImageMeow | | | | ImageSquawk | | | |
|---|---|---|---|---|---|---|---|---|
| Imbalance Factor | 5 | | 10 | | 5 | | 10 | |
| IPC | 1 | 10 | 1 | 10 | 1 | 10 | 1 | 10 |
| Random | $12.8_{\pm0.6}$ | $14.5_{\pm1.0}$ | $10.2_{\pm0.7}$ | $10.3_{\pm0.6}$ | $13.1_{\pm0.9}$ | $13.2_{\pm0.8}$ | $9.6_{\pm0.6}$ | $11.7_{\pm0.8}$ |
| DM | $9.5_{\pm0.8}$ | $12.5_{\pm1.3}$ | $8.4_{\pm1.1}$ | $12.8_{\pm1.3}$ | $10.0_{\pm2.3}$ | $10.8_{\pm0.8}$ | $8.3_{\pm0.8}$ | $10.8_{\pm0.3}$ |
| MTT | $12.3_{\pm0.9}$ | $13.2_{\pm1.0}$ | $11.1_{\pm1.0}$ | $14.1_{\pm0.5}$ | $11.1_{\pm1.4}$ | $12.5_{\pm0.8}$ | $9.2_{\pm0.7}$ | $12.2_{\pm1.1}$ |
| RDED | $9.9_{\pm0.6}$ | $13.7_{\pm0.7}$ | $9.9_{\pm1.0}$ | $12.7_{\pm0.7}$ | $10.5_{\pm1.1}$ | $11.6_{\pm0.5}$ | $7.8_{\pm0.8}$ | $10.8_{\pm0.4}$ |
| **CSDM (Ours)** | $\mathbf{15.6_{\pm0.8}}$ | $\mathbf{23.2_{\pm0.9}}$ | $\mathbf{12.6_{\pm0.8}}$ | $\mathbf{13.7_{\pm1.0}}$ | $\mathbf{15.6_{\pm1.0}}$ | $\mathbf{18.4_{\pm0.9}}$ | $\mathbf{12.6_{\pm0.6}}$ | $\mathbf{12.7_{\pm0.9}}$ |
| Full Dataset | $26.4_{\pm0.7}$ | | $12.4_{\pm0.8}$ | | $15.9_{\pm0.7}$ | | $8.0_{\pm0.8}$ | |
| Dataset | ImageWoof | | | | ImageNette | | | |
| Imbalance Factor | 5 | | 10 | | 5 | | 10 | |
| IPC | 1 | 10 | 1 | 10 | 1 | 10 | 1 | 10 |
| Random | $11.9_{\pm1.1}$ | $15.7_{\pm0.5}$ | $13.1_{\pm0.8}$ | $14.1_{\pm0.9}$ | $13.5_{\pm0.8}$ | $17.2_{\pm0.5}$ | $15.1_{\pm0.6}$ | $19.7_{\pm0.9}$ |
| DM | $10.0_{\pm1.3}$ | $11.1_{\pm1.0}$ | $8.3_{\pm0.8}$ | $11.1_{\pm0.5}$ | $10.6_{\pm0.5}$ | $16.2_{\pm1.2}$ | $10.2_{\pm1.5}$ | $13.2_{\pm0.9}$ |
| MTT | $9.9_{\pm0.9}$ | $11.2_{\pm0.5}$ | $9.8_{\pm0.6}$ | $10.4_{\pm0.9}$ | $15.4_{\pm1.5}$ | $16.2_{\pm1.4}$ | $11.5_{\pm1.1}$ | $15.2_{\pm1.1}$ |
| RDED | $9.3_{\pm1.5}$ | $11.5_{\pm0.4}$ | $8.8_{\pm0.7}$ | $11.1_{\pm0.6}$ | $11.5_{\pm1.3}$ | $25.2_{\pm1.9}$ | $12.0_{\pm1.5}$ | $14.3_{\pm0.9}$ |
| **CSDM (Ours)** | $\mathbf{14.4_{\pm0.7}}$ | $\mathbf{17.3_{\pm1.0}}$ | $\mathbf{12.3_{\pm0.9}}$ | $\mathbf{14.1_{\pm1.2}}$ | $\mathbf{24.3_{\pm0.7}}$ | $\mathbf{29.5_{\pm0.9}}$ | $\mathbf{19.8_{\pm0.6}}$ | $\mathbf{24.8_{\pm0.8}}$ |
| Full Dataset | $15.8_{\pm0.7}$ | | $12.2_{\pm0.7}$ | | $34.7_{\pm0.9}$ | | $23.3_{\pm0.8}$ | |

**CIFAR-100-LT.** On CIFAR-100-LT, the results in Table 2 clearly show that CSDM consistently outperforms all competing methods across all experimental settings. Notably, even under low IPC conditions, its performance is only minimally affected by mild class imbalance, demonstrating a superior level of robustness over alternative methods that do not explicitly handle such imbalance.

**Long-tailed ImageNet Subsets.** Results on Long-tailed ImageNet subsets are in Table 3. On these high-resolution datasets (ImageNette, ImageWoof, etc.) (Howard & Gugger, 2020), the adverse effects of class imbalance are markedly stronger. Despite this challenge, CSDM demonstrates superior resilience, recording the smallest absolute accuracy degradation in most cases. For example, on ImageNette with $\beta = 5$, CSDM increases DM's accuracy from 16.24% to 29.52% (a $1.8\times$ improvement). Similar trends on ImageWoof and ImageSquawk show a clear margin over RDED and DM. This consistent advantage confirms that explicitly modeling the long-tailed distribution is crucial for robustness in high-resolution distillation.

**Cross-Architecture Generalization.** To evaluate the generalization capability of our approach across network architectures, we conducted experiments where the synthetic datasets were distilled using a 3-layer ConvNet and later evaluated on a range of different models, including AlexNet (Krizhevsky et al., 2009), VGG-11 (Simonyan & Zisserman, 2014), and ResNet-18 (He et al., 2016). This setup allows us to assess the robustness of each method when trans-

Table 4: Cross-architecture evaluation.

| Method | ConvNet | ResNet18 | VGG-11 | AlexNet |
|---|---|---|---|---|
| Random | 52 | 49.2 | 47.8 | 47.2 |
| MTT | 49.1 | 42.9 | 42.3 | 40 |
| DATM | 44.4 | 42.4 | 42.2 | 44.9 |
| LAD | 64.7 | 60 | 55.5 | 53.2 |
| **CSDM** | **70.1** | **65.9** | **69.8** | **66.4** |

ferred to architectures not seen during distillation. As summarized in Table 4, our method consistently delivers superior performance on CIFAR-10 under the challenging setting of 50 images per class and an imbalance factor of $\beta = 100$. These results highlight the strong cross-architecture transferability of our approach, which remains effective even when tested on models that differ significantly from the one used for data synthesis.

## 5.3 ABLATION STUDY

### 5.3.1 EFFECT OF KERNEL FUNCTION AND SCALE PARAMETER

We conduct ablation studies on CIFAR-10-LT to investigate different kernel functions. Unlike the linear kernel, Gaussian and Laplace kernels introduce a scale parameter $\gamma$ that critically controls emphasis on different frequency components. The choice of $\gamma$ presents a trade-off: a large $\gamma$ overemphasizes high-frequency differences, causing overfitting, while a small $\gamma$ filters them out, leading to underfitting. As shown in Figure 3, both extremes degrade performance, aligning with prior work

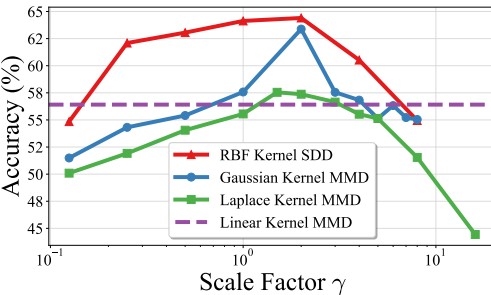

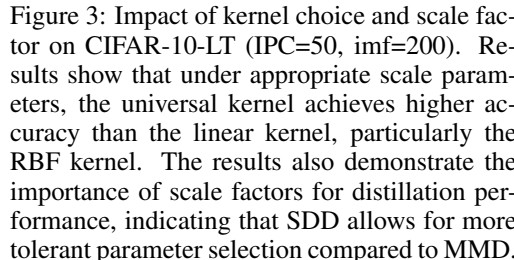

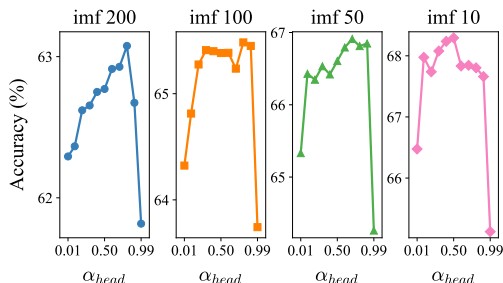

Figure 3: Impact of kernel choice and scale factor on CIFAR-10-LT (IPC=50, imf=200). Results show that under appropriate scale parameters, the universal kernel achieves higher accuracy than the linear kernel, particularly the RBF kernel. The results also demonstrate the importance of scale factors for distillation performance, indicating that SDD allows for more tolerant parameter selection compared to MMD.

Figure 4: Ablation on the class-aware weighting for amplitude and phase on CIFAR-10-LT (IPC=10). We set the amplitude weight for head classes to $\alpha_{head}$ and for tail classes to $1-\alpha_{head}$. The results show that optimal performance is achieved by emphasizing diversity (amplitude) for head classes and realism (phase) for tail classes, with a peak around 0.7–0.9 when the imbalance factor (imf) is large.

on support vector machines that underscores the importance of proper $\gamma$ selection (Hearst et al., 1998; Lu, 2015; Cavoretto et al., 2024). To mitigate this sensitivity and improve generalization, we normalize the feature sets $F_{\mathcal{T}}$ and $F_{\mathcal{S}}$ to a unified scale. This allows a single parameter, $\gamma = 2$, to be applied effectively across all datasets, yielding robustly strong results.

### 5.3.2 IMPACT OF CLASS-AWARE AMPLITUDE-PHASE COEFFICIENT

To evaluate the effect of class-wise balancing between amplitude and phase information, we conduct ablation studies on CIFAR-10-LT by varying the class-specific weighting coefficient $\alpha(c)$, which controls the contribution of amplitude and phase for each class. Prior studies (Mandic & Constantinides, 2009; Oppenheim & Lim, 1981) suggest that amplitude primarily promotes sample diversity, while phase captures structural fidelity. As shown in Figure 4, moderately increasing the amplitude weight for head classes and the phase weight for tail classes improves test accuracy. In contrast, reversing this trend leads to performance degradation—indicating that diversity is more critical for head classes, while realism is more important for tail classes.

## 6 DISCUSSION

Recent dataset distillation (DD) methods increasingly attempt to go beyond first-order alignment. In this section, we situate CSDM within this broader landscape and clarify how our metric-based design relates to three major lines of "higher-order" techniques. Detailed derivations are provided in Appendix B.

**(1) IID/DSDM: Adding handcrafted higher-order terms on top of linear MMD.** IID (Deng et al., 2024) and DSDM (Li et al., 2024) start from the standard DM objective (i.e., linear-kernel MMD) and incorporate additional terms such as covariance matching, class-centralization losses, or prototype regularizers. As shown in Theorem 7 (Appendix), these covariance penalties are mathematically equivalent to matching *only the first two derivatives of the log characteristic function at the origin*—i.e., finite-order Taylor expansion terms. Thus, although effective in practice, they fundamentally remain *finite-order* approximations of distributional structure.

In contrast, CSDM does not append specific moment terms; instead, the use of a universal kernel implies (Appendix Theorem 6) that the resulting SDD objective implicitly matches *all orders of moments*. This replaces ad-hoc higher-order penalties by a principled discrepancy that is distribution-discriminative by construction.

**(2) Universal-kernel MMD: Connection to M3D and the importance of kernel scale.**
M3D (Zhang et al., 2024) is the first DD method to explore nontrivial kernels for MMD. However, its empirical study reports that Gaussian, linear, and polynomial kernels perform similarly, which seems at odds with the theoretical universality of Gaussian kernels. Our analysis (Fig. 3) shows that this phenomenon arises mainly from the choice of the default scale factor $\gamma = 1$: for such a scale, the resulting Gaussian kernel behaves nearly linearly in high dimensions, masking its universal behavior. Once $\gamma$ is properly tuned, the difference between universal and non-universal kernels becomes significant.

**(3) NCFM: Min–max neural weighting.** NCFM (Wang et al., 2025) learns a sampling distribution over frequencies by maximizing a characteristic-function discrepancy (CFD) and then minimizing it with respect to synthetic data. While promising, this min–max formulation introduces two limitations: (i) a learned weighting function does not guarantee that the resulting CFD remains a strict metric; and (ii) maximizing CFD is not directly aligned with downstream classification, especially under long-tailed imbalance, as our experiments (Table 2) show.

Our approach therefore keeps the spectral measure fixed via the kernel's Bochner representation, ensuring metricity and universality, and introduces class-aware amplitude–phase reweighting in a *controlled analytic form*. Proposition 1 (Appendix) proves that any $\alpha(c) \in (0, 1)$ preserves distribution discriminativity, while allowing the optimization to adapt to head–tail asymmetry.

**Summary.** IID/DSDM extend linear MMD with handcrafted low-order statistics; M3D explores nonlinear kernels without fully leveraging universality due to scale choices; and NCFM learns frequency weights without metric guarantees and with suboptimal behavior under imbalance. CSDM differs fundamentally by redesigning the *metric* itself: starting from universal-kernel theory, moving to the spectral domain through Bochner's theorem, and introducing class-aware amplitude–phase weighting while preserving discriminativity. This conceptual foundation makes CSDM complementary to prior regularizers and suggests principled directions for future neural sampling methods.

## 7 CONCLUSION

In this work, we developed a principled metric for long-tailed dataset distillation rooted in kernel theory and proposed Class-Aware Spectral Distribution Matching (CSDM). Through sophisticated kernel design and amplitude-phase decomposition, our method addresses the failure of long-tailed dataset distillation by ensuring a comprehensive alignment of distributions while adaptively balancing the demand for sample diversity in head classes with the need for realism in tail classes. Extensive experiments across diverse datasets demonstrated that our method not only achieves state-of-the-art performance but also exhibits remarkable stability and generalization, showcasing its versatility and effectiveness across datasets with varying types and degrees of class imbalance.

## A  ETHICS STATEMENT

This work is in full compliance with the ICLR Code of Ethics. Our research is purely algorithmic and theoretical, involving no experiments on human subjects or animals. The datasets utilized in this study, including CIFAR-10-LT, CIFAR-100-LT, and various long-tailed ImageNet subsets, were constructed based on standard, publicly-available benchmarks. We have ensured that the use of these original datasets adheres to all licensing and usage terms.

A primary goal of our research is to address the performance degradation on long-tailed datasets, which is a known source of bias in machine learning models. Throughout our research process, we have been diligent in ensuring that our proposed method, Class-Aware Spectral Distribution Matching (CSDM), does not introduce new biases or discriminatory outcomes. The datasets used do not contain personally identifiable information, and our experiments pose no privacy or security risks. We are firmly committed to maintaining the highest standards of transparency and integrity in our research.

## B  REPRODUCIBILITY STATEMENT

To ensure the reproducibility of our findings, we will make our complete source code publicly available in a repository upon publication. The repository will include all necessary scripts to replicate the experiments presented in this paper.

Our paper provides a detailed description of the experimental setup, including the network architectures, specific training procedures, and hyperparameter settings for all compared methods. We have also specified the exact procedure for constructing the long-tailed datasets from their standard counterparts, as referenced in our manuscript, ensuring that identical data splits can be generated. Furthermore, we have offered a thorough explanation of our proposed Class-Aware Spectral Distribution Matching (CSDM) framework and the Spectral Distribution Distance (SDD) metric to facilitate independent implementation by other researchers.

We are confident that these provisions will empower other researchers to verify our results, build upon our work, and further advance the field of dataset distillation.

## C  THE USE OF LARGE LANGUAGE MODELS

In this work, we employed large language models (LLMs) as auxiliary tools to enhance the linguistic quality of the manuscript. Specifically, LLMs were used to refine sentence phrasing and correct grammatical issues, with the goal of improving readability and fluency while strictly preserving the intended technical meaning and content integrity.

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

# A SUPPLEMENTARY THEORETICAL ANALYSIS: BEYOND MOMENT MATCHING

Building on the research trajectory of Distribution Matching in Dataset Distillation, this section aims to systematically clarify several recurrent conceptual confusions and theoretical misinterpretations in the paper. Most Distribution Matching methods adopt the linear kernel MMD introduced by DM (Zhao & Bilen, 2022), which has been erroneously labeled as "MSE" by some researchers. Some studies have recognized that linear kernel MMD essentially only performs first-order moment matching and have begun to explicitly incorporate second- and third-order moment constraints (Zhang et al., 2024; Deng et al., 2024). However, since finite-order moment matching is not equivalent to distribution matching (Sriperumbudur et al., 2010), these attempts have yielded only limited practical improvements. Consequently, a widely accepted view has emerged: "Moment matching / MMD is flawed, as it can only align low-order moments and fails to truly match distributions."

Against this backdrop, many researchers have turned to characteristic function-based methods (CFD; e.g., (Ansari et al., 2020; Li et al., 2020; 2023)) and Wasserstein-type distances (Liu et al., 2023a), leading to the proposal of novel distillation frameworks such as NCFM (Wang et al., 2025).

However, as this section will demonstrate, the issue lies not in the framework of moment matching or MMD per se, but in the fact that most literature has only adopted the linear kernel specialization of MMD. This particular kernel choice offers extremely limited expressive power, which has perpetuated the misconception that "MMD is equivalent to matching mean statistics." In fact, when constructed with universal kernels, MMD not only serves as a strict metric for probability distributions but also naturally translates—via its frequency-domain interpretation—into a spectral integral form of characteristic functions, thereby enabling a complete characterization of the distribution. This point is well-established in the kernel two-sample testing paper (Gretton et al., 2008), yet remains systematically underutilized in the field of Dataset Distillation.

## A.1 MMD AND UNIVERSAL KERNELS: FROM IPM TO DISTRIBUTION MATCHING

(A) Two-Sample Problem and IPM Definition

Let $P$ and $Q$ be two probability distributions defined on the input space $\mathcal{X}$, accessible only through samples:

$$X = \{x_i\}_{i=1}^m \sim P, \quad Y = \{y_j\}_{j=1}^n \sim Q.$$

We aim to determine whether $P$ and $Q$ are identical without making any assumptions about their distributional forms. This is the classical two-sample testing problem [Gretton et al., *A Kernel Method for the Two-Sample Problem*].

A natural approach is to consider a function class $\mathcal{F}$ and define:

$$\mathrm{MMD}[\mathcal{F}, P, Q] = \sup_{f \in \mathcal{F}} \left( \mathbb{E}_{x \sim P}[f(x)] - \mathbb{E}_{y \sim Q}[f(y)] \right),$$

which belongs to the broad category of Integral Probability Metrics (IPM). If the function class $\mathcal{F}$ is sufficiently rich, then:

If $P = Q$, then clearly $\mathbb{E}_P f = \mathbb{E}_Q f$, and thus $MMD = 0$; If $P \neq Q$, we hope there exists some $f \in \mathcal{F}$ such that their expectations differ, leading to $MMD > 0$.

Therefore, the design of $\mathcal{F}$ must simultaneously satisfy:

1. Richness: The function class should be sufficient to distinguish between different distributions;
2. Regularity: The empirical estimate of MMD should have good convergence properties with finite samples.

The most common choice in modern MMD literature is to let $\mathcal{F}$ be the unit ball in a Reproducing Kernel Hilbert Space (RKHS) $\mathcal{H}$, i.e.,

$$\mathcal{F} = \{f \in \mathcal{H} : \|f\|_{\mathcal{H}} \leq 1\}.$$

Below we briefly review the relevant concepts and present key theorems.

**(B) RKHS and Kernel Mean Embedding**

Let $\mathcal{H}$ be a Hilbert space consisting of functions $f : \mathcal{X} \to \mathbb{R}$, with inner product denoted as $\langle \cdot, \cdot \rangle_{\mathcal{H}}$ and norm $\| \cdot \|_{\mathcal{H}}$. If there exists a symmetric positive definite kernel function $k : \mathcal{X} \times \mathcal{X} \to \mathbb{R}$ such that:

1. For any $x \in \mathcal{X}$, $k(\cdot, x) \in \mathcal{H}$; 2. For any $f \in \mathcal{H}, x \in \mathcal{X}$,

$$f(x) = \langle f, k(\cdot, x) \rangle_{\mathcal{H}},$$

then $\mathcal{H}$ is called the Reproducing Kernel Hilbert Space (RKHS) of kernel $k$. Furthermore, treating

$$\phi(x) := k(\cdot, x) \in \mathcal{H}$$

as the feature map, the inner product satisfies

$$\langle \phi(x), \phi(y) \rangle_{\mathcal{H}} = k(x, y),$$

which is the so-called kernel trick: we can perform various computations using only kernel function values without explicitly computing $\phi(x)$.

Within this framework, we can define the kernel mean embedding of distributions [Gretton et al.]:

**Definition 1.** *(Kernel Mean Embedding) Let $X \sim P$ with feature map $\phi : \mathcal{X} \to \mathcal{H}$. If $\mathbb{E}_{x \sim P} \sqrt{k(x,x)} < \infty$, then define*

$$\mu_P := \mathbb{E}_{x \sim P}[\phi(x)] \in \mathcal{H}$$

*as the mean embedding of distribution $P$ in the RKHS.*

This vector $\mu_P$ can be understood as the "representative point of the distribution in the feature space", and MMD is the distance between two mean embeddings $\mu_P, \mu_Q$.

**(C) RKHS Form of MMD and Empirical Estimation**

Let $\mathcal{F}$ be the unit ball of $\mathcal{H}$, i.e., $\|f\|_{\mathcal{H}} \leq 1$. Then we have the following classical result [Gretton et al., *A Kernel Method for the Two-Sample Problem*].

**Theorem 4.** *(RKHS Expression of MMD) Let $\mu_P, \mu_Q$ be the kernel mean embeddings of distributions $P, Q$, respectively. Then*

$$\mathrm{MMD}[\mathcal{F}, P, Q] = \|\mu_P - \mu_Q\|_{\mathcal{H}}.$$

*Proof.* By the reproducing property of the Reproducing Kernel Hilbert Space (RKHS), for any $f \in \mathcal{H}$,

$$\mathbb{E}_{x \sim P}[f(x)] = \mathbb{E}_{x \sim P}\langle f, \phi(x) \rangle_{\mathcal{H}} = \langle f, \mathbb{E}_{x \sim P}[\phi(x)]\rangle_{\mathcal{H}} = \langle f, \mu_P \rangle_{\mathcal{H}}.$$

Similarly,

$$\mathbb{E}_{y \sim Q}[f(y)] = \langle f, \mu_Q \rangle_{\mathcal{H}}.$$

Therefore,

$$\mathbb{E}_{x \sim P}[f(x)] - \mathbb{E}_{y \sim Q}[f(y)] = \langle f, \mu_P - \mu_Q \rangle_{\mathcal{H}}.$$

Under the constraint $\|f\|_{\mathcal{H}} \leq 1$, the maximum of this expression is given by the Cauchy–Schwarz inequality:

$$\sup_{\|f\|_{\mathcal{H}} \leq 1} \langle f, \mu_P - \mu_Q \rangle_{\mathcal{H}} = \|\mu_P - \mu_Q\|_{\mathcal{H}}.$$

This is exactly the definition of $\mathrm{MMD}[\mathcal{F}, P, Q]$, thus proving the theorem. $\square$

**Theorem 5.** *(Kernel Expectation Form of MMD) Let $\mathcal{H}$ be the Reproducing Kernel Hilbert Space (RKHS) corresponding to kernel $k$. Then*

$$\mathrm{MMD}^2[\mathcal{F}, P, Q] = \mathbb{E}_{x,x' \sim P}[k(x, x')] - 2\mathbb{E}_{x \sim P, y \sim Q}[k(x, y)] + \mathbb{E}_{y,y' \sim Q}[k(y, y')].$$

*Proof.* By Theorem 4, we have

$$\mathrm{MMD}^2 = \|\mu_P - \mu_Q\|_{\mathcal{H}}^2 = \langle \mu_P, \mu_P \rangle_{\mathcal{H}} + \langle \mu_Q, \mu_Q \rangle_{\mathcal{H}} - 2\langle \mu_P, \mu_Q \rangle_{\mathcal{H}}.$$

Expanding the first term:

$$\langle \mu_P, \mu_P \rangle_{\mathcal{H}} = \langle \mathbb{E}_{x \sim P}[\phi(x)], \mathbb{E}_{x' \sim P}[\phi(x')]\rangle_{\mathcal{H}} = \mathbb{E}_{x,x' \sim P}[\langle \phi(x), \phi(x') \rangle_{\mathcal{H}}] = \mathbb{E}_{x,x' \sim P}[k(x, x')],$$

Similarly, the second term gives $\mathbb{E}_{y,y' \sim Q}[k(y, y')]$, and the third term:

$$\langle \mu_P, \mu_Q \rangle_{\mathcal{H}} = \mathbb{E}_{x \sim P, y \sim Q}[\langle \phi(x), \phi(y) \rangle_{\mathcal{H}}] = \mathbb{E}_{x \sim P, y \sim Q}[k(x, y)].$$

Substituting these expressions yields the desired result. $\square$

In practical computation, we only have access to finite samples ($\{x_i\} \sim P, \{y_j\} \sim Q$). By replacing the expectations with empirical averages, we obtain either unbiased or biased MMD estimators [Gretton et al.]. This paper adopts the standard biased estimator form in implementation.

(D) Universal Kernels and Distribution Matching

The above derivation shows that the value of MMD is entirely determined by the mean embeddings $\mu_P, \mu_Q$. A key question then arises: Does the mean embedding uniquely characterize the distribution?

If the mapping $P \mapsto \mu_P$ is injective, i.e., $\mu_P = \mu_Q \Rightarrow P = Q$, then the corresponding kernel is called a characteristic kernel [Gretton et al.]. Under certain conditions, universal kernels guarantee this property [Steinwart, *On the Influence of the Kernel on the Consistency of Support Vector Machines*].

We adopt the following classical result:

**Theorem 1.** *Let $\mathcal{X}$ be a compact metric space, and $k$ be a universal kernel defined on $\mathcal{X}$, with its corresponding Reproducing Kernel Hilbert Space (RKHS) denoted as $\mathcal{H}$. Let $\mathcal{F}$ be the unit ball of $\mathcal{H}$. Then for any Borel probability measures $P$ and $Q$, we have*

$$\mathrm{MMD}[\mathcal{F}, P, Q] = 0 \iff P = Q.$$

*Proof.* According to the definition of universal kernels (see Steinwart), the corresponding RKHS $\mathcal{H}$ is dense in $\mathcal{C}(\mathcal{X})$ with respect to the supremum norm. That is, for any $g \in \mathcal{C}(\mathcal{X})$ and any $\varepsilon > 0$, there exists $f \in \mathcal{H}$ such that

$$\sup_{x \in \mathcal{X}} |f(x) - g(x)| < \varepsilon.$$

By Theorem 4, we have

$$\mathrm{MMD}[\mathcal{F}, P, Q] = 0 \iff \|\mu_P - \mu_Q\|_{\mathcal{H}} = 0,$$

which implies $\mu_P = \mu_Q$. According to the definition of mean embeddings, we obtain that for all $f \in \mathcal{H}$,

$$\mathbb{E}_P[f] = \langle f, \mu_P \rangle_{\mathcal{H}} = \langle f, \mu_Q \rangle_{\mathcal{H}} = \mathbb{E}_Q[f].$$

Let $g \in \mathcal{C}(\mathcal{X})$ be arbitrary and take $\varepsilon > 0$. By density, there exists $f \in \mathcal{H}$ satisfying

$$\sup_{x \in \mathcal{X}} \|f(x) - g(x)\| < \varepsilon.$$

Then

$$\|\mathbb{E}_P[g] - \mathbb{E}_Q[g]\| \leq \|\mathbb{E}_P[g - f]\| + \|\mathbb{E}_P[f] - \mathbb{E}_Q[f]\| + \|\mathbb{E}_Q[f - g]\|$$
$$\leq \mathbb{E}_P\|g - f\| + 0 + \mathbb{E}_Q\|f - g\|$$
$$\leq 2\varepsilon,$$

where the second term is zero because $f \in \mathcal{H}$ and for all $f \in \mathcal{H}$ we already have $\mathbb{E}_P[f] = \mathbb{E}_Q[f]$. Letting $\varepsilon \to 0$ yields

$$\mathbb{E}_P[g] = \mathbb{E}_Q[g], \qquad \forall g \in \mathcal{C}(\mathcal{X}).$$

On a compact metric space, a Borel probability measure is uniquely determined by its integrals against continuous functions (this can be derived from the Riesz representation theorem or representation theorems for positive linear functionals in measure theory). Therefore, if for all $g \in \mathcal{C}(\mathcal{X})$ we have

$$\int g \, \mathrm{d}P = \int g \, \mathrm{d}Q,$$

then it must be that $P = Q$.

In summary, $\mathrm{MMD}[\mathcal{F}, P, Q] = 0$ implies $P = Q$; the converse implication $P = Q \Rightarrow \mathrm{MMD} = 0$ follows directly from the definition. This completes the proof. $\square$

Below, we provide an explicit expansion of the Gaussian kernel to intuitively demonstrate how MMD implicitly matches all orders of moments.

## A.2 MOMENT EXPANSION OF GAUSSIAN KERNEL MMD: FROM "FINITE-ORDER" TO "ALL-ORDER"

To more intuitively understand how "universal kernel MMD implicitly includes all orders of moments", we take the most commonly used Gaussian kernel as an example and provide a detailed derivation of the moment expansion. Technically, this expansion can be viewed as an infinite-order generalization of high-order polynomial kernels.

Consider the Gaussian kernel on a $d$-dimensional input space:

$$k(x, y) = \exp\Big( -\frac{\|x - y\|^2}{2\sigma^2} \Big), \quad x, y \in \mathbb{R}^d.$$

We rewrite it as a product of three terms:

$$k(x, y) = \exp\Big( -\frac{\|x\|^2}{2\sigma^2} \Big) \exp\Big( -\frac{\|y\|^2}{2\sigma^2} \Big) \exp\Big( \frac{x^\top y}{\sigma^2} \Big).$$

Applying power series expansion to the third factor gives:

$$\exp\Big( \frac{x^\top y}{\sigma^2} \Big) = \sum_{n=0}^{\infty} \frac{1}{n! \sigma^{2n}} (x^\top y)^n.$$

Further expand $(x^\top y)^n$ in multi-index notation. Let $\alpha = (\alpha_1, \ldots, \alpha_d) \in \mathbb{N}^d$, $|\alpha| = \sum_{i=1}^{d} \alpha_i$, $\alpha! = \prod_{i=1}^{d} \alpha_i!$, $x^\alpha = \prod_{i=1}^{d} x_i^{\alpha_i}$. Then

$$(x^\top y)^n = \sum_{|\alpha|=n} \frac{n!}{\alpha!} x^\alpha y^\alpha.$$

Substituting back into the power series gives:

$$\exp\Big( \frac{x^\top y}{\sigma^2} \Big) = \sum_{n=0}^{\infty} \frac{1}{n! \sigma^{2n}} \sum_{|\alpha|=n} \frac{n!}{\alpha!} x^\alpha y^\alpha$$

$$= \sum_{n=0}^{\infty} \sum_{|\alpha|=n} \frac{1}{\alpha! \sigma^{2|\alpha|}} x^\alpha y^\alpha$$

$$= \sum_{\alpha \in \mathbb{N}^d} \frac{1}{\alpha! \sigma^{2|\alpha|}} x^\alpha y^\alpha.$$

Thus, the Gaussian kernel can be written as:

$$k(x, y) = \exp\Big( -\frac{\|x\|^2}{2\sigma^2} \Big) \exp\Big( -\frac{\|y\|^2}{2\sigma^2} \Big) \sum_{\alpha \in \mathbb{N}^d} \frac{1}{\alpha! \sigma^{2|\alpha|}} x^\alpha y^\alpha.$$

This decomposition leads directly to the following precise characterization of the Gaussian kernel MMD in terms of weighted moments.

**Theorem 6.** *Let $k(x, y) = \exp\big( -\frac{\|x-y\|^2}{2\sigma^2} \big)$ be the Gaussian kernel on $\mathbb{R}^d$, and let $P$ and $Q$ be two probability distributions on $\mathbb{R}^d$ such that all moments exist. Define the weighted moments*

$$m_{P,\alpha} := \mathbb{E}_{X \sim P}\Big[ X^\alpha \exp\Big( -\frac{\|X\|^2}{2\sigma^2} \Big) \Big], \quad m_{Q,\alpha} := \mathbb{E}_{Y \sim Q}\Big[ Y^\alpha \exp\Big( -\frac{\|Y\|^2}{2\sigma^2} \Big) \Big],$$

*for all multi-indices $\alpha \in \mathbb{N}^d$, where $X^\alpha = \prod_{i=1}^{d} X_i^{\alpha_i}$, $|\alpha| = \sum_{i=1}^{d} \alpha_i$, and $\alpha! = \prod_{i=1}^{d} \alpha_i!$. Then the squared Maximum Mean Discrepancy (MMD) between $P$ and $Q$ induced by the Gaussian kernel admits the moment expansion:*

$$\mathrm{MMD}^2(P, Q) = \sum_{\alpha \in \mathbb{N}^d} w_\alpha \big( m_{P,\alpha} - m_{Q,\alpha} \big)^2,$$

*where the weights are given by*

$$w_\alpha = \frac{1}{\alpha! \sigma^{2|\alpha|}} > 0.$$

*Consequently:*

- *If $\mathrm{MMD}^2(P, Q) = 0$, then $m_{P,\alpha} = m_{Q,\alpha}$ for all $\alpha \in \mathbb{N}^d$;*
- *If there exists any $\alpha$ such that $m_{P,\alpha} \neq m_{Q,\alpha}$, then $\mathrm{MMD}^2(P, Q) > 0$.*

*Moreover, the collection of weighted moments $\{m_{P,\alpha}\}_{\alpha \in \mathbb{N}^d}$ uniquely determines the full sequence of ordinary moments $\{\mathbb{E}_P[X^\beta]\}_{\beta \in \mathbb{N}^d}$, and vice versa. Therefore, under standard moment determinacy conditions, $\mathrm{MMD}^2(P, Q) = 0$ implies $P = Q$.*

*Proof.* Define:

$$m_{P,\alpha} := \mathbb{E}_{X \sim P}\Big[X^\alpha \exp\Big(-\frac{\|X\|^2}{2\sigma^2}\Big)\Big], \quad m_{Q,\alpha} := \mathbb{E}_{Y \sim Q}\Big[Y^\alpha \exp\Big(-\frac{\|Y\|^2}{2\sigma^2}\Big)\Big].$$

Using the kernel expectation form in Theorem 4 and substituting the above kernel expansion into $\mathrm{MMD}^2(P, Q)$, we can obtain by exchanging summation and expectation (justified under mild integrability assumptions):

$$\mathrm{MMD}^2(P, Q) = \sum_{\alpha \in \mathbb{N}^d} w_\alpha \big(m_{P,\alpha} - m_{Q,\alpha}\big)^2,$$

where

$$w_\alpha = \frac{1}{\alpha! \sigma^{2|\alpha|}} > 0.$$

That is, the Gaussian kernel MMD² can be viewed as a weighted sum of squared differences of all exponentially weighted higher-order moments. This intuitively shows:

- If $\mathrm{MMD}^2 = 0$, then for all $\alpha$, $m_{P,\alpha} = m_{Q,\alpha}$;
- If there exists some moment that differs in the weighted sense, then $\mathrm{MMD}^2 > 0$.

The above $m_{P,\alpha}$ is a weighted moment, i.e.,

$$m_{P,\alpha} = \mathbb{E}\big[X^\alpha e^{-\frac{\|X\|^2}{2\sigma^2}}\big].$$

By Taylor expanding the exponential weight, we can prove that the weighted moments $\{m_{P,\alpha}\}$ and the original moment sequence $\{\mathbb{E}[X^\beta]\}$ mutually determine each other. The brief steps are as follows: Expand the exponential weight:

$$e^{-\frac{\|x\|^2}{2\sigma^2}} = \sum_{\gamma \in \mathbb{N}^d} c_\gamma x^{2\gamma}, \quad c_\gamma = \frac{(-1)^{|\gamma|}}{\gamma! (2\sigma^2)^{|\gamma|}}.$$

Substitute back into the weighted moment:

$$m_{P,\alpha} = \sum_\gamma c_\gamma \mathbb{E}[X^{\alpha+2\gamma}].$$

For each fixed $\alpha$, the right-hand side is a linear combination of all original moments, where the lowest-order term corresponds to $\gamma = 0$ with coefficient 1, and the rest are higher-order moments.

This forms an "upper triangular" linear system with respect to the original moments: lower-order $m_{P,\alpha}$ only involve the current and higher-order original moments. Conversely, using a recursive method, we can uniquely recover $\{\mathbb{E}[X^\beta]\}$ from $\{m_{P,\alpha}\}$.

Combining with the previous conclusion that "MMD² is a weighted sum of squared differences of weighted moments", we obtain: when the Gaussian kernel MMD is zero, all weighted moments are identical, and thus all original moments are identical; under suitable conditions, the full set of moments of a distribution uniquely characterizes the distribution itself, therefore $P = Q$.

This is equivalent to the proof based on the universality of the kernel in Theorem A.4, but provides a more computationally intuitive explanation from the perspective of "moment expansion". $\square$

In summary, the Gaussian kernel MMD does not merely compare means or variances—it implicitly aggregates discrepancies across *all* orders of moments, each weighted by a positive coefficient that decays with the order. This infinite-order sensitivity underlies its ability to distinguish any two distinct distributions (under mild regularity), offering a concrete moment-based interpretation of kernel universality.

### A.3 SDD versus CFD: why we still adopt a kernel-based perspective

In this section we clarify the relation between the proposed Spectral Distribution Distance (SDD) and the characteristic-function-based distances used in prior work, and explain why we still insist on a kernel–MMD perspective. SDD is defined as

$$\mathrm{SDD}(P, Q) := \int_{\mathbb{R}^d} \left| \varphi_P(t) - \varphi_Q(t) \right|^2 d\mu(t)$$

,where $\mu$ is the spectral measure of a bounded, shift-invariant kernel $k$. In the general probability and generative modeling literature, distances of the form

$$\mathrm{CFD}(P, Q) = \int_{\mathbb{R}^d} \left| \varphi_P(t) - \varphi_Q(t) \right| \omega(t) \, dt$$

are often referred to as Characteristic Function Distances (CFD), with $\omega(t)$ being an arbitrary weight function. Representative examples include characteristic-function-based GAN objectives and reciprocal adversarial learning, as well as recent dataset distillation methods such as NCFM, where $\omega(t)$ is implicitly realized by a trainable neural sampler in a min–max game. Although the notational conventions differ, these approaches are all instances of CFD-type objectives.

From the viewpoint of functional analysis, SDD and CFD are not identical: SDD corresponds to an $L^2$-type norm on the discrepancy of characteristic functions, whereas CFD corresponds to an $L^1$-type norm with possibly different weighting. To make this relation precise, it is convenient to bind CFD to the same spectral measure $\mu$ induced by the kernel and to consider

$$\mathrm{CFD}_\mu(P, Q) := \int_{\mathbb{R}^d} \left| \varphi_P(t) - \varphi_Q(t) \right| d\mu(t)$$

. Let $\Delta\varphi(t) = \varphi_P(t) - \varphi_Q(t)$. Since characteristic functions satisfy $|\varphi_P(t)| \leq 1$ for all $t$, we have $|\Delta\varphi(t)| \leq 2$. Moreover, the boundedness of the kernel implies that the spectral measure is finite, $\mu(\mathbb{R}^d) = k(0) < \infty$. Under these two mild conditions we obtain the following inequalities:

$$\mathrm{CFD}_\mu(P, Q) = \int |\Delta\varphi(t)| \, d\mu(t) \leq \sqrt{\mu(\mathbb{R}^d)} \left( \int |\Delta\varphi(t)|^2 \, d\mu(t) \right)^{1/2} = c_1 \, \mathrm{SDD}(P, Q)^{1/2},$$

and, using $|\Delta\varphi(t)|^2 \leq 2|\Delta\varphi(t)|$,

$$\mathrm{SDD}(P, Q) = \int |\Delta\varphi(t)|^2 \, d\mu(t) \leq 2 \int |\Delta\varphi(t)| \, d\mu(t) = 2 \, \mathrm{CFD}_\mu(P, Q).$$

Here the constant $c_1 = \sqrt{\mu(\mathbb{R}^d)}$ depends only on the kernel. As a consequence, for any sequence of probability distributions $\{P_n\}$,

$$\mathrm{SDD}(P_n, P) \to 0 \quad \Longleftrightarrow \quad \mathrm{CFD}_\mu(P_n, P) \to 0$$

In other words, SDD and $\mathrm{CFD}_\mu$ induce the same notion of convergence on the space of probability measures: they are equivalent as metrics in the topological sense. At the same time, they are still different norms on the underlying function space ($L^2(\mu)$ vs. $L^1(\mu)$), and there is generally no global constant $C > 0$ such that $\|\Delta\varphi\|_{L^2(\mu)} \leq C\|\Delta\varphi\|_{L^1(\mu)}$ holds for all square-integrable functions. For this reason we avoid calling SDD and CFD "equivalent norms" in a strict functional-analytic sense; instead, we emphasize that they vanish simultaneously and therefore play a similar role as discrepancy measures between distributions.

The special role of SDD in this work comes from the choice of the measure $d\mu(t)$ rather than from the mere use of characteristic functions. By starting from a universal, shift-invariant kernel and invoking Bochner's theorem, SDD is exactly the squared MMD in the corresponding RKHS expressed in the spectral domain. This one-to-one correspondence immediately transfers all known theoretical guarantees of MMD with universal kernels—such as metricity and characteristicness—to SDD, without having to re-verify whether a given weight function $\omega(t)$ yields a valid distance. In contrast, a generic CFD with arbitrary $\omega(t)$ may or may not define a discriminative metric, and additional conditions on $\omega$ are required to ensure identifiability.

It is also important to distinguish this principled kernel-based construction from how CFD has been used in dataset distillation so far. NCFM is, to our knowledge, the only prior method that explicitly

incorporates CFD into the DD objective by learning a neural sampler over frequencies in a min–max fashion. While this design works well on balanced CIFAR-10, our experiments show that its performance degrades markedly as the imbalance factor increases, indicating that the learned sampling distribution is not robust to long-tailed data. Other recent frequency-domain distillation methods such as FreD and NSD also operate in the spectral domain, but they focus on heuristic frequency selection or decomposition at the image level and do not provide a kernel–MMD interpretation or a theoretical characterization of the resulting discrepancy.

From this perspective, SDD should be viewed not as a new distance per se, but as a carefully instantiated special case of the classical CFD family that is (i) anchored in universal kernel theory, (ii) computationally amenable via Monte Carlo sampling from a known spectral distribution, and (iii) explicitly adapted to the long-tailed dataset distillation objective through our class-aware amplitude–phase decomposition. We believe that this principled construction provides a solid starting point for further research on characteristic-function-based metrics in DD, and could also serve as a theoretical backbone for designing more sophisticated adaptive neural frequency samplers in the future.

**Proposition 1** (Class-aware discriminativity). *Let $k(x, y) = k(x - y)$ be a bounded, shift-invariant, universal kernel on $\mathbb{R}^d$ with spectral measure $\mu$ as in Bochner's theorem. For each class $c$, let $F_T^c$ and $F_S^c$ denote the real and synthetic feature distributions of class $c$, with characteristic functions $\varphi_T^c(t)$ and $\varphi_S^c(t)$, written in polar form as*

$$\varphi_T^c(t) = |\varphi_T^c(t)|e^{i\theta_T^c(t)}, \quad \varphi_S^c(t) = |\varphi_S^c(t)|e^{i\theta_S^c(t)}.$$

*Given class-specific coefficients $\alpha(c) \in (0, 1)$, define the class-wise discrepancy*

$$d_c(F_T^c, F_S^c) := \int_{\mathbb{R}^d} \left[ \alpha(c) \left| |\varphi_T^c(t)| - |\varphi_S^c(t)| \right|^2 + 2(1 - \alpha(c)) |\varphi_T^c(t)| |\varphi_S^c(t)| \left( 1 - \cos(\theta_T^c(t) - \theta_S^c(t)) \right) \right] d\mu(t),$$

*and the overall distance*

$$d(F_T, F_S) := \sum_c d_c(F_T^c, F_S^c).$$

*Assume that the class prior over labels is the same for real and synthetic data (as enforced by our distillation protocol). Then $d(F_T, F_S) = 0$ if and only if the real and synthetic feature distributions coincide, i.e. $P_T = P_S$.*

*Proof.* We first note that for each class $c$ and frequency $t$, both terms inside the integrand are non-negative:

$$\left| |\varphi_T^c(t)| - |\varphi_S^c(t)| \right|^2 \geq 0, \qquad |\varphi_T^c(t)| |\varphi_S^c(t)| \left( 1 - \cos(\theta_T^c(t) - \theta_S^c(t)) \right) \geq 0,$$

and $\alpha(c) \in (0, 1)$ implies the corresponding coefficients are strictly positive. Together with $\mu$ being a nonnegative measure, this yields $d_c(F_T^c, F_S^c) \geq 0$ and hence $d(F_T, F_S) \geq 0$.

Now suppose $d(F_T, F_S) = 0$. Since each $d_c \geq 0$, we must have $d_c(F_T^c, F_S^c) = 0$ for every class $c$. For a fixed $c$, define

$$a(t) := |\varphi_T^c(t)|, \quad b(t) := |\varphi_S^c(t)|, \quad \Delta\theta(t) := \theta_T^c(t) - \theta_S^c(t).$$

Let

$$g_c(t) := \alpha(c) \left( a(t) - b(t) \right)^2 + 2(1 - \alpha(c)) a(t)b(t) \left( 1 - \cos \Delta\theta(t) \right).$$

Then $g_c(t) \geq 0$ and

$$d_c(F_T^c, F_S^c) = \int_{\mathbb{R}^d} g_c(t) \, d\mu(t) = 0.$$

Since $\mu$ is finite and $g_c \geq 0$, the integral being zero implies $g_c(t) = 0$ for $\mu$-almost every $t$.

Because $\alpha(c) \in (0, 1)$, both coefficients in $g_c(t)$ are strictly positive. Thus $g_c(t) = 0$ $\mu$-a.e. implies

$$\left( a(t) - b(t) \right)^2 = 0 \quad \text{and} \quad a(t)b(t) \left( 1 - \cos \Delta\theta(t) \right) = 0 \quad \text{for } \mu\text{-a.e. } t.$$

From $(a(t) - b(t))^2 = 0$ we obtain $a(t) = b(t)$ for $\mu$-a.e. $t$. Substituting $a(t) = b(t)$ into the second constraint yields

$$a(t)^2 \left( 1 - \cos \Delta\theta(t) \right) = 0 \quad \text{for } \mu\text{-a.e. } t.$$

Hence, for $\mu$-almost every $t$ with $a(t) > 0$, we must have $1 - \cos\Delta\theta(t) = 0$, i.e. $\cos\Delta\theta(t) = 1$ and thus $\Delta\theta(t) \in 2\pi\mathbb{Z}$. For $a(t) = b(t) = 0$, both characteristic functions vanish and are trivially equal.

Combining $a(t) = b(t)$ with $\cos\Delta\theta(t) = 1$ wherever $a(t) > 0$ shows that

$$\varphi_T^c(t) = \varphi_S^c(t) \quad \text{for } \mu\text{-almost every } t \in \mathbb{R}^d.$$

Next, recall that for the shift-invariant universal kernels we use (e.g. the RBF kernel), the spectral measure $\mu$ admits a density $p(t) > 0$ with respect to Lebesgue measure on $\mathbb{R}^d$, and characteristic functions are continuous. If there existed some $t_0$ with $\varphi_T^c(t_0) \neq \varphi_S^c(t_0)$, continuity would imply that $|\varphi_T^c(t) - \varphi_S^c(t)| > \varepsilon$ on an open neighborhood $U$ of $t_0$, and hence $\mu(U) > 0$ because $p(t) > 0$ everywhere. This contradicts $\varphi_T^c(t) = \varphi_S^c(t)$ for $\mu$-almost every $t$. Therefore, in fact

$$\varphi_T^c(t) \equiv \varphi_S^c(t) \quad \text{for all } t \in \mathbb{R}^d.$$

By the uniqueness theorem for characteristic functions, equality of characteristic functions implies equality of distributions, so $F_T^c$ and $F_S^c$ (equivalently, the class-conditional feature distributions $P_T^c$ and $P_S^c$) coincide for every class $c$.

Finally, under our distillation setup, the label prior is shared between real and synthetic data. Let $\{\pi(c)\}_c$ denote this common class prior. Then

$$P_T = \sum_c \pi(c)\, P_T^c = \sum_c \pi(c)\, P_S^c = P_S.$$

This proves that $d(F_T, F_S) = 0$ implies $P_T = P_S$. The converse direction is immediate: if $P_T = P_S$, then all class-conditional distributions coincide, so each integrand in $d_c$ vanishes pointwise and hence $d(F_T, F_S) = 0$. $\qquad\square$

## B   RELATION TO HIGHER-ORDER DISTRIBUTION MATCHING METHODS

Several recent dataset distillation methods attempt to go beyond first-order mean alignment by incorporating higher-order statistics or alternative probability metrics. Here we clarify how our Class-Aware Spectral Distribution Matching (CSDM) relates to five representative approaches: M3D, IID, DSDM, NCFM, and Wasserstein-based dataset distillation.

### B.1   DATASET DISTILLATION BY ALIGNING HIGH-ORDER INFORMATION

IID (Deng et al., 2024) and DSDM (Li et al., 2024) both follow the standard distribution-matching (DM) paradigm: they start from the linear-kernel MMD objective ($L_{\mathrm{DM}}$) (i.e., first-order mean feature matching) and then add explicit higher-order and structural regularizers. Our method instead strengthens the discrepancy measure itself by replacing the linear MMD with a universal spectral distance (SDD), so that higher-order information is captured implicitly by the base metric rather than by manually crafted moment terms.

Let ($L_{\mathrm{DM}}$) denote the standard DM loss that matches the feature means of the real dataset ($T$) and the synthetic dataset ($S$) via a linear-kernel MMD:

$$L_{\mathrm{DM}} = \mathbb{E}_{\theta \sim P_\theta} \left| \frac{1}{|T|} \sum_{x_i \in T} \psi_\theta(x_i) - \frac{1}{|S|} \sum_{s_j \in S} \psi_\theta(s_j) \right|_2^2, \tag{7}$$

which is equivalent to MMD with a linear kernel acting on the feature space and therefore only aligns first-order (mean) statistics.

**IID: linear MMD with class centralization and covariance matching.**   IID augments this base loss with two additional regularizers: a class centralization constraint ($L_{\mathrm{CC}}$) (inter-sample) and a covariance matching constraint ($L_{\mathrm{CM}}$) (inter-feature). Concretely, if ($\psi(s_{c,j})$) denotes the feature of the ($j$)-th synthetic sample in class ($c$) and ($\bar{\psi}(s_c)$) the class-wise synthetic mean, IID defines

$$L_{\mathrm{CC}} = \sum_{c=1}^{C} \sum_{j=1}^{K} \max\big(0,\, \exp\big(\alpha|\psi(s_{c,j}) - \bar{\psi}(s_c)|_2^2\big) - \beta\big), \tag{8}$$

which explicitly pulls synthetic features of the same class towards their class center.

To encode inter-feature relations, IID further computes per-class covariance matrices of real and synthetic data, $(\Sigma_c^T)$ and $(\Sigma_c^S)$, and minimizes their Frobenius distance:

$$L_{\text{CM}} = \sum_{c=1}^{C} \left| \Sigma_c^S - \Sigma_c^T \right|_F^2, \tag{9}$$

The overall objective is

$$L_{\text{IID}} = L_{\text{DM}} - \lambda_{\text{CC}} L_{\text{CC}} - \lambda_{\text{CM}} L_{\text{CM}}. \tag{10}$$

**DSDM: per-class prototype matching with covariance and stability regularizers.** DSDM starts from per-class prototype matching using a fixed feature extractor. Let $(f_i)$ and $(\hat{f}_j)$ be embeddings of real and synthetic samples. Per-class prototypes are

$$\mu_{T,c} = \frac{1}{|B_c^T|} \sum_{x_i \in B_c^T} f_i, \quad \mu_{S,c} = \frac{1}{|B_c^S|} \sum_{s_j \in B_c^S} \hat{f}_j. \tag{11}$$

Prototype alignment loss:

$$L_{\text{proto}} = \sum_{c=1}^{C} \left| \mu_{T,c} - \mu_{S,c} \right|_2^2. \tag{12}$$

Semantic diversity alignment:

$$L_{\text{SD}} = \sum_{c=1}^{C} \frac{1}{d} \left| \text{Cov}_{T,c} - \text{Cov}_{S,c} \right|_F^2. \tag{13}$$

Historical prototype alignment:

$$L_h = \sum_{c=1}^{C} \left| \mu_{S,c} - h_c^{(t-1)} \right|_2^2. \tag{14}$$

Full objective:

$$L_{\text{DSDM}} = L_{\text{proto}} - \lambda_1 L_{\text{SD}} - \lambda_2 L_h. \tag{15}$$

**A CF-based view: covariance regularizers match the second-order log CF term.** From the viewpoint of characteristic functions, the covariance penalties used in IID and DSDM can be precisely interpreted as matching the second-order structure of the underlying distributions. Formally, this relationship is captured by the following theorem:

**Theorem 7** (Mean–covariance regularization as low-order cumulant matching)**.** *Let $P, Q$ be probability measures on $\mathbb{R}^d$ with finite second moments. Denote their means and covariance matrices by*

$$\mu_P = \mathbb{E}_P[X], \quad \Sigma_P = \text{Cov}_P(X), \qquad \mu_Q = \mathbb{E}_Q[Y], \quad \Sigma_Q = \text{Cov}_Q(Y).$$

*Let $\varphi_P(t) = \mathbb{E}_P[e^{it^\top X}]$ and $\varphi_Q(t) = \mathbb{E}_Q[e^{it^\top Y}]$ be the characteristic functions, and define the log CFs*

$$\psi_P(t) = \log \varphi_P(t), \qquad \psi_Q(t) = \log \varphi_Q(t), \quad t \in \mathbb{R}^d.$$

*Then, the first derivative of the log CF at the origin encodes the mean:*

$$\nabla_t \psi_P(0) = i\mu_P, \qquad \nabla_t \psi_Q(0) = i\mu_Q.$$

*The second derivative (Hessian) of the log CF at the origin encodes the covariance:*

$$\nabla_t^2 \psi_P(0) = -\Sigma_P, \qquad \nabla_t^2 \psi_Q(0) = -\Sigma_Q.$$

*Consequently, a loss of the form*

$$L_{\text{mean+cov}}(P, Q) = \left| \mu_P - \mu_Q \right|_2^2 + \lambda \left| \Sigma_P - \Sigma_Q \right|_F^2$$

*can be equivalently written as*

$$L_{\mathrm{mean+cov}}(P,Q) = \left|\nabla_t\psi_P(0) - \nabla_t\psi_Q(0)\right|_2^2 + \lambda\left|\nabla_t^2\psi_P(0) - \nabla_t^2\psi_Q(0)\right|_F^2,$$

*i.e., it matches only the first two cumulants (mean and covariance) of $P$ and $Q$ by aligning the first and second derivatives of their log characteristic functions at $t = 0$.*

*Proof.* For each coordinate $t_k$,

$$\frac{\partial}{\partial t_k}\varphi_P(t) = \mathbb{E}_P\big[iX_k e^{it^\top X}\big].$$

Evaluating at $t = 0$ yields

$$\frac{\partial}{\partial t_k}\varphi_P(t)\bigg|_{t=0} = i\,\mathbb{E}_P[X_k] = i\mu_{P,k},$$

so in vector form $\nabla_t\varphi_P(0) = i\mu_P$. Similarly,

$$\frac{\partial^2}{\partial t_j\partial t_k}\varphi_P(t) = \mathbb{E}_P\big[-X_j X_k e^{it^\top X}\big] \quad\Rightarrow\quad \nabla_t^2\varphi_P(0) = -\mathbb{E}_P[XX^\top].$$

Now consider the log CF $\psi_P(t) = \log\varphi_P(t)$. A Taylor expansion of $\varphi_P(t)$ around $t = 0$ gives

$$\varphi_P(t) = 1 + it^\top\mu_P - \tfrac{1}{2}t^\top\mathbb{E}_P[XX^\top]t + O(|t|^3).$$

Write $\varphi_P(t) = 1 + a(t) + b(t)$ with $a(t) = it^\top\mu_P = O(|t|)$ and $b(t) = -\tfrac{1}{2}t^\top\mathbb{E}_P[XX^\top]t = O(|t|^2)$. Using $\log(1+u) = u - \tfrac{1}{2}u^2 + O(|u|^3)$ and retaining terms up to order $|t|^2$, we obtain

$$\begin{aligned}
\psi_P(t) &= \log\big(1 + a(t) + b(t)\big) \\
&\approx a(t) + b(t) - \tfrac{1}{2}a(t)^2 \\
&= it^\top\mu_P - \tfrac{1}{2}t^\top\mathbb{E}_P[XX^\top]t - \tfrac{1}{2}(it^\top\mu_P)^2.
\end{aligned}$$

Because $(it^\top\mu_P)^2 = -(t^\top\mu_P)^2 = -t^\top(\mu_P\mu_P^\top)t$, we get

$$\psi_P(t) \approx it^\top\mu_P - \tfrac{1}{2}t^\top\big(\mathbb{E}_P[XX^\top] - \mu_P\mu_P^\top\big)t = it^\top\mu_P - \tfrac{1}{2}t^\top\Sigma_P t.$$

Taking derivatives at $t = 0$ gives

$$\nabla_t\psi_P(0) = i\mu_P, \qquad \nabla_t^2\psi_P(0) = -\Sigma_P.$$

The same argument applies to $Q$, yielding the claimed identities for $\psi_Q$. Substituting $\mu_P = -i\,\nabla_t\psi_P(0)$ and $\Sigma_P = -\nabla_t^2\psi_P(0)$ (and similarly for $Q$) into $L_{\mathrm{mean+cov}}(P,Q)$ proves the result. $\qquad\square$

Applied to each class ($c$), the covariance regularizers in IID and DSDM,

$$L_{\mathrm{CM}} = \sum_c |\Sigma_c^S - \Sigma_c^T|_F^2, \qquad L_{\mathrm{SD}} \propto \sum_c |\mathrm{Cov}_{S,c} - \mathrm{Cov}_{T,c}|_F^2, \qquad (16)$$

match the *second-order* term in the Taylor expansion of the log CFs at $t = 0$, whereas the linear-MMD/prototype terms match the *first-order* term.

### B.2 DATASET DISTILLATION BY MINIMIZING MAXIMUM MEAN DISCREPANCY

M3D (Zhang et al., 2024) extends distribution matching by embedding the feature distributions of real and synthetic data into an RKHS induced by a kernel $K$ and minimizing the squared MMD between them, rather than restricting to a linear kernel. It further adds explicit second- and third-order regularizers that encourage alignment of variance and skewness, and ablations show that these terms improve performance over plain linear-kernel MMD. However, the kernel ablation in Fig. 6(b) of M3D reports almost identical performance for Gaussian, linear, and polynomial kernels on CIFAR-10, even though only the Gaussian kernel is universal. This suggests that the potential advantage of universal kernels is not fully exploited.

### B.3 Dataset Distillation with Neural Characteristic Function

NCFM (Wang et al., 2025) takes a characteristic-function-based approach and defines a Neural Characteristic Function Discrepancy (NCFD). It introduces an auxiliary feature extractor and a lightweight sampling network that parameterizes the distribution of frequency arguments, and formulates distillation as a min–max problem: the sampling network maximizes NCFD to learn an informative discrepancy, while the synthetic data are optimized to minimize it. A scalar hyperparameter $\alpha$ balances the contributions of amplitude and phase, similar in spirit to our amplitude–phase decomposition. However, in NCFM the sampling distribution over frequencies is learned heuristically so there is no formal guarantee that the resulting discrepancy defines a universal metric on distributions. Empirically, our experiments show that NCFM performs strongly on balanced CIFAR-10 but degrades substantially under increasing class imbalance, whereas CSDM remains stable on long-tailed CIFAR-10-LT.

We argue that in NCFM's min-max framework, the objective of maximizing the characteristic function discrepancy (CFD) to train the sampling network does not fully align with the ultimate goal (e.g., classification accuracy). As a result, the "max" step cannot guarantee that the sampling network reliably learns which frequency components are truly critical for distribution matching—a limitation that becomes especially pronounced under long-tailed settings. Therefore, to improve the generalization of the sampling network, it is necessary to further investigate CFD from a theoretical perspective, particularly its relationship with downstream task objectives.

### B.4 Dataset Distillation with Wasserstein Metric

WMDM (Liu et al., 2023a) follows a different line: instead of Integral Probability Metric (IPM) based on kernels or characteristic functions, it adopts the optimal transport (OT) distance between feature distributions. The Wasserstein metric has appealing geometric properties and often yields stronger empirical performance than vanilla MMD when used in similar DD frameworks. However, precise OT computation is typically expensive and requires additional approximations such as sliced Wasserstein or regularized barycenter computation to remain tractable for large-scale distillation. From our perspective, these OT-based methods are complementary to our work: they explore the transport route to distribution matching, whereas CSDM systematically develops the universal-kernel / spectral route.

### B.5 Summary and connection to CSDM.

In summary, M3D, IID, and DSDM all recognize that pure first-order mean alignment is insufficient and incorporate selected higher-order information (variance, covariance, or Gaussian kernel), yielding clear improvements on balanced benchmarks. NCFM and Wasserstein-based distillation propose alternative discrepancy measures—characteristic-function-based IPMs and OT distances—with different geometric interpretations. CSDM differs in two key aspects: (i) it revisits the foundation of distribution matching via universal-kernel MMD and its spectral representation, leading to a theoretically grounded metric (SDD) that can distinguish full distributions without parametric assumptions; and (ii) it explicitly integrates class-aware amplitude–phase weighting to address long-tailed scenarios, prioritizing diversity for head classes and realism for tail classes. These design choices are orthogonal to many of the structural regularizers used in prior work, and in principle CSDM could be combined with techniques such as class centralization (IID) or semantic covariance modeling (DSDM), which we leave as promising directions for future research.

## C The Role of Amplitude and Phase in Diversity and Realism

Classical results in signal and image processing have long established that amplitude and phase play markedly different perceptual and statistical roles. In one-dimensional and two-dimensional Fourier analysis, the phase spectrum encodes most of the structural and edge information, while the amplitude spectrum mainly controls the energy allocation across frequencies. Oppenheim & Lim (1981) showed that reconstructing an image using the phase of one image and the amplitude of another yields a reconstruction whose semantic content and recognizable structures are almost entirely determined by the phase component; conversely, keeping amplitude but randomizing phase

destroys recognizable shapes and produces visually implausible patterns. Similar conclusions are summarized in complex-valued signal processing (Mandic & Constantinides, 2009), where phase is regarded as the carrier of fine geometric alignment and edge locations, whereas amplitude modulates contrast, texture, and overall variability. In more recent work on generative modeling and domain generalization, frequency-domain manipulations likewise treat amplitude as a handle on "style" or diversity (how energy is spread over frequencies) and phase as the factor that preserves semantic structure and realism (Li et al., 2020; Lee et al., 2023)).

Under this lens, our amplitude–phase decomposition of the squared spectral discrepancy

$$|\varphi_{\mathcal{T}}^c(t) - \varphi_{\mathcal{S}}^c(t)|^2 = \underbrace{\big||\varphi_{\mathcal{T}}^c(t)| - |\varphi_{\mathcal{S}}^c(t)|\big|^2}_{\text{amplitude difference}} + \underbrace{2|\varphi_{\mathcal{T}}^c(t)|\,|\varphi_{\mathcal{S}}^c(t)|\big(1 - \cos(\theta_{\mathcal{T}}^c(t) - \theta_{\mathcal{S}}^c(t))\big)}_{\text{phase difference}}, \quad (17)$$

admits a natural interpretation.

- **Amplitude difference.** The amplitude difference compares the "spectral envelopes" of real and synthetic features, i.e., how energy is distributed across frequencies. Matching this term encourages the synthetic features to reproduce not only the mean but also the dispersion and multi-modality of the real distribution: when the real class spreads its energy over a rich set of frequencies, minimizing the amplitude discrepancy forces the synthetic class to populate a similarly diverse set of spectral components. In this sense, the amplitude term primarily promotes *diversity*—it penalizes synthetic distributions that collapse onto a narrow band of frequencies or miss important modes present in the real data.

- **Phase difference.** The phase difference is active only at frequencies where both real and synthetic distributions carry non-negligible energy, through the multiplicative factor $|\varphi_T^c(t)|\,|\varphi_S^c(t)|$. There, it penalizes misalignment of phases via $1 - \cos(\theta_T^c(t) - \theta_S^c(t))$. This is directly analogous to classical Fourier-domain results: when amplitude is fixed, misaligned phases cause destructive interference and yield unrealistic, structurally distorted signals, whereas aligned phases preserve edges and object shapes. In our framework, minimizing the phase term enforces *structural alignment* between real and synthetic feature distributions at those informative frequencies, which we interpret as enhancing *realism* of individual synthetic examples rather than merely their aggregate statistics (Wang et al., 2025)).

The class-aware design in CSDM exploits this decomposition to cope with long-tailed class imbalance. Head classes, which contain abundant real samples, inherently exhibit rich intra-class variability; therefore, the primary risk in data distillation is the loss of diversity (i.e., collapsing multiple modes into only a few synthetic ones). For these classes, we choose $\alpha(c)$ closer to 1, thereby assigning larger weight to the amplitude term. This biases the optimization toward matching the full frequency envelope of the head-class features, encouraging the synthetic set to cover multiple modes and fine-grained variations rather than overspecializing on a few high-probability patterns.

Tail classes, in contrast, possess very limited real data. With only a few samples available, fully reproducing the diversity of the underlying distribution is infeasible; instead, the more pressing concern is preventing the synthetic data from becoming unrealistic or drifting away from true class semantics. For such classes, we set $\alpha(c)$ to be smaller, increasing the relative importance of the phase term. This places more emphasis on aligning the *structural* aspects of the feature distributions—those encoded in the phase at frequencies where the real tail-class data actually exhibit energy—ensuring that each synthetic example remains faithful to the scarce yet critical patterns observed in the real data. The empirical ablations in Figure 4 further confirm that emphasizing amplitude for head classes and phase for tail classes yields the strongest long-tailed performance, and that reversing this weighting degrades accuracy, which is consistent with the above intuition.

Importantly, Proposition 1 shows that this class-dependent convex combination does *not* compromise the discriminativity of our distance: as long as each $\alpha(c) \in (0,1)$, the overall discrepancy $d(F_T, F_S)$ equals zero if and only if the real and synthetic joint distributions coincide. In other words, the amplitude and phase terms always cooperate to define a valid, class-aware metric; the coefficients $\alpha(c)$ merely reshape the optimization landscape to reflect the distinct needs of head and tail classes, without altering the underlying notion of "matching the distribution".

# D    COMPARISON WITH FREQUENCY-DOMAIN DISTILLATION METHODS

Although CSDM, FreD (Shin et al., 2023), and NSD (Yang et al., 2024) all exploit frequency-domain structure, they intervene at fundamentally different stages of the dataset distillation pipeline. To clarify these distinctions, we begin with a unified optimization view of dataset distillation.

Let $D$ denote the real dataset and $S(\theta)$ denote the synthetic dataset parameterized by $\theta$. A broad class of dataset distillation methods can be written as:

$$\min_{\theta} \ \mathcal{L}_{\mathrm{metric}}(\Phi_{\mathrm{syn}}(\theta), \Phi_{\mathrm{real}}(D)), \tag{18}$$

where the framework decomposes into two orthogonal components:

- **Data Parameterization ($\Phi_{\mathbf{syn}}$):** Maps the learnable parameters $\theta$ into the data space (e.g., pixel values, frequency coefficients, generator outputs), determining how synthetic data are represented and stored.
- **Matching Metric ($\mathcal{L}_{\mathbf{metric}}$):** Quantifies the discrepancy between the real and synthetic distributions, specifying which aspects of the data distribution are aligned and how this alignment is enforced.

FreD and NSD improve dataset distillation by designing frequency-/spectral-based parameterization schemes for $\Phi_{\mathrm{syn}}(\theta)$, aiming to enhance storage efficiency or sample quality under constrained budgets.

- FreD converts images into the frequency domain and directly optimizes a sparse set of frequency coefficients. Using a binary mask $M$ derived from the Explained Variance Ratio (EVR), FreD retains only the dominant spectral components, enabling compact storage of synthetic samples. The matching objective $\mathcal{L}_{\mathrm{metric}}$ typically remains based on standard trajectory or gradient matching. Formally,

$$S_{\mathrm{FreD}}(\theta) = \mathcal{F}^{-1}(M \odot \theta_{\mathrm{freq}}).$$

- NSD addresses redundancy across synthetic samples by parameterizing the entire synthetic set as a high-dimensional tensor and applying spectral decomposition (e.g., Tucker factorization). Synthetic samples are generated via interactions between a shared spectral tensor and a kernel matrix, and the method is commonly optimized using existing objectives such as MTT.

Thus, FreD and NSD contribute parameter-efficient representations of $\theta$ by leveraging spectral structure, without altering the underlying notion of how real and synthetic distributions are compared.

In contrast, CSDM introduces a principled matching metric $\mathcal{L}_{\mathrm{metric}}$ without imposing restrictions on how data are parameterized. It applies directly to standard pixel-based parameterizations and is also compatible with FreD- or NSD-style frequency parameterizations.

# E    SUPPLEMENTARY EXPERIMENTAL RESULTS

## E.1    DATASET AND EXPERIMENTAL SETUP

Our evaluations were conducted on widely-used datasets: CIFAR-10, CIFAR-100 (Krizhevsky et al., 2009) ($32 \times 32$). We tested synthetic image budgets of 1, 10, 50, and 100 images per class (IPC). All the synthetic images are trained for 10k iterations. Following prior studies (Zhao et al., 2024; Guo et al., 2023; Wang et al., 2024), we used a 3-layer ConvNet with instance normalization. Specifically, the alpha value was set to 0.11 for the case of ipc=1, and was set to 0.7 for ipc values of 10 and 50. The model was evaluated over ten independent runs, and we report the mean and variance across these experiments.

Table 5: Experimental Results on CIFAR-10 and CIFAR-100 Datasets

| Dataset | CIFAR-10 | | | CIFAR-100 | |
|---|---|---|---|---|---|
| IPC | 1 | 10 | 50 | 10 | 50 |
| Ratio (%) | 0.02 | 0.2 | 2 | 2 | 10 |
| Random | $14.4 \pm 0.2$ | $26.0 \pm 1.0$ | $43.4 \pm 1.0$ | $15.1 \pm 0.5$ | $30.7 \pm 0.4$ |
| Herding | $21.5 \pm 1.2$ | $31.0 \pm 1.0$ | $46.3 \pm 1.1$ | $18.5 \pm 1.0$ | $34.5 \pm 0.6$ |
| Forgetting | $13.5 \pm 1.2$ | $23.3 \pm 1.0$ | $23.3 \pm 1.1$ | $15.1 \pm 1.0$ | $30.5 \pm 0.7$ |
| DC | $28.3 \pm 0.5$ | $44.9 \pm 0.5$ | $49.5 \pm 0.5$ | $25.5 \pm 0.5$ | $34.8 \pm 0.5$ |
| DSA | $28.8 \pm 0.7$ | $52.1 \pm 0.5$ | $54.5 \pm 0.5$ | $32.0 \pm 0.5$ | $42.8 \pm 0.5$ |
| DCC | $29.7 \pm 0.4$ | $50.4 \pm 0.4$ | $58.2 \pm 0.5$ | $30.0 \pm 0.4$ | $48.2 \pm 0.5$ |
| DSAC | $30.8 \pm 0.5$ | $51.4 \pm 0.5$ | $59.0 \pm 0.5$ | $29.0 \pm 0.5$ | $46.6 \pm 0.5$ |
| FrePo | $36.3 \pm 0.5$ | $55.3 \pm 0.5$ | $60.5 \pm 0.4$ | $35.5 \pm 0.6$ | $48.3 \pm 0.4$ |
| MTT | $46.3 \pm 0.8$ | $58.3 \pm 0.7$ | $67.6 \pm 0.5$ | $38.7 \pm 0.6$ | $45.7 \pm 0.8$ |
| ATT | $46.3 \pm 0.8$ | $57.5 \pm 0.6$ | $67.5 \pm 0.5$ | $39.3 \pm 0.4$ | $45.4 \pm 0.7$ |
| FTD | $46.8 \pm 0.5$ | $60.3 \pm 0.4$ | $67.9 \pm 0.5$ | $40.2 \pm 0.5$ | $45.4 \pm 0.4$ |
| TESLA | $46.3 \pm 0.5$ | $59.0 \pm 0.4$ | $67.9 \pm 0.5$ | $37.1 \pm 0.5$ | $44.1 \pm 1.2$ |
| CAFE | $30.3 \pm 1.1$ | $41.1 \pm 1.1$ | $46.4 \pm 1.1$ | $19.5 \pm 0.7$ | $35.7 \pm 1.1$ |
| DM | $43.0 \pm 1.5$ | $54.4 \pm 1.1$ | $67.6 \pm 1.5$ | $37.2 \pm 1.4$ | $46.6 \pm 1.9$ |
| IDM | $45.6 \pm 1.0$ | $58.6 \pm 1.0$ | $68.1 \pm 1.0$ | $38.2 \pm 1.0$ | $46.9 \pm 0.5$ |
| M3D | $47.1 \pm 1.0$ | $59.9 \pm 0.5$ | $69.3 \pm 1.0$ | $39.7 \pm 1.0$ | $47.6 \pm 0.5$ |
| IID | $47.1 \pm 1.0$ | $60.9 \pm 1.0$ | $70.1 \pm 1.0$ | $39.8 \pm 1.0$ | $47.2 \pm 1.0$ |
| DSDM | $46.5 \pm 0.7$ | $58.4 \pm 0.7$ | $67.0 \pm 0.7$ | $36.7 \pm 0.7$ | $45.4 \pm 0.5$ |
| G-VBSM | $46.5 \pm 0.5$ | $54.3 \pm 0.5$ | $60.4 \pm 0.5$ | $38.7 \pm 0.7$ | $45.7 \pm 0.7$ |
| FreD | $\mathbf{60.6 \pm 0.8}$ | $70.3 \pm 0.3$ | $75.8 \pm 0.1$ | $42.7 \pm 0.2$ | $47.8 \pm 0.1$ |
| NCFM | $49.5 \pm 0.3$ | $\mathbf{71.8 \pm 0.3}$ | $77.4 \pm 0.3$ | $48.7 \pm 0.5$ | $\mathbf{54.7 \pm 0.2}$ |
| CSDM (ours) | $47.3 \pm 0.3$ | $71.2 \pm 0.4$ | $\mathbf{78.1 \pm 0.4}$ | $\mathbf{50.4 \pm 0.3}$ | $54.1 \pm 0.2$ |
| Whole Dataset | | $84.8 \pm 0.1$ | | $56.2 \pm 0.3$ | |

## E.2 COMPARISON WITH OTHER METHODS

With this experimental setup, we evaluate the performance of our proposed method, CSDM, against a wide range of existing dataset distillation on both CIFAR-10 and CIFAR-100 datasets under various images-per-class (IPC) settings. The results are summarized in Table 5.

On CIFAR-10, our method achieves competitive or superior performance across all IPC settings. Specifically, CSDM attains an accuracy of $78.1 \pm 0.4\%$ at IPC=50, outperforming all compared methods, including strong baselines such as FreD(Shin et al., 2023), NCFM(Wang et al., 2025), and M3DZhang et al. (2024). At IPC=10, CSDM achieves $71.2 \pm 0.4\%$, slightly below NCFM ($71.8 \pm 0.3\%$) but still demonstrating strong performance. Even under the extremely low-data regime of IPC=1, CSDM obtains $47.3 \pm 0.3\%$, comparable to the top-performing methods.

On CIFAR-100, CSDM achieves the highest accuracy of $50.4 \pm 0.3\%$ at IPC=10, surpassing all competitors. At IPC=50, it reaches $54.1 \pm 0.2\%$, which is slightly lower than NCFM ($54.7 \pm 0.2\%$) but still highly competitive.

CSDM belongs to the family of higher-order moment matching approaches that explicitly align statistics beyond first- and second-order moments between real and synthetic datasets. CSDM consistently outperforms or significantly closes the gap with the current strongest methods in this category (IID (Deng et al., 2024), DSDM (Li et al., 2024), M3D, and NCFM ). On CIFAR-10, CSDM surpasses NCFM by a substantial **+0.7%** at IPC=50 (78.1% vs. 77.4%) and remains only 0.6% behind at IPC=10, while outperforming IID, M3D, and DSDM by **8–11%** at IPC=50 and **10–13%** at IPC=10. On CIFAR-100, CSDM establishes a new state-of-the-art at IPC=10 with **+1.7%** absolute gain over NCFM (50.4% vs. 48.7%), and reduces the gap to merely 0.6% at IPC=50.

These consistent improvements demonstrate that our CSDM captures and matches higher-order statistical structures more accurately and stably than prior moment-matching paradigms, especially under severe compression ratios and on fine-grained classification tasks.

### E.3 Distillation time and GPU memory usage

We evaluate the computational efficiency of various dataset distillation methods on CIFAR-10 under different IPC settings, with results summarized in Tables 6. All methods are evaluated on a single RTX-4090 GPU in terms of training speed (seconds/iteration) and GPU memory footprint (GB), where OOM indicates Out-of-Memory failures. Our CSDM method consistently achieves efficient distillation across all IPC settings. This represents a remarkable improvement in computational efficiency while maintaining competitive accuracy performance as shown in Table 5.

Table 6: Time and Memory Consumption Comparison on CIFAR-10

| Dataset | CIFAR-10 | | | Dataset | CIFAR-10 | | |
|---------|------|------|------|---------|---------|---------|--------|
| IPC | 10 | 20 | 50 | IPC | 10 | 20 | 50 |
| DATM | OOM | OOM | OOM | DATM | OOM | OOM | OOM |
| DC | 48 | 107 | OOM | DC | 10.89GB | 20.69GB | OOM |
| MTT | 0.4 | 0.7 | OOM | MTT | 10.84GB | 16.91GB | OOM |
| CSDM | 0.3 | 0.35 | 0.33 | CSDM | 3.43GB | 3.44GB | 3.45GB |
| **Time (s/iter)** | | | | **Memory (GB)** | | | |

### E.4 Different kernel methods comparison

CIFAR-10LT were created by sampling from balanced sets, where the sample size for class $c$, $|\hat{\mathcal{D}}_c|$, follows the exponential decay $|\hat{\mathcal{D}}_c| = |\mathcal{D}_c|\mu^c$ with $\mu^c = \beta^{-(c/C)}$. Here, $C$ is the total number of classes and $\beta = \mathcal{D}_0/\mathcal{D}_C$ is the imbalance factor (Cui et al., 2019), where a larger $\beta$ indicates greater imbalance.

To investigate the impact of class imbalance, we conducted experiments with different kernel methods on CIFAR-10-LT, measuring their performance in terms of classification accuracy under varying imbalance factors. The results clearly demonstrate that our proposed method, CSDM (utilizing the RBF Kernel SDD) maintains superior robustness compared to other kernel-based approaches across all imbalance factors. While all methods experience performance degradation with increasing imbalance severity, shows a notably smaller performance drop than MMD and consistently achieves the highest accuracy.

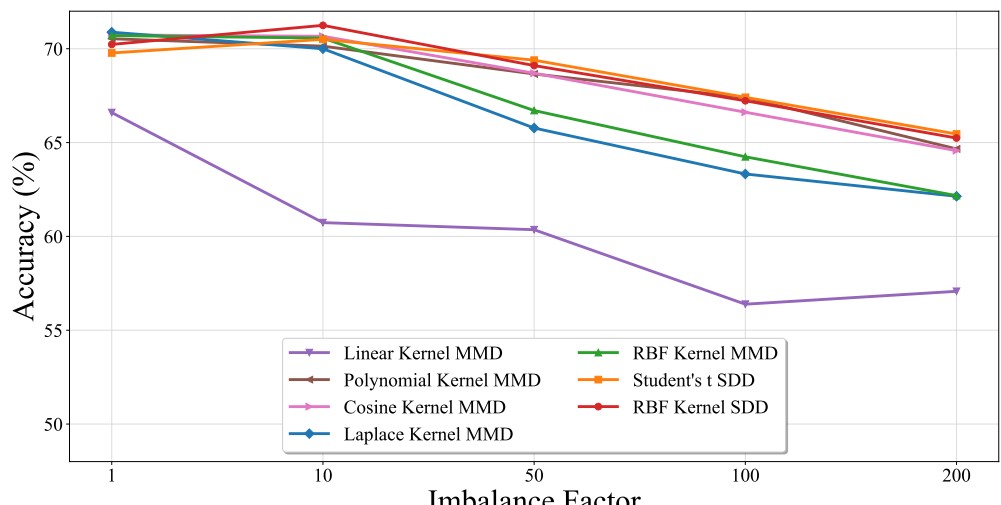

Figure 5: **Performance of Different Kernel Methods on CIFAR-10LT with Varying Imbalance Factors.**

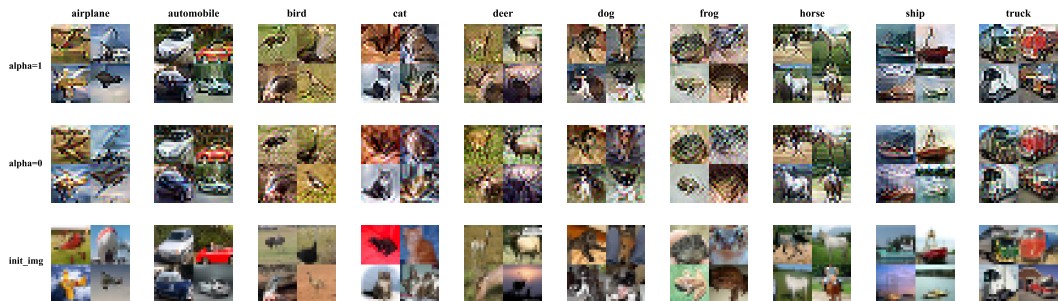

Figure 6: **Effect of the $\alpha$ parameter: Images synthesized by CSDM on CIFAR-10 (IPC=10) for values of 0 and 1.** As shown in the images, amplitude-only distillation ($\alpha = 1$) yields diverse but less realistic images, whereas phase-only distillation ($\alpha = 0$) produces realistic images but suffers from limited diversity.

