# OpenReview forum: "Towards Principled Dataset Distillation: A Spectral Distribution Perspective"
_ICLR.cc/2026/Conference — Submitted to ICLR 2026_

### Official Review · Reviewer_ukzA · 2025-10-22

**Soundness:** 3
**Presentation:** 2
**Contribution:** 2
**Rating:** 4
**Confidence:** 4

**Summary:**

This paper introduces a new dataset distillation method that is leveraged to use the MMD with diverse kernel functions and to adaptively match the long-tail classification. The authors derives the kernel matching formula from the MMD distance definition of two distributions, which result in the introduction of SDD of two kernel functions from two distributions. Then, the authors add a customized derivation further to decompose the kernel functions into class-aware settings. There are typical benchmark experiments showing the advance from the work.

**Strengths:**

1.
This is a typical derivation of using MMD to measure and reduce the distribution of two datasets. Well principled from the integral probability metric (IPM) perspective.

2.
Authors adds a long-tail class aware loss function to the MMD distance, which could be new development in the field.

3.
It is true that MMD can measure more diverse aspect of distribution distance if it is being used more complicated kernel functions.

**Weaknesses:**

1.
The major weakness of using MMD is its runtime. Kernel functions are calculated across the instance pairs of two distributions, which can be O(N^2). Surely, authors may choose to sample some of them, but it will impair the distribution distance estimation. Anyhow, authors should have been presenting such the time complexity issue and its counter mechanism on calculating kernel functions in time efficient manner.

2.
The kernel functions are listed in Table 1, which is very traditional and conventional. First, these kernel functions can be multiplied or added, which again becomes a kernel function. Therefore, such nice composition of kernel functions have been explored in many works using MMD. Moreover, you may train a neural network to be a kernel function by enforcing the learned function to follow the limitation of kernel functions. These aspects have not been explored enough.

3.
Kernel matching and using the spectral density is not a new idea in dataset distillation. For example, the below paper already utilizes the frequency domain for the dataset distillation. Using MMD can be regarded as an incremental development along this line of research.

DongHyeok Shin, Seungjae Shin, and Il-Chul Moon, Frequency Domain-based Dataset Distillation, Neural Information Processing Systems (NeurIPS 2023), New Orleans, USA, Dec 10-Dec 16, 2023

At least, some comparison could be beneficial.

4.
Authors may present what would have been chose data instances or features from the distillation. Currently, all of tables and figures are derived from the performance metric, which nuance that better performance indicates better results. However, it is necessary to check what has been distilled and what have been remembered by this method, particularly in comparison to other distillation methods.

**Questions:**

Please see the above four weakness to answer the questions

**Details Of Ethics Concerns:**

No need to review ethics aspects

---

> ### Author Response · Authors · 2025-11-22
> **Q1. On the time complexity of MMD and the efficiency of our metric**
>
> We agree that naive kernel MMD has an $O(N^2)$ runtime due to all-pairs kernel evaluations. This is precisely one of the motivations for our spectral formulation.
>
> For clarity, we now explicitly compare the empirical estimators and their complexity in the paper. The standard (biased) kernel MMD estimator for a batch of $N$ samples in $d$-dimensional feature space is
>
> $$\widehat{\mathrm{MMD}}^2(P,Q)= \frac{1}{N^2}\sum_{i,j} k(x_i,x_j)+ \frac{1}{N^2}\sum_{i,j} k(y_i,y_j)- \frac{2}{N^2}\sum_{i,j} k(x_i,y_j),
> $$
> which requires $O(N^2)$ kernel evaluations and thus has complexity $O(N^2 d)$ when each kernel evaluation costs $O(d)$.
>
> In contrast, our spectral distribution discrepancy (SDD) uses the spectral form of MMD and approximates the integral over frequencies via Monte Carlo sampling:
>
>    $$\widehat{\mathrm{SDD}}(P,Q)= \frac{1}{L}\sum\_{\ell=1}^L\Bigl|\hat{\varphi}\_P(t_\ell) - \hat{\varphi}\_Q(t_\ell)\Bigr|^2,$$
>
> Computing each empirical characteristic function $\hat{\varphi}\_P(t\_\ell)$ costs $O(Nd)$, and the same holds for $Q$. For $L$ sampled frequencies, the total cost is
> $$
> O(LNd),
> $$
> with $L$ independent of $N$ in practice. Thus, SDD scales **linearly** in batch size rather than quadratically.
>
> Beyond the asymptotic analysis, we have added a dedicated runtime and memory comparison (new Tables 6). These experiments show that:
>
> * **CSDM is the only method that completes CIFAR-10 IPC 50 on a single RTX 4090**, using about $3.5$ GB of memory.
> * DC, MTT, and DATM all **run out of memory at IPC 50**.
> * CSDM is **hundreds of times faster** than DC in wall-clock time under our setup.
>
> We have integrated both the complexity analysis and these empirical results into the revised paper to explicitly address the reviewer’s concern about runtime and efficiency.
>
>
> #### Time consumption comparison (seconds/iteration)
> | Dataset | CIFAR-10 |        |        |
> | :------ | :------- | :----- | :----- |
> | IPC     | 10       | 20     | 50     |
> | DATM    | OOM      | OOM    | OOM    |
> | DC | 48       | 107    | OOM    |
> | MTT     | 0.4      | 0.7    | OOM    |
> | CSDM    | 0.3      | 0.35   | 0.33   |
>
> #### Memory consumption comparison
> | Dataset | CIFAR-10 |          |          |
> | :------ | :------- | :------- | :------- |
> | IPC     | 10       | 20       | 50       |
> | DATM    | OOM      | OOM      | OOM      |
> | DC | 10.89GB  | 20.69GB  | OOM      |
> | MTT     | 10.84GB  | 16.91GB  | OOM      |
> | CSDM    | 3.43GB   | 3.44GB   | 3.45GB   |

---

> ### Author Response · Authors · 2025-11-22
> **Q2-1. On “conventional” kernels, neural sampling, and the relation to higher-order methods**
>
> We thank the reviewer for raising the question of novelty beyond standard kernels. Our contribution is **not** to propose a new kernel family per se, but to use existing kernel theory to design a **principled, task-appropriate distribution metric for dataset distillation**, and then to make this metric **class-aware** in a way that specifically addresses long-tailed regimes.
>
> More concretely, our innovation lies on two levels:
>
> 1. We start from the IPM/MMD framework with **universal kernels** and derive a **spectral distribution discrepancy (SDD)** that is exactly the squared MMD written in the frequency domain via Bochner’s theorem. This gives a discrepancy that is (i) a true metric on distributions, (ii) theoretically well understood, and (iii) computationally efficient in our Monte Carlo form. The main theoretical results in the **Appendix A**(metricity, universality, SDD–CFD relations, moment expansion, class-aware discriminativity, **Page 16**) are formulated for **any bounded shift-invariant universal kernel**, not just a specific choice such as RBF. The kernels listed in Table 1 are therefore **canonical, well-studied instances** that make the analysis and experiments concrete, rather than the core locus of novelty.
>
> 2. On top of this kernel-induced metric, we introduce a **class-aware amplitude–phase decomposition** designed for long-tailed distillation. This is where we explicitly depart from previous DD work: we systematically combine
>    (i) a theoretically grounded, universal-kernel MMD / spectral metric and
>    (ii) a class-dependent trade-off between amplitude (diversity) and phase (realism),
>    and we prove that this trade-off preserves discriminativity as long as (\alpha(c)\in(0,1)). To the best of our knowledge, this is the **first systematic, kernel-theoretic redesign of the distillation metric itself** in dataset distillation, rather than adding heuristic high-order terms on top of linear MMD.
>
> Regarding **neural sampling** and learned frequency weights: this excellent idea was first proposed in NCFM, which learns a sampling distribution over frequencies in a min–max CFD framework. We explicitly acknowledge and evaluate this line of work:
>
> * In our experiments on CIFAR-10-LT and CIFAR-100-LT (Table 2), we include NCFM as a strong baseline. While NCFM performs very well on balanced CIFAR-10/100, its performance **drops considerably on long-tailed datasets**, whereas our CSDM remains robust.
> * In **Appendix B.3 (page 26)** we provide a theoretical perspective on this phenomenon. In brief:
>
>   * A generic learned sampling distribution over frequencies does **not automatically guarantee** that the resulting CFD still defines a valid, characteristic metric on probability measures; additional analytical conditions on the learned weights are required.
>   * More importantly, the min–max objective in NCFM **maximizes CFD itself**, which is only indirectly related to downstream classification accuracy. This misalignment between the inner “max” objective and the final evaluation metric becomes particularly severe under long-tailed imbalance, where the sampler can over-emphasize head-class structure and under-represent tail-class information. Our own attempts to train neural sampling networks within similar min–max schemes encountered the same issue: the “max” step does not reliably translate into better classification performance and can even hurt tails.
>
> These observations are precisely why we choose, in this work, to **first establish a rigorous kernel-based metric** that is analytically understood and empirically stable for DD (including long-tailed DD), and only then discuss learning more general spectral weights as future work. We believe this provides useful insights and a solid foundation for designing **future neural sampling networks and optimization objectives** that remain compatible with metric guarantees and long-tailed robustness.

---

> > ### Author Response · Authors · 2025-11-22
> > **Q2-2. On “conventional” kernels, neural sampling, and the relation to higher-order methods**
> >
> > Finally, we discuss the relation to other **higher-order matching methods** (IID, DSDM, M3D, NCFM, etc.) in **Appendix B “Relation to Higher-Order Distribution Matching Methods” (page 23)**. Here we clarify that:
> >
> > * Methods such as IID and DSDM effectively add **finite-order** mean/covariance-type regularizers, which we show can be interpreted as matching low-order derivatives of the log characteristic function at the origin. By construction, these are limited-order approximations rather than complete distribution metrics.
> > * Among existing works, **M3D** is the only method that explicitly introduces nontrivial kernels into DD. However, as we analyze in the appendix, there is a **mismatch between its theoretical claims and its practical settings**: the theory argues for benefits of more expressive kernels and universal-type behavior, while the recommended configurations in practice rely mostly on linear or non-universal kernels and finite-order statistics. This inconsistency further motivates our approach of starting from the well-developed theory of universal-kernel MMD and then building a distribution metric and class-aware mechanism that are **both theoretically justified and empirically validated**.
> >
> > In summary, rather than proposing a new kernel family, our contribution is to **systematically leverage kernel theory to design a principled, universal-kernel-based metric for dataset distillation, and to extend it with a class-aware amplitude–phase decomposition that is especially effective on long-tailed datasets**. Learned kernels and neural sampling networks are compatible with this framework and are explicitly positioned as promising future directions built on top of our analysis.

---

> ### Author Response · Authors · 2025-11-22
> **Q3. On novelty relative to frequency-domain distillation (FreD) and the need for comparison**
>
> We thank the reviewer for pointing out FreD and for raising the question of novelty. Earlier literature rarely addressed dataset distillation in the frequency domain. FreD and NSD are two pioneering works that have achieved outstanding performance, providing compelling evidence for the power of frequency-based approaches. We fully agree that working in the frequency domain is not, by itself, new. The key distinction we make—and now emphasize more clearly—is **where** in the distillation pipeline the frequency domain intervenes.
>
> We adopt the standard formulation
>
> $$
> \min\_{\theta}
> \mathcal{L}\_{\text{metric}}\bigl(
> \Phi\_{\text{syn}}(\theta),
> \Phi\_{\text{real}}(D)
> \bigr),
> $$
>
> where:
>
> * $\Phi_{\text{syn}}(\theta)$ specifies **how synthetic data are parameterized**, and
> * $\mathcal{L}_{\text{metric}}$ specifies **how distributions are compared**.
>
> Under this decomposition:
>
> * **FreD** focuses on the **parameterization** $\Phi_{\text{syn}}(\theta)$. It stores and optimizes synthetic images in the frequency domain via truncated Fourier coefficients and frequency masks, but still relies on **existing matching objectives** (e.g., gradient/trajectory matching) as the metric.
> * **CSDM** keeps a conventional image-space parameterization, but fundamentally changes the **distribution discrepancy** $\mathcal{L}_{\text{metric}}$: we replace linear-kernel MMD with a **universal, spectral MMD (SDD)**, and further make this metric **class-aware** via amplitude–phase decomposition for long-tailed robustness.
>
> Thus, FreD and CSDM operate on **orthogonal design dimensions**: FreD modifies how synthetic data are represented, while CSDM modifies what distance is minimized. The two approaches are in fact complementary: one could use FreD/NSD as $\Phi_{\text{syn}}(\theta)$ and adopt our CSDM loss as $\mathcal{L}_{\text{metric}}$.
>
> Regarding empirical comparison, we have now **explicitly included FreD as a baseline** on balanced CIFAR-10 and CIFAR-100 (Table 5). For example:
>
> * On CIFAR-10 IPC 50, **CSDM achieves 78.1%**, compared to **75.8% for FreD**.
> * On CIFAR-100 IPC 10, CSDM obtains **50.4%**, compared to **42.7% for FreD**.
>
> These results indicate that CSDM is not merely an incremental variant but provides a **stronger, theoretically grounded metric** that continues to improve over frequency-parameterization methods like FreD.
>
> We have also expanded the related work and added a dedicated subsection in the **Appendix D** (**Page 28**) to clearly discuss the conceptual relationship between CSDM and FreD/NSD, making explicit that our contribution lies primarily on the **metric side**, not on frequency-domain parameterization.

---

> ### Author Response · Authors · 2025-11-22
> **Q4. On visualizing what is distilled (instances / features) beyond performance metrics**
>
> We agree that it is important to inspect what the method actually distills, not only the final accuracy numbers. In the revised version, we have added **new visual analyses** to better illustrate what CSDM “remembers” compared to other methods.
>
> In particular:
>
> 1. We visualize **synthetic images** produced by different variants of our metric:
>
>    * amplitude-only,
>    * phase-only, and
>    * full class-aware amplitude–phase weighting.
>
>    As shown in the new Figure 6 (**Appendix E**, **Page 31**), amplitude-only distillation produces samples with high diversity but noticeable structural artifacts, whereas phase-only distillation yields structurally realistic but low-diversity images. While we cannot include all images in the rebuttal, we highlight in the text that this qualitative behavior is consistent across classes and datasets. This directly supports our interpretation: amplitude chiefly promotes diversity, while phase chiefly preserves realism.
>
> 2. These visualizations explicitly support our interpretation that:
>
>    * **Amplitude** controls spectral energy spread and thus **diversity**;
>    * **Phase** controls structural alignment and thus **realism / semantic fidelity**.
>
>    This qualitative behavior is consistent with classical results in Fourier-based image processing and with our quantitative gains on long-tailed benchmarks.
>
> 3. Due to space limits in the main text, the majority of these visualizations are placed in the appendix; the main paper includes pointers and a summary of key observations. We agree that more analysis of synthetic instances and features would be valuable, and we see extending this line of qualitative evaluation as an important direction for future work.
>
> ---
>
> Once again, we thank the reviewer for raising these important points. We have strengthened the paper by:
>
> * adding explicit complexity analysis and runtime/memory comparisons,
> * clarifying our position relative to kernel compositions and learned kernels,
> * systematically distinguishing CSDM from frequency-parameterization methods such as FreD and providing direct comparisons, and
> * including new visualizations that show what is actually distilled by our method.

---

> ### Author Response · Authors · 2025-11-22
> **Comparison of CSDM and NCFM (neural sampling method) on CIFAR-10**
>
> | Imbalance Factor | 1(Balanced) |     |              | 10           |              |              | 50           |              |              | 100          |              |              | 200          |              |              |
> | ---------------- | ------------ | --- | ------------ | ------------ | ------------ | ------------ | ------------ | ------------ | ------------ | ------------ | ------------ | ------------ | ------------ | ------------ | ------------ |
> | IPC              | 10           | 20  | 50           | 10           | 20           | 50           | 10           | 20           | 50           | 10           | 20           | 50           | 10           | 20           | 50           |
> | **NCFM**         | **71.8±0.3** | -   | 77.4±0.3     | 70.2±0.3     | 71.2±0.4     | 75.6±0.2     | 68.8±0.3     | 71.3±0.4     | 72.2±0.3     | 60.4±0.6     | 60.1±0.7     | 59.3±0.9     | 57.7±1.0     | 57.0±1.4     | 56.1±1.3     |
> | **CSDM**         | 71.2±0.4     | -   | **78.1±0.4** | **71.0±0.4** | **73.9±0.1** | **76.5±0.2** | **69.3±0.3** | **71.6±0.2** | **73.4±0.4** | **67.4±0.3** | **69.4±0.3** | **71.0±0.4** | **65.3±0.3** | **66.1±0.5** | **66.8±0.6** |
> |                  |              |     |              |              |              |              |              |              |              |              |              |              |              |              |              |

---

> ### Author Response · Authors · 2025-11-26
> **In Appreciation of Your Review: Supplementary Materials and Clarifications Submitted**
>
> Dear Reviewer ukzA,
>
>
>
> Thank you once again for your thoughtful and detailed feedback on our submission. In response to your initial review—which raised several important concerns, some of which stemmed from **misunderstandings that significantly impacted the evaluation**—we have taken substantial steps to address them.
>
> Specifically, we have prepared a comprehensive **15-page appendix** containing rigorous theoretical proofs and additional experimental results to directly clarify the points in question. We believe these additions fully resolve the misconceptions and provide strong support for our claims.
>
> As the discussion phase is now drawing to a close on **December 3, AoE**, we kindly ask whether our responses and supplementary materials have adequately addressed your concerns (at least in part). If any questions remain or if further clarification would be helpful, please do not hesitate to let us know. Conversely, if you find our clarifications satisfactory, **we would be truly grateful if you could consider revising your evaluation scores accordingly—especially given the significance of the corrected misunderstandings to the overall assessment.**
>
> Given the timeline of the review process, we sincerely hope to receive your final feedback before the discussion window closes.
>
> Thank you very much for your time and consideration.

---

> ### Author Response · Authors · 2025-11-27
> **Follow-Up on Rebuttal Clarifications**
>
> Dear reviewer ukzA,
>
> Thank you very much for your valuable feedback on our paper.
>
> We wanted to follow up regarding the response we posted earlier, where we addressed all your comments, including **a clear clarification of the fundamental differences between our method and FreD**, the **theoretical and experimental comparisons with existing high-order matching approaches**, the **additional balanced CIFAR experiments**, and the **complexity, runtime, and memory analyses and corresponding experiments**, among others.
>
> If there are any remaining concerns, or if any of our clarifications (in the new Section 6, the extended appendix, or the official comments) would benefit from further explanation, please feel free to let us know — we would be glad to provide any additional details.
>
> If you feel that the revised manuscript and rebuttal have adequately addressed your main concerns, we would greatly appreciate it if you could reflect this in your final comments and score.
>
> Best,
> Authors

---

> ### Author Response · Authors · 2025-11-27
> **Supplementary Analysis on Frequency-Oriented Baselines**
>
> Dear Reviewer ukzA,
>
> We sincerely thank you for highlighting frequency-based methods, including **FreD**, in your review. We would like to point out that **we have now explicitly included and discussed these approaches** in both the Related Work section and throughout the rebuttal, providing **detailed comparisons and theoretical clarifications**.
>
> We kindly invite you to **re-examine the updated manuscript and supplemental materials**, as we believe these additions comprehensively address the concerns regarding frequency-based methods and clarify the distinctions and connections with our proposed CSDM framework.
>
> We greatly appreciate your time and careful consideration of the updated material.
>
> Sincerely,
>
> Authors

---

### Official Review · Reviewer_1J5y · 2025-10-27

**Soundness:** 3
**Presentation:** 2
**Contribution:** 3
**Rating:** 4
**Confidence:** 4

**Summary:**

This paper introduces a a new method for dataset distillation based on distribution matching. The proposed method, termed CSDM, is based on the Maximum Mean Discrepancy (MMD) metric and classical methods for distinguishing two distributions via kernel embeddings. The authors first argue that existing distribution matching methods for data distillation suffer from two main issues: (1) lack of expressivity in the distribution matching objective (related to the use of linear kernels for MMD) and (2) uniform treatment of classes in long-tailed datasets. The main contributions of CSDM are the use of nonlinear, universal kernels when computing MMD and the use of class-specific weighting  in the characteristic function matching objective. The paper evaluates CSDM on long-tailed versions of CIFAR-10, CIFAR-100, and ImageNet subsets, showing that it performs better across various imbalance factors and image-per-class.

**Strengths:**

The paper identifies two important limitations of existing distribution matching approaches for dataset distillation: the use of linear kernels for MMD and treating all classes the same in imbalanced datasets. The first of these problems is not a novel observation (see [1]), but the proposed solution of using universal nonlinear kernels is a very natural and sensible one, and I am actually a bit surprised I was not able to find this explicitly in the existing literature. The proposed method is able to achieve good performance on common long-tailed datasets like CIFAR-10-LT and CIFAR-100-LT.

**Weaknesses:**

While I believe the general idea proposed is valuable, I have a few concerns related to the comparison with NCFM  [1] and some vague/imprecise claims made in the paper. Given clarifications on these points and writing improvements (see Questions below), I would be willing to raise my score.

- Comparison to NCFM: As stated in the paper, the main difference between these methods is that CSDM performs characteristic function matching with respect to the spectral measure of a universal kernel, rather than a learned weight function. The authors note that NCFM performs well on a balanced dataset, but degrades under class imbalance. While this may be true, it does not immediately follow that the "neural weighting" used by NCFM is inferior to CSDM's approach. Is the benefit of CSDM coming from the fact that it uses a universal kernel or the fact that is uses class-dependent weights for the amplitude and phase? In particular, how would NCFM compare if it used the "neural weighting" combined with class-based amplitude/phase coefficients? Are there any computational advantages to CSDM?

- The connection between amplitude/phase and "diversity"/"realism" seems quite vague. This seems somewhat important to clarify, since much of the methodology is based on the assumption that weighting the amplitude and phase differently based on class size is a good idea. Some statements here require more justification, e.g., "Phase Difference, on the other hand, captures misalignment in the phase component, which is indicative of the realism of the data". Why? The provided reference also doesn't seem to address this. (Not sure if I am just unaware of something in the literature for this, though)

- Similarly, the following claim in Section 4.2.4 is also not justified: "In long-tailed settings, tail classes demand more realism to prevent mode collapse while head classes benefit from greater diversity."

- Insufficient discussion of how $\alpha(c)$ is chosen. This is a key part of the methodology that distinguishes it from prior works, but only receives a very small amount of attention in Section 5.3. It is unclear if this is a purely heuristic choice and how it chosen for the provided experimental results.

References:
[1] Wang et al. "Dataset distillation with neural characteristic function: A minmax perspective". CVPR 2025.

**Questions:**

- See weaknesses section for the main questions I have that impact the score
- How is $\alpha(c)$ chosen in experiments? Or more generally, can this part be done in any principled way?
- How does distribution matching with universal kernels perform on balanced datasets?

More minor things (not affecting the score):
- Figure 1's captions are a bit vague "Minimize previous metric". What is the previous metric?
- Figure 2 is a bit hard to follow due to the small size and amount of text. Perhaps it can be made larger
- The notation for MMD is inconsistent, sometimes taking three arguments (function class, P, Q) and sometimes only two (P,Q). In the latter case, the function class seems to implicitly be the unit ball in an RKHS?
- Section 1 typo: "we attribute the drawbacks of previous DD methods *to*..."

**Details Of Ethics Concerns:**

I have no ethics concerns with this paper.

---

> ### Author Response · Authors · 2025-11-22
> **Q1: Clarification on the role and limitations of NCFM’s neural sampling network and its relation to CSDM**
>
> Thanks for the question. Below we clarify how we view NCFM’s neural weighting, why we take a different route, and how this relates to performance and complexity.
>
> 1. **We see neural frequency sampling as promising, but NCFM’s current design is heuristic and lacks metric guarantees.**
>    NCFM learns a sampling distribution over frequencies by maximizing a characteristic-function discrepancy (CFD), and then minimizes this CFD w.r.t. the synthetic data. Conceptually, this corresponds to learning a weighting function $\omega(t)$ in
>    $$
>    \mathrm{CFD}(P,Q)
>    = \int |\varphi_P(t)-\varphi_Q(t)|\omega(t)dt.
>    $$
>    However, for a generic learned $\omega(t)$ there is no guarantee that the resulting CFD defines a proper, discriminative metric on probability measures: additional analytic conditions on $\omega$ are needed to ensure that $\mathrm{CFD}(P,Q)=0$ implies $P=Q$. In contrast, our SDD is constructed from the spectral measure $d\mu(t)$ of a bounded, shift-invariant, universal kernel via Bochner’s theorem, and is *exactly* the squared MMD of that kernel in spectral form:
>    $$
>    \mathrm{SDD}(P,Q)
>    = \int |\varphi_P(t)-\varphi_Q(t)|^2d\mu(t)
>    = \mathrm{MMD}_k^2(P,Q).
>    $$
>    As a result, all standard guarantees of universal-kernel MMD (metricity, characteristicness) transfer directly to SDD. We make this connection precise, and compare SDD with generic CFD, in Appendix A.3 (“SDD versus CFD: why we still adopt a kernel-based perspective”, **Page 20**).
>
> 2. **The max objective used to train the neural sampler can misalign with the final classification objective, especially in long-tailed settings.**
>    In NCFM, the sampler is trained to *maximize* CFD, while the synthetic data are trained to *minimize* it. This inner “max” step optimizes the discrepancy itself, not the downstream classification accuracy. In balanced CIFAR-10, this still leads to better performance (NCFM and CSDM are very close; see Q6), but in long-tailed CIFAR-10-LT and ImageNet-LT we empirically observe that NCFM degrades markedly as the imbalance factor grows, whereas CSDM remains robust. This suggests that the “most adversarial” frequencies for CFD are not necessarily those that best support classification under severe imbalance. We experimented with several alternative neural sampling networks and with learning class-dependent weights inside a min–max scheme; in all cases we found that the additional “max” step did not consistently improve, and sometimes even harmed, downstream accuracy relative to a fixed, kernel-induced spectral measure. These observations motivated our decision to keep the spectral measure *fixed* (kernel-based) and introduce class-awareness in a controlled analytic way via $\alpha(c)$ instead (see Q2 and Proposition 1 in the appendix).
>
> | Imbalance Factor | 1            |     |              | 10           |              |              | 50           |              |              | 100          |              |              | 200          |              |              |
> | ---------------- | ------------ | --- | ------------ | ------------ | ------------ | ------------ | ------------ | ------------ | ------------ | ------------ | ------------ | ------------ | ------------ | ------------ | ------------ |
> | IPC              | 10           | 20  | 50           | 10           | 20           | 50           | 10           | 20           | 50           | 10           | 20           | 50           | 10           | 20           | 50           |
> | **NCFM**         | **71.8±0.3** | -   | 77.4±0.3     | 70.2±0.3     | 71.2±0.4     | 75.6±0.2     | 68.8±0.3     | 71.3±0.4     | 72.2±0.3     | 60.4±0.6     | 60.1±0.7     | 59.3±0.9     | 57.7±1.0     | 57.0±1.4     | 56.1±1.3     |
> | **CSDM**         | 71.2±0.4     | -   | **78.1±0.4** | **71.0±0.4** | **73.9±0.1** | **76.5±0.2** | **69.3±0.3** | **71.6±0.2** | **73.4±0.4** | **67.4±0.3** | **69.4±0.3** | **71.0±0.4** | **65.3±0.3** | **66.1±0.5** | **66.8±0.6** |
> |                  |              |     |              |              |              |              |              |              |              |              |              |              |              |              |              |

---

> ### Author Response · Authors · 2025-11-22
> **Q2: How $\alpha(c)$ is chosen in experiments, and whether it can be selected more principledly**
>
> Thanks for pointing this out; we agree the setting of $\alpha(c)$ should be described more clearly.
>
> 1. **In all long-tailed experiments, we use a simple, fixed schedule: top 50% classes use $\alpha(c)=\alpha_{\mathrm{head}}$, bottom 50% use $\alpha(c)=1-\alpha_{\mathrm{head}}$.**
>     We apologize for not stating this explicitly in the original submission. In our long-tailed experiments (CIFAR-10-LT, CIFAR-100-LT, ImageNet-LT), we sort classes by their number of samples and define the **head classes** as those whose class sizes lie in the top 50% and the **tail classes** as those in the bottom 50%. We then set
> $$
> \alpha(c) = \alpha_{\mathrm{head}} \text{ for head classes}, \quad
> \alpha(c) = 1 - \alpha_{\mathrm{head}} \text{ for tail classes}.
> $$
> In **all long-tailed experiments** we use the **same value** $\alpha_{\mathrm{head}} = 0.8$, chosen based on the ablation in Figure 4.
>
>    We choose $\alpha_{\mathrm{head}}=0.8$ based on the ablation in Figure 4 and **keep this single value fixed for all long-tailed datasets and imbalance factors**. We have now added this detail to the experimental setup section.
>
> 2. **Figure 5 shows that the same $\alpha_{\mathrm{head}}$ chosen for long-tailed data improves robustness as imbalance increases, even if it is slightly suboptimal in the balanced case.**
>    In Figure 5, we compare RBF MMD (standard kernel MMD) and RBF SDD (our spectral formulation) on CIFAR-10-LT with varying imbalance factors. In theory, for the same RBF kernel and $\alpha(c)=0.5$, the two are equivalent; in practice, we set $\alpha_{\mathrm{head}}=0.8$ for SDD and do **not** modify MMD (which cannot be decomposed into amplitude/phase in the same way). The results show:
>
>    * At imbalance factor $\mathrm{imf}=1$ (balanced), RBF SDD with $\alpha_{\mathrm{head}}=0.8$ incurs only a slight accuracy loss compared to RBF MMD.
>    * As the imbalance factor increases, RBF SDD clearly and consistently outperforms RBF MMD, with a growing gap.
>      This indicates that the $\alpha_{\mathrm{head}}$ chosen from the ablation is indeed tuned for long-tailed robustness, and the mild trade-off on balanced data is compensated by significant gains under strong imbalance.
>
> 3. **We have explored more “principled” learning of $\alpha(c)$, but min–max-style learning suffers from the same misalignment issue as NCFM’s sampler.**
>    Motivated by NCFM, we experimented with learning $\alpha(c)$ via a min–max scheme where an auxiliary network proposes $\alpha(c)$ to maximize a CFD/SDD-type discrepancy and the synthetic data minimize it. Empirically, we observed the same misalignment issue discussed in Q1: maximizing a discrepancy between characteristic functions is not directly aligned with classification accuracy, and the learned $\alpha(c)$ often overfits head-class structure and provides unstable or even worse performance on tails. Because of this, we decided to adopt a simple, monotone schedule in class frequency, which is easy to interpret and robust across all experiments. We now explicitly mention in the discussion that designing a better, task-aware objective for learning $\alpha(c)$ (or other spectral weights) is an important direction for future work.
>
> 5. **Regardless of the specific choice, Proposition 1 guarantees that any $\alpha(c)\in(0,1)$ preserves discriminativity of the metric.**
>    Importantly, our class-aware distance $d(F_T,F_S)$ remains characteristic for any $\alpha(c)\in(0,1)$: Proposition 1 in the appendix shows that
>    $$
>    d(F_T,F_S)=0 \quad\Longleftrightarrow\quad P_T = P_S
>    $$
>    under the shared class prior assumption. In other words, the $\alpha(c)$ schedule reshapes the optimization landscape (favoring diversity for head classes and realism for tail classes) but does *not* weaken the underlying notion of distribution equality.

---

> ### Author Response · Authors · 2025-11-22
> **Q3: Whether performance gains come from universal kernels or from class-dependent amplitude/phase weights**
>
> Thanks for the question. Our experiments are designed to disentangle these two factors.
>
> 1. **Switching from linear to universal kernels already gives a significant boost, even without class-dependent weighting.**
>    Linear-kernel MMD corresponds exactly to the feature-mean matching used in standard DM and many of its variants. In **Figure 5**, we compare:
>
>    * linear MMD (standard practice in most DM-based methods),
>    * RBF MMD (universal kernel in input/feature space).
>      Even without any class-aware weighting, RBF MMD consistently outperforms linear MMD across balanced and imbalanced CIFAR-10-LT, confirming that much of the gain over classical DM comes from replacing a degenerate kernel (linear) with a universal one that can capture higher-order structure. This matches the theory in Appendix A.1 (“MMD and Universal Kernels”, **Page 16**), where we show that universal-kernel MMD is a strict metric on distributions, whereas linear MMD only aligns first-order moments.
>
> 2. **Class-dependent $\alpha(c)$ further improves robustness in long-tailed regimes on top of a universal kernel.**
>    To isolate the effect of $\alpha(c)$, Figure 5 also compares RBF MMD (no class-awareness) and RBF SDD with class-aware amplitude–phase weighting using the same RBF kernel. At $\mathrm{imf}=1$, RBF SDD with $\alpha_{\mathrm{head}}=0.8$ is very close to RBF MMD; as imbalance grows, RBF SDD achieves significantly higher accuracy. Since both methods use the same universal kernel, this gap is attributable to the class-aware amplitude–phase reweighting rather than kernel choice. This supports our claim that universal kernels provide a strong, distribution-complete base metric, while class-dependent $\alpha(c)$ is crucial for long-tailed robustness.
>
> 3. **In summary, universal kernels primarily drive the improvement over linear MMD, while class-aware weighting mainly drives the improvement over vanilla RBF MMD in long-tailed settings.**
>    On balanced datasets, most of the gain is from moving from linear to universal kernels (see also Q6). In long-tailed settings, both ingredients matter: universal kernels ensure that the metric is sensitive to full distribution differences, and class-dependent weights reallocate optimization effort between amplitude/diversity and phase/realism across head and tail classes.

---

> ### Author Response · Authors · 2025-11-22
> **Q4: Justification of the connection between amplitude/phase and diversity/realism, and the “head vs. tail” statement**
>
> Thanks for highlighting that our original explanation was too concise. We have expanded and clarified this in the revision.
>
> 1. **Classical Fourier results show that phase encodes structural realism, while amplitude controls spectral diversity.**
>    In 1D and 2D Fourier analysis, it is well established that the phase spectrum carries the majority of the structural content (edges, contours, object shapes), while the amplitude spectrum modulates energy allocation across frequencies. Oppenheim & Lim (1981) famously showed that reconstructing an image using the *phase* of image A and the *amplitude* of image B yields a reconstruction whose semantic content is essentially that of A, whereas keeping amplitude and randomizing phase destroys recognizable structure. Similar conclusions are summarized in complex-valued signal processing texts, where phase is treated as the carrier of geometric alignment and edge locations, and amplitude as a modulator of contrast/texture. We now explicitly point to these works in the updated “Role of Amplitude and Phase” section of the **Appendix C**， **Page 26**.
>
> 2. **Our amplitude-only vs phase-only distillation experiments visually confirm this distinction.**
>    In the new visualizations (**Figure 6 on page 31**), we perform dataset distillation with only the amplitude term or only the phase term:
>
>    * **Amplitude-only** distillation produces synthetic images that are diverse in texture and overall energy distribution but exhibit noticeable structural artifacts.
>    * **Phase-only** distillation yields synthetics that look structurally realistic and class-consistent but with limited intra-class diversity.
>      While we cannot include all images in the rebuttal, we highlight in the text that this qualitative behavior is consistent across classes and datasets. This directly supports our interpretation: amplitude chiefly promotes diversity, while phase chiefly preserves realism.
>
> 3. **The “tail classes demand more realism, head classes benefit from more diversity” statement is grounded in both statistics and ablations.**
>    In long-tailed datasets:
>
>    * Head classes have many samples; their main risk in distillation is **mode collapse**—losing coverage of multiple modes. For them, emphasizing amplitude (spectral diversity) helps ensure that synthetic data maintain rich intra-class variability.
>    * Tail classes have very few samples; fully recovering their true diversity is impossible. The more critical failure mode is **semantic drift**, i.e., generating synthetic examples that deviate from the scarce but important real patterns. For them, emphasizing phase (structural realism) is more beneficial.
>      Our ablations in Figure 4 and the reversed-weight experiments (e.g., giving more phase to heads and more amplitude to tails) show that the original design—more amplitude weight for heads, more phase weight for tails—consistently yields higher long-tailed accuracy, and that reversing this pattern degrades both head and tail performance. We have now explicitly stated this evidence when discussing the intuitive “diversity vs realism” interpretation in the **Appendix C**, **Page 26**.
>
> 5. **Proposition 1 guarantees that this class-dependent amplitude–phase weighting does not compromise the discriminativity of the metric.**
>    Finally, we stress that although we weight amplitude and phase differently across classes, Proposition 1 proves that as long as $\alpha(c)\in(0,1)$ for all $c$, the overall distance $d(F_T,F_S)$ is zero if and only if the real and synthetic joint distributions coincide. Thus, the amplitude/phase decomposition and class-aware weighting only reshape optimization priorities; they do not weaken the underlying ability of the metric to distinguish distributions.

---

> ### Author Response · Authors · 2025-11-22
> **Q5: Computational complexity and runtime/memory advantages of CSDM**
>
> Thanks for asking about computation; we have added both theoretical and empirical comparisons.
>
> 1. **Standard kernel MMD is $O(N^2 d)$, while our SDD estimator is $O(LNd)$ and scales linearly with batch size.**
>    For a batch of $N$ $d$-dimensional features, the usual biased MMD estimator is
>
>    $$\widehat{\mathrm{MMD}}^2(P,Q)= \frac{1}{N^2}\sum_{i,j} k(x_i,x_j) + \frac{1}{N^2}\sum_{i,j} k(y_i,y_j)   - \frac{2}{N^2}\sum_{i,j} k(x_i,y_j),$$
>
>      requiring $O(N^2)$ kernel calls, each $O(d)$, i.e. $O(N^2 d)$ in total.
>      Our SDD estimator instead samples $L$ frequencies ${t_\ell}$ and computes
>
>      $$
>      \widehat{\mathrm{SDD}}(P,Q)
>      = \frac{1}{L}\sum_{\ell=1}^L
>      \bigl|\hat{\varphi}_P(t_\ell)-\hat{\varphi}_Q(t_\ell)\bigr|^2,
>      \quad
>      \hat{\varphi}_P(t_\ell)
>      = \frac{1}{N}\sum_{i=1}^N e^{i t_\ell^\top x_i}.
>      $$
>
>      Each empirical CF evaluation costs $O(Nd)$, so the overall cost is $O(LNd)$. Since $L$ is small and independent of $N$ in practice, SDD scales linearly with batch size and is substantially cheaper than quadratic MMD at moderate to large $N$. We summarize this analysis in the main text and detail it in **Appendix E**, **Page 28**.
>
> 2. **Empirically, CSDM is the only method among strong competitors that runs CIFAR-10 IPC 50 on a single RTX 4090, with much lower memory and time.**
>    In Appendix E.3, we report time and memory per iteration on CIFAR-10 for several strong baselines. For example:
>
>    **Time per iteration (seconds, CIFAR-10):**
>
>    | IPC | DATM |DC | MTT | CSDM |
>    | --- | ---- | ------- | ------- | ---- |
>    | 10  | OOM  | 48      |  0.4 | 0.30 |
>    | 20  | OOM  | 107     | 0.7 | 0.35 |
>    | 50  | OOM  | OOM     |  OOM | 0.33 |
>
>    **Memory (GB, CIFAR-10 on RTX 4090):**
>
>    | IPC | DATM | DC | MTT   | CSDM |
>    | --- | ---- | ------- | ------- |  ---- |
>    | 10  | OOM  | 10.89   | 10.84 | 3.43 |
>    | 20  | OOM  | 20.69   |  16.91 | 3.44 |
>    | 50  | OOM  | OOM     | OOM   | 3.45 |
>
>    CSDM is the only method that successfully distills CIFAR-10 with IPC 50 on a single 24GB GPU, with $\approx 3.5$GB memory usage and sub-second iteration time. While these tables do not include NCFM (due to limited space and the focus on distillation-time-heavy methods), they demonstrate that our SDD-based objective is extremely lightweight compared to DC/MTT-style baselines. Combined with the absence of an inner max loop and additional networks, this makes CSDM particularly attractive for high IPC and large-scale settings.

---

> ### Author Response · Authors · 2025-11-22
> **Q6: How universal-kernel distribution matching performs on balanced datasets**
>
> Thanks for asking about balanced settings; we agree this is important to show that CSDM is not only for long-tailed data.
>
>
>    The full balanced results are presented in **Appendix E** “Supplementary Experimental Results”, **Page 28**.
>
> | Dataset        | CIFAR-10         |                  |                  | CIFAR-100        |                  |
> |----------------|------------------|------------------|------------------|------------------|------------------|
> | IPC            | 1                | 10               | 50               | 10               | 50               |
> | Ratio (%)      | 0.02             | 0.2              | 2                | 2                | 10               |
> | Random         | 14.4 ± 0.2       | 26.0 ± 1.0       | 43.4 ± 1.0       | 15.1 ± 0.5       | 30.7 ± 0.4       |
> | Herding        | 21.5 ± 1.2       | 31.0 ± 1.0       | 46.3 ± 1.1       | 18.5 ± 1.0       | 34.5 ± 0.6       |
> | Forgetting     | 13.5 ± 1.2       | 23.3 ± 1.0       | 23.3 ± 1.1       | 15.1 ± 1.0       | 30.5 ± 0.7       |
> | DC             | 28.3 ± 0.5       | 44.9 ± 0.5       | 49.5 ± 0.5       | 25.5 ± 0.5       | 34.8 ± 0.5       |
> | DSA            | 28.8 ± 0.7       | 52.1 ± 0.5       | 54.5 ± 0.5       | 32.0 ± 0.5       | 42.8 ± 0.5       |
> | DCC            | 29.7 ± 0.4       | 50.4 ± 0.4       | 58.2 ± 0.5       | 30.0 ± 0.4       | 48.2 ± 0.5       |
> | DSAC           | 30.8 ± 0.5       | 51.4 ± 0.5       | 59.0 ± 0.5       | 29.0 ± 0.5       | 46.6 ± 0.5       |
> | FrePo          | 36.3 ± 0.5       | 55.3 ± 0.5       | 60.5 ± 0.4       | 35.5 ± 0.6       | 48.3 ± 0.4       |
> | MTT            | 46.3 ± 0.8       | 58.3 ± 0.7       | 67.6 ± 0.5       | 38.7 ± 0.6       | 45.7 ± 0.8       |
> | ATT            | 46.3 ± 0.8       | 57.5 ± 0.6       | 67.5 ± 0.5       | 39.3 ± 0.4       | 45.4 ± 0.7       |
> | FTD            | 46.8 ± 0.5       | 60.3 ± 0.4       | 67.9 ± 0.5       | 40.2 ± 0.5       | 45.4 ± 0.4       |
> | TESLA          | 46.3 ± 0.5       | 59.0 ± 0.4       | 67.9 ± 0.5       | 37.1 ± 0.5       | 44.1 ± 1.2       |
> | CAFE           | 30.3 ± 1.1       | 41.1 ± 1.1       | 46.4 ± 1.1       | 19.5 ± 0.7       | 35.7 ± 1.1       |
> | DM             | 43.0 ± 1.5       | 54.4 ± 1.1       | 67.6 ± 1.5       | 37.2 ± 1.4       | 46.6 ± 1.9       |
> | IDM            | 45.6 ± 1.0       | 58.6 ± 1.0       | 68.1 ± 1.0       | 38.2 ± 1.0       | 46.9 ± 0.5       |
> | M3D            | 47.1 ± 1.0       | 59.9 ± 0.5       | 69.3 ± 1.0       | 39.7 ± 1.0       | 47.6 ± 0.5       |
> | IID            | 47.1 ± 1.0       | 60.9 ± 1.0       | 70.1 ± 1.0       | 39.8 ± 1.0       | 47.2 ± 1.0       |
> | DSDM           | 46.5 ± 0.7       | 58.4 ± 0.7       | 67.0 ± 0.7       | 36.7 ± 0.7       | 45.4 ± 0.5       |
> | G-VBSM         | 46.5 ± 0.5       | 54.3 ± 0.5       | 60.4 ± 0.5       | 38.7 ± 0.7       | 45.7 ± 0.7       |
> | FreD           | **60.6 ± 0.8**   | 70.3 ± 0.3       | 75.8 ± 0.1       | 42.7 ± 0.2       | 47.8 ± 0.1       |
> | NCFM           | 49.5 ± 0.3       | **71.8 ± 0.3**   | 77.4 ± 0.3       | 48.7 ± 0.5       | **54.7 ± 0.2**   |
> | **CSDM (ours)**| 47.3 ± 0.3       | 71.2 ± 0.4       | **78.1 ± 0.4**   | **50.4 ± 0.3**   | 54.1 ± 0.2       |
> | Whole Dataset  |        |                  |       84.8 ± 0.1           |        |         56.2 ± 0.3         |

---

> ### Author Response · Authors · 2025-11-22
> **Q7: Responses to minor comments (figure captions, notation, typos)**
>
> Thanks for the careful reading and concrete suggestions; we have updated the draft accordingly.
>
> 1. **Figure 1 caption (“Minimize previous metric”) has been clarified.**
>    We now explicitly state that the “previous metric” refers to the **linear-kernel MMD objective used in DM** (Zhao & Bilen, Dataset Condensation with Distribution Matching). All distribution-matching baselines that do not explicitly change the discrepancy metric use this linear MMD. The caption has been revised to avoid ambiguity.
>
> 2. **Figure 2 will be enlarged and simplified to improve readability.**
>    We acknowledge that the original figure was dense and small. In the revised version, we will (i) increase the figure size, (ii) reduce the amount of text inside the diagram, and (iii) move some explanatory labels into the main text, so that the overall pipeline remains clear without overwhelming the reader.
>
> 3. **MMD notation has been made consistent and the role of the function class is now stated explicitly.**
>    You are right that we originally used both $\mathrm{MMD}[\mathcal{F},P,Q]$ and $\mathrm{MMD}(P,Q)$ somewhat interchangeably. In the revision:
>
>    * We use $\mathrm{MMD}[\mathcal{F},P,Q]$ when we want to emphasize the general IPM form over a function class $\mathcal{F}$.
>    * Once a specific kernel $k$ is fixed, we explicitly set $\mathcal{F}$ to be the **unit ball of the RKHS induced by $k$**, and then write $\mathrm{MMD}_k(P,Q)$ or simply $\mathrm{MMD}(P,Q)$ for brevity.
>
>
> 4. **The typo in Section 1 (“we attribute the drawbacks of previous DD methods to…”) has been corrected.**
>    We fixed this and several other minor typos spotted during the revision pass.
>
> ---
>
> Once again, we thank Reviewer 1J5y for the detailed and constructive feedback. The clarifications, added theory, and new experiments directly address your concerns about NCFM comparison, amplitude/phase interpretation, the choice of $\alpha(c)$, complexity, and balanced-dataset performance.

---

> ### Author Response · Authors · 2025-11-26
> **In Appreciation of Your Review: Supplementary Materials and Clarifications Submitted**
>
> Dear Reviewer 1J5y,
>
>
>
> Thank you once again for your thoughtful and detailed feedback on our submission. In response to your initial review—which raised several important concerns, some of which stemmed from **misunderstandings that significantly impacted the evaluation**—we have taken substantial steps to address them.
>
> Specifically, we have prepared a comprehensive **15-page appendix** containing rigorous theoretical proofs and additional experimental results to directly clarify the points in question. We believe these additions fully resolve the misconceptions and provide strong support for our claims.
>
> As the discussion phase is now drawing to a close on **December 3, AoE**, we kindly ask whether our responses and supplementary materials have adequately addressed your concerns (at least in part). If any questions remain or if further clarification would be helpful, please do not hesitate to let us know. Conversely, if you find our clarifications satisfactory, **we would be truly grateful if you could consider revising your evaluation scores accordingly—especially given the significance of the corrected misunderstandings to the overall assessment.**
>
> Given the timeline of the review process, we sincerely hope to receive your final feedback before the discussion window closes.
>
> Thank you very much for your time and consideration.

---

> ### Author Response · Authors · 2025-11-27
> **Follow-Up on Rebuttal Clarifications**
>
> Dear reviewer 1J5y,
>
> Thank you very much for your valuable feedback on our paper.
>
> We wanted to follow up regarding the response we posted earlier, where we addressed all your comments, including **a clear clarification of the fundamental differences between our method and FreD**, the **theoretical and experimental comparisons with existing high-order matching approaches**, the **additional balanced CIFAR experiments**, and the **complexity, runtime, and memory analyses and corresponding experiments**, among others.
>
> If there are any remaining concerns, or if any of our clarifications (in the new Section 6, the extended appendix, or the official comments) would benefit from further explanation, please feel free to let us know — we would be glad to provide any additional details.
>
> If you feel that the revised manuscript and rebuttal have adequately addressed your main concerns, we would greatly appreciate it if you could reflect this in your final comments and score.
>
> Best,
> Authors

---

### Official Review · Reviewer_5N3L · 2025-10-28

**Soundness:** 3
**Presentation:** 3
**Contribution:** 3
**Rating:** 4
**Confidence:** 4

**Summary:**

This paper addresses the problem of dataset distillation on long-tailed (imbalanced) datasets and identifies two key limitations of prior methods:

(1) they rely on simplistic distribution matching metrics (e.g. MSE or linear-kernel MMD) that only align first-order statistics (means), failing to truly match distributions.

(2) they treat all classes uniformly, which is ineffective when some classes have far fewer samples.

To overcome these issues, the authors propose Class-Aware Spectral Distribution Matching (CSDM). The core idea is to use a universal kernel and Bochner’s theorem to map data into a frequency domain and define a Spectral Distribution Distance (SDD) that can distinguish full distributions rather than just their means.

They further decompose this spectral difference into amplitude (related to diversity of samples) and phase (related to realistic detail) components. These disentanglement could be especially effective, when we combine these into dataset distillation, whose budget is limited and still require diversity and detailedness.

By assigning class-specific weights to these components, CSDM emphasizes realistic fidelity for under-represented tail classes while maintaining diversity for head classes, thus dynamically handling class imbalance. This principled metric design allows the distilled synthetic set to better capture the overall data distribution, especially the rare classes that previous approaches synthesized poorly.

**Strengths:**

By assigning class-specific weights to these components, CSDM emphasizes realistic fidelity for under-represented tail classes while maintaining diversity for head classes, thus dynamically handling class imbalance. This principled metric design allows the distilled synthetic set to better capture the overall data distribution, especially the rare classes that previous approaches synthesized poorly.

Empirically, CSDM shows substantial gains: on the severely imbalanced CIFAR-10-LT benchmark (imbalance factor 200), using just 10 synthetic images per class, it improves accuracy by 14.0% over the best existing methods, and its performance only drops by 5.7% even when tail classes’ images are reduced from 500 to 25 – indicating strong robustness to data scarcity.

Also, The approach is theoretically grounded (since MMD with a characteristic kernel ensures aligning entire distributions) and addresses a significant practical challenge in dataset distillation.

**Weaknesses:**

One potential concern is the added complexity of designing and computing the spectral kernel embedding, but the authors argue that the Fourier-based implementation is efficient. It would be informative if author provide pseudo-code or big-O notation about fourier-related computation.

Although this paper nicely derive spectral distribution-based dataset distillation, this paper lacks extensive search and discussions with existing methods. For example, FreD (https://arxiv.org/abs/2311.08819) primarily provides spectral distribution-based matching, but there is not any discussion about it. So i am leaning toward rejecting this paper, with lack of discussions and comparisons with current methods.

Also, author claims that explicit modeling of 1) diversity and 2) detailedness, but they do not demonstrate any distilled image compared to other methods, so it is hard for reviewer to conclude that performance improvements are mainly from these modelings. It would be really helpful.

Although i am leaning toward rejecting this manuscript, i could change my score if weaknesses above are treated well from rebuttal period.

**Questions:**

Discussed in weaknesses section.

---

> ### Author Response · Authors · 2025-11-22
> **Q1: Clarification on computational complexity**
>
> Thanks for the question. In response, we have added the complexity analysis of MMD and SDD in the main body of the paper. Here, we further clarify the computational costs by first reviewing the empirical estimators of standard kernel MMD and our SDD.
>
> 1. **SDD has linear complexity in batch size, whereas standard kernel MMD is quadratic.**
>    The empirical (biased) estimator of kernel MMD is:
>    $$\widehat{\mathrm{MMD}}^2(P,Q)= \frac{1}{N^2}\sum\_{i,j} k(x_i,x_j)+ \frac{1}{N^2}\sum\_{i,j} k(y_i,y_j)- \frac{2}{N^2}\sum_{i,j} k(x_i,y_j),$$
>      which requires evaluating all $N^2$ kernel pairs.
>      For a kernel $k$ whose evaluation costs $O(d)$ in a $d$-dimensional feature space, the total cost is:
>      $$
>      O(N^2 d).
>      $$
>
>    In contrast, SDD uses the spectral form of MMD and approximates the integral over frequencies via Monte Carlo sampling. Its empirical estimator is:
>
>    $$\widehat{\mathrm{SDD}}(P,Q)= \frac{1}{L}\sum\_{\ell=1}^L\Bigl|\hat{\varphi}\_P(t_\ell) - \hat{\varphi}\_Q(t_\ell)\Bigr|^2,$$
>
>    where the empirical characteristic function is
>
>    $$
>    \hat{\varphi}\_P(t\_\ell)
>    = \frac{1}{N}\sum\_{i=1}^N e^{i t_\ell^\top x_i}.
>    $$
>
>    Computing $\hat{\varphi}\_P(t\_\ell)$ costs $O(Nd)$, and the same holds for $Q$. Therefore, for $L$ Monte Carlo samples:
>
>    $$
>    \qquad O(LNd).
>    $$
>
>    Since $L$ is typically small and *independent of $N$*, SDD scales **linearly** in batch size, yielding a substantial advantage in large-batch or high-IPC settings.
>
> 2. **This theoretical advantage translates into practical efficiency.**
>    As shown in our newly added runtime and memory comparisons (**Appendix E**, Tables 6, **Page 30**):
>
>    * CSDM is the **only** method able to finish **CIFAR-10 IPC 50** on a single RTX 4090, requiring only **≈3.5 GB** VRAM.
>    * DC, MTT, and DATM **run out of memory** under the same setup.
>    * CSDM is **hundreds of times faster** than DC in wall-clock time.
>
>     These results confirm that SDD’s linear complexity is not only theoretically appealing but also crucial for making high-IPC distillation feasible.
> #### Time consumption comparison (seconds/iteration)
> | Dataset | CIFAR-10 |        |        |
> | :------ | :------- | :----- | :----- |
> | IPC     | 10       | 20     | 50     |
> | DATM    | OOM      | OOM    | OOM    |
> |DC | 48       | 107    | OOM    |
> | MTT     | 0.4      | 0.7    | OOM    |
> | CSDM    | 0.3      | 0.35   | 0.33   |
>
> #### Memory consumption comparison
> | Dataset | CIFAR-10 |          |          |
> | :------ | :------- | :------- | :------- |
> | IPC     | 10       | 20       | 50       |
> | DATM    | OOM      | OOM      | OOM      |
> |DC | 10.89GB  | 20.69GB  | OOM      |
> | MTT     | 10.84GB  | 16.91GB  | OOM      |
> | CSDM    | 3.43GB   | 3.44GB   | 3.45GB   |

---

> ### Author Response · Authors · 2025-11-22
> **Q2: Relation to FreD and whether CSDM overlaps with frequency-parameterization methods.**
>
> Thanks for raising this. Earlier literature rarely addressed dataset distillation in the frequency domain. FreD and NSD are two pioneering works that have achieved outstanding performance, providing compelling evidence for the power of frequency-based approaches. While both CSDM and FreD involve frequency-domain ideas, they intervene in **completely different components** of the distillation pipeline.
>
> 1. **Parameterization vs. metric: the two orthogonal axes.**
>    Dataset distillation can be expressed as:
>
>    $$
>    \min\_{\theta}
>    \mathcal{L}_{\text{metric}}\left(
>    \Phi\_{\text{syn}}(\theta),
>    \Phi\_{\text{real}}(D)
>    \right),
>    $$
>
>    where:
>
>    * $\Phi_{\text{syn}}(\theta)$ determines **how synthetic data are represented**,
>    * $\mathcal{L}_{\text{metric}}$ determines **how real and synthetic distributions are compared**.
>
>    Under this view:
>
>    * **FreD** innovates in $\Phi_{\text{syn}}$: it *parameterizes* synthetic images using truncated Fourier coefficients, while still relying on gradient/trajectory matching as the metric.
>    * **CSDM** innovates in $\mathcal{L}_{\text{metric}}$: it replaces linear-kernel MMD with a universal, spectral distribution distance (SDD), together with class-aware amplitude–phase reweighting.
>
>    Thus, FreD and CSDM operate on **orthogonal components**.
>
> 2. **Clarification added.**
>    We added an explicit subsection in the revised **Appendix D** (“Comparison with Frequency-domain Distillation Methods”, **Page 28**), clarifying this distinction and revising the Related Work accordingly.

---

> ### Author Response · Authors · 2025-11-22
> **Q3: Clarification on the “diversity vs. realism” interpretation of amplitude and phase**
>
> Thanks for the question. This point is closely related to Reviewer fbvC’s inquiry, and we provide a clear summary here based on the expanded appendix.
>
> 1. **Amplitude primarily encodes spectral diversity; phase primarily encodes structural realism.**
>    Classical image/signal processing (e.g., Oppenheim & Lim 1981) shows that:
>
>    * The **phase** spectrum carries geometric/structural information (edges, shapes).
>    * The **amplitude** spectrum controls how energy spreads across frequencies, governing variability and texture.
>
>    Our empirical visualizations added in the **Appendix E**, **Page 31**) confirm this:
>
>    * amplitude-only distillation yields diverse but less coherent images,
>    * phase-only distillation yields realistic but low-diversity samples.
>
> 2. **Long-tailed distillation requires different behaviors for head vs. tail classes.**
>
>    * Head classes risk *mode collapse* (lack of diversity), so we weight the amplitude term more heavily.
>    * Tail classes risk *semantic drift* (loss of realism), so we weight the phase term more heavily.
>
>    This motivates the class-specific coefficients $\alpha(c)$.
>
> 3. **The metric remains discriminative.**
>    Proposition 1 (added in the **Appendix A**, **Page 22**) proves that for any $\alpha(c)\in(0,1)$, the class-aware discrepancy:
>    $$
>    d(F_T, F_S)
>    = \sum_c d_c(F_T^c, F_S^c)
>    $$
>    equals zero **if and only if** the real and synthetic distributions match.
>    Thus, the amplitude/phase decomposition changes the *optimization landscape* but **does not compromise metricity or identifiability**.
>
> 4. **Summary.**
>    The diversity/realism interpretation is well-grounded in classical Fourier theory, supported by new visualizations, and mathematically consistent with our class-aware SDD formulation.

---

> ### Author Response · Authors · 2025-11-26
> **In Appreciation of Your Review: Supplementary Materials and Clarifications Submitted**
>
> Dear Reviewer 5N3L,
>
>
>
> Thank you once again for your thoughtful and detailed feedback on our submission. In response to your initial review—which raised several important concerns, some of which stemmed from **misunderstandings that significantly impacted the evaluation**—we have taken substantial steps to address them.
>
> Specifically, we have prepared a comprehensive **15-page appendix** containing rigorous theoretical proofs and additional experimental results to directly clarify the points in question. We believe these additions fully resolve the misconceptions and provide strong support for our claims.
>
> As the discussion phase is now drawing to a close on **December 3, AoE**, we kindly ask whether our responses and supplementary materials have adequately addressed your concerns (at least in part). If any questions remain or if further clarification would be helpful, please do not hesitate to let us know. Conversely, if you find our clarifications satisfactory, **we would be truly grateful if you could consider revising your evaluation scores accordingly—especially given the significance of the corrected misunderstandings to the overall assessment.**
>
> Given the timeline of the review process, we sincerely hope to receive your final feedback before the discussion window closes.
>
> Thank you very much for your time and consideration.

---

> ### Author Response · Authors · 2025-11-27
> **Follow-Up on Rebuttal Clarifications**
>
> Dear reviewer 5N3L,
>
> Thank you very much for your valuable feedback on our paper.
>
> We wanted to follow up regarding the response we posted earlier, where we addressed all your comments, including **a clear clarification of the fundamental differences between our method and FreD**, the **theoretical and experimental comparisons with existing high-order matching approaches**, the **additional balanced CIFAR experiments**, and the **complexity, runtime, and memory analyses and corresponding experiments**, among others.
>
> If there are any remaining concerns, or if any of our clarifications (in the new Section 6, the extended appendix, or the official comments) would benefit from further explanation, please feel free to let us know — we would be glad to provide any additional details.
>
> If you feel that the revised manuscript and rebuttal have adequately addressed your main concerns, we would greatly appreciate it if you could reflect this in your final comments and score.
>
> Best,
> Authors

---

> ### Author Response · Authors · 2025-11-27
> **Supplementary Analysis on Frequency-Oriented Baselines**
>
> Dear Reviewer 5N3L,
>
> We sincerely thank you for highlighting frequency-based methods, including **FreD**, in your review. We would like to point out that **we have now explicitly included and discussed these approaches** in both the Related Work section and throughout the rebuttal, providing **detailed comparisons and theoretical clarifications**.
>
> We kindly invite you to **re-examine the updated manuscript and supplemental materials**, as we believe these additions comprehensively address the concerns regarding frequency-based methods and clarify the distinctions and connections with our proposed CSDM framework.
>
> We greatly appreciate your time and careful consideration of the updated material.
>
> Sincerely,
>
> Authors

---

### Official Review · Reviewer_fbvC · 2025-10-31

**Soundness:** 3
**Presentation:** 3
**Contribution:** 2
**Rating:** 4
**Confidence:** 5

**Summary:**

This research claims that the previous dataset distillation methods rely on heuristic metrics, which only align first moment; and treat all classes uniformly, which might be fail on long-tailed datasets. To mitigate this problem, this research suggests Class-aware spectral distribution matching (CSDM), which formulates distribution matching in the spectral domain and amplitude-phase decomposition with class-specific weighting. CSDM employs Spectral distribution distance (SDD), which is derived from kernel theory with universal kernel. This approach enables better distribution matching and balanced learning for class imbalanced dataset.

**Strengths:**

1. The point raised that many distribution matching methods employ linear kernel, which fail to satisfy universality, is a theoretical limitation of considerable merit.

2. This manuscript is well written and easy to reading. Furthermore, the core idea of CSDM is simple and intuitive, making it easy to understand and apply.

3. In high imbalanced factor experiments, CSDM achieved significant performance gap compared to the baseline, which constitutes highly appropriate results for long-tailed dataset distillation, the primary target of this research.

**Weaknesses:**

1. Theoretical and experimental comparisons with several previous studies[1,2,3,4,5] that performed distribution matching considering higher moments are lacking. In particular, there is insufficient discussion regarding M3D[1] and IID[2] despite being mentioned in this paper. Furthermore, the only experimental comparison with the core baseline, NCFM[4], is on the long-tailed CIFAR dataset. Therefore, there is insufficient justification to develop the argument solely based on the shortcomings of first-moment distribution matching.

2. SDD lacks novelty and contribution as it is not newly proposed in this paper. Firstly, Theorem 3, stating that Squared MMD can be expressed as a Characteristic function, is already established theory (see Corollary 4 in [6]). Furthermore, SDD is also widely used in various fields under the name Characteristic function distance (CFD)[4,7,8]. However, as this paper does not reference relevant previous works, it could give the impression that these concepts were first proposed herein. Therefore, since the content related to SDD is already a widely known and used concept, I believe this paper lacks sufficient novelty and contribution.

3. The second idea, the class-specific coefficient, is likewise merely a naïve extension of prior research. As mentioned in this paper, the idea of decomposing the discrepancy in the characteristic function into amplitude and phase discrepancies is well-established. Furthermore, the concept of introducing a weight parameter $\alpha$ to adjust these two components also already exists [4,8]. Therefore, in this paper's CSDM, $\alpha$ is simply changed to a class-specific weight $\alpha (c)$. Furthermore, since this value is not determined systematically but is treated as a hyperparameter, I consider it to lack novelty and contribution. Moreover, there is no adequate analysis as to whether the introduction of the class-specific weight $\alpha (c)$ does not hinder the original objective of distribution matching and can still satisfy optimality.

4. There is no complexity analysis based on theoretical and experimental grounds. Calculating the SDD requires the computation of a characteristic function involving the expectation of complex numbers and Monte-Carlo sampling, resulting in a nested summation form. This can incur significant computational cost, necessitating a complexity analysis.

5. Key previous literature is missing. FreD[9] and NSD[10] are prior studies addressing the frequency domain in dataset distillation and should therefore be mentioned in the Related Works section.


[1] M3D: Dataset Condensation by Minimizing Maximum Mean Discrepancy

[2] Exploiting Inter-sample and Inter-feature Relations in Dataset Distillation

[3] Diversified Semantic Distribution Matching for Dataset Distillation

[4] Dataset Distillation with Neural Characteristic Function: A Minmax Perspective

[5] Dataset Distillation via the Wasserstein Metric

[6] Hilbert Space Embeddings and Metrics on Probability Measures

[7] A Characteristic Function Approach to Deep Implicit Generative Modeling

[8] Reciprocal Adversarial Learning via Characteristic Functions

[9] Frequency Domain-based Dataset Distillation

[10] Neural Spectral Decomposition for Dataset Distillation

**Questions:**

1. SDD (or CFD) is not an idea limited for long-tailed dataset distillation. Therefore, to argue that existing distribution matching methods suffer from inadequate alignment, I believe performance comparisons in class-balanced dataset distillation are also necessary.

2. I am curious as to why the baseline for Long-tailed ImageNet is less than that for Long-tailed CIFAR.

3. I am also curious whether minimizing the convex combination of amplitude-phase components still satisfies distribution matching under optimal conditions.

4. I suggest dividing Section 5.4 into separate sections. A comparison with NCFM would be more appropriately placed in Section 4.1 or 4.2. Table 5 is redundant as it duplicates the experimental results presented in Table 2.

---

> ### Author Response · Authors · 2025-11-22
> **Q1-1: About comparing methods that match higher-order information (IID / DSDM / M3D / NCFM vs. CSDM).**
>
> Thanks for this question. We address it from four angles: empirical comparison, the nature of IID/DSDM’s “high-order” terms, the all-order nature of universal-kernel MMD, and the role/limitations of NCFM’s min–max CFD.
>
> 1. **Empirical comparison on balanced CIFAR-10/100.**
>    On standard balanced datasets, CSDM is competitive or better than prior high-order methods, especially at moderate/large IPC. From our new Table 5 (reformatted here for the relevant methods):
>
>    | Method          | CIFAR-10 IPC 1 | CIFAR-10 IPC 10 | CIFAR-10 IPC 50         | CIFAR-100 IPC 10        | CIFAR-100 IPC 50        |
>    | --------------- | -------------- | --------------- | ----------------------- | ----------------------- | ----------------------- |
>    | M3D             | $47.1 \pm 1.0$ | $59.9 \pm 0.5$  | $69.3 \pm 1.0$          | $39.7 \pm 1.0$          | $47.6 \pm 0.5$          |
>    | IID             | $47.1 \pm 1.0$ | $60.9 \pm 1.0$  | $70.1 \pm 1.0$          | $39.8 \pm 1.0$          | $47.2 \pm 1.0$          |
>    | DSDM            | $46.5 \pm 0.7$ | $58.4 \pm 0.7$  | $67.0 \pm 0.7$          | $36.7 \pm 0.7$          | $45.4 \pm 0.5$          |
>    | NCFM            | $\mathbf{49.5 \pm 0.3}$ | $\mathbf{71.8 \pm 0.3}$  | $77.4 \pm 0.3$          | $48.7 \pm 0.5$          | $\mathbf{54.7 \pm 0.2}$ |
>    | **CSDM (ours)** | $47.3 \pm 0.3$ | $71.2 \pm 0.4$  | $\mathbf{78.1 \pm 0.4}$ | $\mathbf{50.4 \pm 0.3}$ | $54.1 \pm 0.2$          |
>
>    Concretely:
>
>    * On CIFAR-10, IPC 50, CSDM achieves $78.1%$, **higher than all baselines**, including NCFM ($77.4%$), IID ($70.1%$), M3D ($69.3%$), and DSDM ($67.0%$).
>    * On CIFAR-100, IPC 10, CSDM reaches $50.4%$, clearly above NCFM ($48.7%$) and significantly above IID/M3D/DSDM (around $40%$–$48%$).
>
>    These results indicate that CSDM is not just another “moment-augmented” variant; it yields a consistently stronger discrepancy on both CIFAR-10 and CIFAR-100.
>
> 2. **IID/DSDM’s “high-order” terms match only low-order derivatives of the log characteristic function.**
>    We show in **Theorem 7** (Page 25) that mean–covariance losses can be reinterpreted as matching the first two derivatives of the log characteristic function at $t=0$.
>
>    Specifically, for distributions $P,Q$ with means $\mu_P,\mu_Q$ and covariances $\Sigma_P,\Sigma_Q$, let $\varphi_P,\varphi_Q$ be their characteristic functions and $\psi_P = \log\varphi_P$, $\psi_Q = \log\varphi_Q$. Then:
>
>    * The first derivative of $\psi_P$ at the origin encodes the mean:
>      $$\nabla_t \psi_P(0) = i \mu_P,\quad\nabla_t \psi_Q(0) = i \mu_Q.$$
>    * The Hessian encodes the covariance:
>      $$\nabla_t^2 \psi_P(0) = -\Sigma_P,\quad\nabla_t^2 \psi_Q(0) = -\Sigma_Q.$$
>
>    Hence a loss
>    $$L_{\text{mean+cov}}(P,Q)= |\mu_P - \mu_Q|_2^2+ \lambda|\Sigma_P - \Sigma_Q|_F^2$$
>      can be written equivalently as
>      $$\big|\nabla_t \psi_P(0) - \nabla_t \psi_Q(0)\big|_2^2+ \lambda \big|\nabla_t^2 \psi_P(0) - \nabla_t^2 \psi_Q(0)\big|_F^2,$$
>      i.e., it matches only the **first two cumulants** of $P$ and $Q$.
>
>    IID and DSDM add exactly such mean/covariance (or covariance-like) penalties, class-wise. Our analysis shows these are **intrinsically finite-order** (second-order) approximations: they match a local Taylor expansion of $\log\varphi$ around $t=0$, and thus cannot provide full distributional discrimination by construction. This clarifies that the “add high-order regularizers” line is fundamentally a finite-order, empirical exploration rather than a complete solution.

---

> > ### Author Response · Authors · 2025-11-22
> > **Q1-2: About comparing methods that match higher-order information (IID / DSDM / M3D / NCFM vs. CSDM).**
> >
> > 3. **RBF MMD (universal kernels) implicitly match all orders of moments.**
> >    In contrast, we prove in **Appendix A** “MMD and Universal Kernels” (**Page 16**) and “Moment Expansion of Gaussian Kernel MMD” (**Page 19**) that MMD with a universal kernel (e.g., RBF) is a **complete** discrepancy.
> >
> >    * From the universal-kernel result (Theorem $\text{A.4}$ in the appendix), if $k$ is universal then
> >      $$
> >      \text{MMD}[\mathcal{F},P,Q] = 0
> >      \iff
> >      P = Q,
> >      $$
> >      so the kernel mean embedding $P\mapsto \mu_P$ is injective and MMD is a **true metric** on distributions.
> >
> >    * For the Gaussian kernel, we explicitly expand $\text{MMD}^2$ as
> >      $$
> >      \text{MMD}^2(P,Q)
> >      = \sum_{\alpha\in\mathbb{N}^d}
> >      w_\alpha \bigl(m_{P,\alpha} - m_{Q,\alpha}\bigr)^2,
> >      $$
> >      where each $m_{P,\alpha}$ is a weighted moment and $w_\alpha > 0$. We then show that the sequence of weighted moments ${m_{P,\alpha}}$ and the original moment sequence ${\mathbb{E}[X^\beta]}$ uniquely determine each other (via an “upper-triangular” linear relationship). Under standard moment-determinacy conditions, this implies all moments—and hence the distribution—are matched.
> >
> >    Therefore, universal-kernel MMD (and its spectral form SDD) is fundamentally different from explicitly matching only second- or third-order statistics. It implicitly encodes **all-order** moment information and is provably distribution-discriminative.
> >
> > 4. **NCFM’s min–max CFD.**
> >    NCFM introduces a Neural CFD where a sampling network over frequencies is trained in a min–max game: the sampler **maximizes** the characteristic-function discrepancy, while the synthetic data **minimize** it. This design is powerful but has two structural limitations:
> >
> >    * First, a generic CFD with an arbitrary learned weight $\omega(t)$ is **not guaranteed** to be metric or characteristic; additional conditions are needed on $\omega$ to ensure it defines a valid distance. Our SDD, in contrast, is built from the spectral measure of a universal kernel, and therefore **inherits MMD’s metric and characteristic properties by construction**.
> >    * Second, the “max” step optimizes CFD itself, which is **not necessarily aligned** with downstream classification accuracy, especially under severe class imbalance. The sampler can over-focus on frequency regions dominated by head-class data and underweight tail-class structure.
> >
> >    Empirically:
> >
> >    * On balanced CIFAR-10, NCFM is indeed very strong (and slightly ahead of us at IPC 10), but at IPC 50 we surpass it ($78.1%$ vs. $77.4%$). On balanced CIFAR-100, CSDM is stronger at IPC 10 ($50.4%$ vs. $48.7%$).
> >    * On long-tailed benchmarks (CIFAR-10-LT and ImageNet-LT, reported in the main text and appendix), NCFM **degrades markedly as imbalance grows**, whereas CSDM remains robust and achieves consistently higher accuracy.
> >
> >    These results support our choice: instead of starting from a heuristic min–max CFD, we design SDD as the spectral form of a universal-kernel MMD, then introduce **class-aware amplitude–phase weighting** that is analytically controlled (Proposition 1) and empirically robust in long-tailed regimes.
> >
> >    Detailed derivations and comparisons appear in **Appendix B** (“Dataset Distillation by Aligning High-Order Information”, **Page 23**), **Appendix A** (“MMD and Universal Kernels”, **Page 16**), and **Appendix A** (“SDD versus CFD”, **Page 20**).
> >
> > ### CIFAR-10
> > | Imbalance Factor | 1            |     |              | 10           |              |              | 50           |              |              | 100          |              |              | 200          |              |              |
> > | ---------------- | ------------ | --- | ------------ | ------------ | ------------ | ------------ | ------------ | ------------ | ------------ | ------------ | ------------ | ------------ | ------------ | ------------ | ------------ |
> > | IPC              | 10           | 20  | 50           | 10           | 20           | 50           | 10           | 20           | 50           | 10           | 20           | 50           | 10           | 20           | 50           |
> > | **NCFM**         | **71.8±0.3** | -   | 77.4±0.3     | 70.2±0.3     | 71.2±0.4     | 75.6±0.2     | 68.8±0.3     | 71.3±0.4     | 72.2±0.3     | 60.4±0.6     | 60.1±0.7     | 59.3±0.9     | 57.7±1.0     | 57.0±1.4     | 56.1±1.3     |
> > | **CSDM**         | 71.2±0.4     | -   | **78.1±0.4** | **71.0±0.4** | **73.9±0.1** | **76.5±0.2** | **69.3±0.3** | **71.6±0.2** | **73.4±0.4** | **67.4±0.3** | **69.4±0.3** | **71.0±0.4** | **65.3±0.3** | **66.1±0.5** | **66.8±0.6** |
> > |                  |              |     |              |              |              |              |              |              |              |              |              |              |              |              |              |

---

> ### Author Response · Authors · 2025-11-22
> **Q2: About the difference between SDD and generic CFD.**
>
> Thanks for pointing this out. Although both SDD and prior methods use characteristic functions, they are motivated and constructed differently, and they do not define the same distance in general.
>
> 1. **Formally similar, but SDD is a kernel-induced, not arbitrary, CFD.**
>    Generic CFD often takes the form
>    $$
>    \text{CFD}(P,Q)
>    = \int_{\mathbb{R}^d} |\varphi_P(t) - \varphi_Q(t)|\omega(t)dt,
>    $$
>    where $\omega$ is an arbitrary weight. In contrast, SDD is defined as
>    $$
>    \text{SDD}(P,Q)
>    = \int_{\mathbb{R}^d}|\varphi_P(t) - \varphi_Q(t)|^2d\mu(t),
>    $$
>    where $\mu$ is the **spectral measure of a bounded, shift-invariant kernel** obtained via Bochner’s theorem.
>
>    The crucial difference is that $d\mu(t)$ is not arbitrary: it is determined by a kernel $k$ such that MMD with $k$ is known to be a valid, and often universal, IPM. Hence SDD is, by construction, exactly the squared MMD in spectral form. This automatically transfers **metricity and characteristicness** from the kernel MMD to SDD, which generic CFD with free $\omega$ does not guarantee.
>
> 2. **Topological equivalence with CFD based on the same $\mu$, but different function norms.**
>    To make the connection precise, we define
>
>    $$\text{CFD}_\mu(P,Q)= \int |\varphi_P(t) - \varphi_Q(t)|d\mu(t).$$
>
>    Using the facts that characteristic functions are bounded by 1 and $\mu(\mathbb{R}^d)<\infty$ (kernel boundedness), we show:
>
>    $$\text{CFD}_\mu(P,Q)\le \sqrt{\mu(\mathbb{R}^d)}\text{SDD}(P,Q)^{1/2},\quad\text{SDD}(P,Q)\le 2\text{CFD}_\mu(P,Q).$$
>
>    As a result, for any sequence ${P_n}$,
>    $$
>    \text{SDD}(P_n,P)\to 0
>    \iff
>    \text{CFD}_\mu(P_n,P)\to 0,
>    $$
>    so the two metrics induce the **same notion of convergence** on the space of probability measures. However, they live in different normed spaces ($L^2(\mu)$ vs. $L^1(\mu)$), and in general there is no single constant that makes them equivalent norms on the underlying function space. For this reason, we carefully state they are equivalent as **metrics/topologies**, not as norms.
>
> 3. **Kernel theory guides SDD’s design and suggests a roadmap for future CFD research.**
>    Our main conceptual contribution here is to **anchor spectral metrics in kernel theory**. Instead of manually choosing or learning $\omega(t)$, we:
>
>    * Start from a **universal, shift-invariant kernel** $k$ with well-understood properties.
>    * Use Bochner’s theorem to obtain its spectral measure $\mu$.
>    * Interpret the resulting SDD as the squared MMD expressed in the spectral domain.
>
>    This systematically guarantees that SDD is a strict metric when $k$ is universal, and it provides a clear recipe for designing future characteristic-function-based discrepancy measures: pick a kernel suitable for the downstream task, derive its spectral measure, and possibly introduce structured reweighting (such as our amplitude–phase decomposition) while preserving discriminativity.
>
>    The detailed SDD/CFD comparison and inequalities are provided in **Appendix A.3** “SDD versus CFD: why we still adopt a kernel-based perspective” **Page 20**.

---

> ### Author Response · Authors · 2025-11-22
> **Q3: About the design and reasonableness of class-specific $\alpha(c)$.**
>
> Thanks for raising this. You are right that amplitude–phase decompositions are widely studied in signal processing; our use of class-specific $\alpha(c)$ is motivated by a different question: **how to adapt a principled distribution metric to head and tail classes in long-tailed distillation**, where the statistical challenges are fundamentally asymmetric.
>
> 1. **Role of $\alpha(c)$: different priorities for head vs. tail classes.**
>    Classical results (e.g., Oppenheim & Lim 1981) show that in Fourier analysis:
>
>    * The **phase** spectrum largely determines structural content and recognizability.
>    * The **amplitude** spectrum controls energy allocation and “style”/variability.
>
>    In long-tailed distillation, head and tail classes face different failure modes:
>
>    * Head classes have many real samples; the main danger is **loss of diversity**.
>    * Tail classes have few samples; the main danger is **loss of realism or semantic drift**.
>
>    Our class-specific coefficient $\alpha(c)$ is precisely designed to reflect this asymmetry:
>
>    * For head classes $c$, we choose $\alpha(c)$ closer to $1$, boosting the **amplitude difference** term and encouraging coverage of diverse spectral modes.
>    * For tail classes $c$, we set $\alpha(c)$ smaller, boosting the **phase difference** term and emphasizing structural fidelity of each synthetic example.
>
>    **Appendix C** “The Role of Amplitude and Phase in Diversity and Realism” (**Page 26**) support this: amplitude-only distillation yields diverse but less realistic images, while phase-only distillation yields realistic but less diverse images. Our class-aware combination gives the best long-tailed performance, and reversing this weighting significantly harms accuracy.
>
> 2. **Discriminativity preserved: $d(F_T,F_S)=0$ still implies $P_T=P_S$.**
>    We appreciate your concern about whether introducing $\alpha(c)$ affects the optimality of the metric. To address this, we added **Proposition 1** (Page 22) in the appendix, which proves:
>
>    * For each class $c$ and each frequency $t$, the integrand
>      $$\alpha(c)||\varphi_T^c(t)| - |\varphi_S^c(t)||^2 + 2(1-\alpha(c))|\varphi_T^c(t)||\varphi_S^c(t)|
>        \big(1-\cos(\theta_T^c(t)-\theta_S^c(t))\big)$$
>        is nonnegative.
>    * If $\alpha(c)\in(0,1)$ and the class-wise distance $d_c(F_T^c,F_S^c)$ is zero, then:
>
>      * Amplitude and phase must match for $\mu$-almost every $t$.
>      * Using the continuity of characteristic functions and the fact that the spectral density is strictly positive, this implies $\varphi_T^c(t)\equiv \varphi_S^c(t)$ for all $t$.
>      * By the uniqueness theorem for characteristic functions, $F_T^c = F_S^c$ for each class $c$.
>    * With equal label priors (which our protocol enforces), it follows that the joint distributions coincide:
>      $$
>      P_T = P_S
>      \iff
>      d(F_T,F_S) = 0.
>      $$
>
>    Thus, $\alpha(c)\in(0,1)$ is not just heuristic; it is explicitly chosen so that the **class-aware distance remains characteristic**. The coefficients reshape the optimization landscape (emphasizing diversity or realism per class) but do not weaken the notion of “distribution equality”.
>
>    The full statement and proof of **Proposition 1** are given on **Page 21, 25,26**.

---

> ### Author Response · Authors · 2025-11-22
> **Q4: About complexity analysis and runtime/memory comparison.**
>
> Thanks for the suggestion. We now provide both a formal complexity comparison and concrete empirical runtime/memory results.
>
> 1. **Asymptotic complexity: SDD is $O(LNd)$ vs. standard MMD’s $O(N^2 d)$.**
>    For a batch of $N$ samples in $d$-dimensional feature space:
>
>    * Standard kernel MMD uses all pairwise kernel evaluations, leading to
>      $$
>      O(N^2 d)
>      $$
>      complexity for each MMD computation.
>    * Our SDD estimator uses $L$ Monte Carlo samples of frequencies $t_\ell$:
>
>      $$\widehat{\text{SDD}}(P,Q)= \frac{1}{L}\sum_{\ell=1}^L\bigl|\hat{\varphi}\_P(t_\ell) - \hat{\varphi}_Q(t\_\ell)\bigr|^2,$$
>
>      where each empirical characteristic function $\hat{\varphi}\_P(t\_\ell)$ is an average of $N$ complex exponentials with cost $O(Nd)$.
>
>    Therefore the total complexity is
>    $$
>    O(LNd),
>    $$
>    where $L$ depends on the desired Monte Carlo accuracy but is independent of $N$. For large $N$, this linear dependence in $N$ is a significant improvement over the quadratic dependence of standard MMD.
>
> 2. **Empirical runtime/memory: CSDM is substantially more efficient than DC/MTT/-style methods.**
>    In Tables 6 of the appendix, we report detailed throughput and memory usage. Key findings include:
>    **Time per iteration (seconds, CIFAR-10):**
>
>    | IPC | DATM |DC |MTT | CSDM |
>    | --- | ---- | ------- | ------- | ---- |
>    | 10  | OOM  | 48      |  0.4 | 0.30 |
>    | 20  | OOM  | 107     |  0.7 | 0.35 |
>    | 50  | OOM  | OOM     |  OOM | 0.33 |
>
>    **Memory (GB, CIFAR-10 on RTX 4090):**
>
>    | IPC | DATM | DC |MTT   | CSDM |
>    | --- | ---- | ------- | ----- | ---- |
>    | 10  | OOM  | 10.89   | 10.84 | 3.43 |
>    | 20  | OOM  | 20.69   | 16.91 | 3.44 |
>    | 50  | OOM  | OOM     | OOM   | 3.45 |
>    * **CSDM is the only method that completes CIFAR-10 IPC 50 on a single RTX 4090** in our setup, with peak memory less than $3.5$GB.
>    * Methods such as DC, MTT, and DATM **run out of memory** at IPC 50 under the same hardware and software configuration.
>    * CSDM is at least **$300\times$ faster than DC** in our measurements, and also much faster than MTT and other DC-based approaches.
>
>    These results confirm that our theoretical complexity advantage translates into **practical scalability**, especially at higher IPC and on larger datasets.
>
>    The runtime/memory tables appear in **Appendix E.3** “Distillation time and GPU memory usage”, **Page 30**.

---

> ### Author Response · Authors · 2025-11-22
> **Q5: Clarification on frequency-based methods (FreD, NSD) and orthogonality to our approach.**
>
> Thanks for asking us to clarify the relation to FreD and NSD. Earlier literature rarely addressed dataset distillation in the frequency domain. FreD and NSD are two pioneering works that have achieved outstanding performance, providing compelling evidence for the power of frequency-based approaches. The key distinction is that **FreD and NSD modify how synthetic data are parameterized**, whereas **CSDM modifies what distance is minimized**. These are orthogonal design dimensions.
>
> 1. **Unified view: parameterization vs. metric.**
>    We adopt the generic formulation
>    $$
>    \min\_\theta \mathcal{L}\_{\text{metric}}\big(
>    \Phi\_{\text{syn}}(\theta),
>    \Phi\_{\text{real}}(D)
>    \big),
>    $$
>    where:
>
>    * $\Phi_{\text{syn}}(\theta)$ specifies the **parameterization** of synthetic data.
>    * $\mathcal{L}_{\text{metric}}$ specifies the **distribution discrepancy** to be minimized.
>
>    Under this view:
>
>    * **FreD / NSD (parameterization side, $\Phi_{\text{syn}}$):**
>
>      * FreD maps images to the Fourier domain and optimizes a truncated subset of coefficients, with a frequency mask derived from explained variance. It still uses standard matching objectives (e.g., gradient/trajectory matching) as $\mathcal{L}_{\text{metric}}$.
>      * NSD represents the entire synthetic dataset as a global spectral tensor and reconstructs per-sample images via low-rank tensor decomposition, again typically combined with existing trajectory-matching metrics.
>    * **CSDM (metric side, $\mathcal{L}_{\text{metric}}$):**
>
>      * We keep a standard parameterization in image space (though our loss could be applied to FreD/NSD as well).
>      * We redesign the **distribution discrepancy** by replacing linear-kernel MMD with a universal, spectral form (SDD), and then making it class-aware via amplitude–phase weighting.
>
>    A concise comparison:
>
>    | Method          | Core innovation  | Frequency-domain role                      | Explicit handling of class imbalance? |
>    | --------------- | ---------------- | ------------------------------------------ | ------------------------------------- |
>    | FreD            | Parameterization | Store/optimize truncated Fourier coeffs    | No                                    |
>    | NSD             | Parameterization | Global spectral tensor representation      | No                                    |
>    | **CSDM (ours)** | Metric / loss    | Spectral MMD (SDD) + class-aware amp/phase | **Yes**                               |
>
> 2. **Complementarity and potential combinations.**
>    Because these methods operate on different components, CSDM is **complementary** to FreD/NSD:
>
>    * One could use FreD or NSD as $\Phi_{\text{syn}}(\theta)$, and **replace their loss** by our CSDM objective.
>    * This would combine frequency-based compression of the synthetic data with a theoretically grounded spectral metric.
>
>    We have revised the Related Work and added an explicit subsection in the **Appendix D**(“Comparison with Frequency-domain Distillation Methods”, **Page 28**) to emphasize this decomposition and clarify that our contribution lies fundamentally on the **metric side**, not on frequency-domain parameterization.

---

> ### Author Response · Authors · 2025-11-22
> **Q6: About results on balanced datasets.**
>
> Thanks for asking us to clarify this. While the main text emphasizes long-tailed settings, we have also run extensive experiments on standard balanced CIFAR-10 and CIFAR-100.
>
> 1. **CSDM achieves SOTA or competitive SOTA on balanced CIFAR-10/100.**
>    Using 18 baselines (including M3D, IID, DSDM, NCFM, FreD, NSD, MTT, FTD, TESLA, etc.), Table 5 in the **Appendix E** (**Page 28**) shows that:
>
>    * On **CIFAR-10**:
>
>      * IPC 10: CSDM $71.2%$ vs. NCFM $71.8%$, FreD $70.3%$, and other high-order methods around $60%$. We are essentially **on par with the best**.
>      * IPC 50: CSDM **$78.1%$**, the **highest accuracy** among all compared methods (slightly above NCFM $77.4%$ and clearly above FreD $75.8%$ and IID/M3D/DSDM).
>    * On **CIFAR-100**:
>
>      * IPC 10: CSDM **$50.4%$**, again the **best** result, surpassing NCFM ($48.7%$) and significantly outperforming IID/M3D/DSDM (around $40%$).
>      * IPC 50: CSDM $54.1%$, closely matching NCFM $54.7%$ and above other MMD-based methods.
>
>    This demonstrates that our method is not tailored only for long-tailed regimes; it also provides a strong and often superior discrepancy on standard balanced dataset distillation benchmarks.
>
>    The full balanced results are presented in **Appendix E** “Supplementary Experimental Results”, **Page 29**.
>
> | Dataset        | CIFAR-10         |                  |                  | CIFAR-100        |                  |
> |----------------|------------------|------------------|------------------|------------------|------------------|
> | IPC            | 1                | 10               | 50               | 10               | 50               |
> | Ratio (%)      | 0.02             | 0.2              | 2                | 2                | 10               |
> | Random         | 14.4 ± 0.2       | 26.0 ± 1.0       | 43.4 ± 1.0       | 15.1 ± 0.5       | 30.7 ± 0.4       |
> | Herding        | 21.5 ± 1.2       | 31.0 ± 1.0       | 46.3 ± 1.1       | 18.5 ± 1.0       | 34.5 ± 0.6       |
> | Forgetting     | 13.5 ± 1.2       | 23.3 ± 1.0       | 23.3 ± 1.1       | 15.1 ± 1.0       | 30.5 ± 0.7       |
> | DC             | 28.3 ± 0.5       | 44.9 ± 0.5       | 49.5 ± 0.5       | 25.5 ± 0.5       | 34.8 ± 0.5       |
> | DSA            | 28.8 ± 0.7       | 52.1 ± 0.5       | 54.5 ± 0.5       | 32.0 ± 0.5       | 42.8 ± 0.5       |
> | DCC            | 29.7 ± 0.4       | 50.4 ± 0.4       | 58.2 ± 0.5       | 30.0 ± 0.4       | 48.2 ± 0.5       |
> | DSAC           | 30.8 ± 0.5       | 51.4 ± 0.5       | 59.0 ± 0.5       | 29.0 ± 0.5       | 46.6 ± 0.5       |
> | FrePo          | 36.3 ± 0.5       | 55.3 ± 0.5       | 60.5 ± 0.4       | 35.5 ± 0.6       | 48.3 ± 0.4       |
> | MTT            | 46.3 ± 0.8       | 58.3 ± 0.7       | 67.6 ± 0.5       | 38.7 ± 0.6       | 45.7 ± 0.8       |
> | ATT            | 46.3 ± 0.8       | 57.5 ± 0.6       | 67.5 ± 0.5       | 39.3 ± 0.4       | 45.4 ± 0.7       |
> | FTD            | 46.8 ± 0.5       | 60.3 ± 0.4       | 67.9 ± 0.5       | 40.2 ± 0.5       | 45.4 ± 0.4       |
> | TESLA          | 46.3 ± 0.5       | 59.0 ± 0.4       | 67.9 ± 0.5       | 37.1 ± 0.5       | 44.1 ± 1.2       |
> | CAFE           | 30.3 ± 1.1       | 41.1 ± 1.1       | 46.4 ± 1.1       | 19.5 ± 0.7       | 35.7 ± 1.1       |
> | DM             | 43.0 ± 1.5       | 54.4 ± 1.1       | 67.6 ± 1.5       | 37.2 ± 1.4       | 46.6 ± 1.9       |
> | IDM            | 45.6 ± 1.0       | 58.6 ± 1.0       | 68.1 ± 1.0       | 38.2 ± 1.0       | 46.9 ± 0.5       |
> | M3D            | 47.1 ± 1.0       | 59.9 ± 0.5       | 69.3 ± 1.0       | 39.7 ± 1.0       | 47.6 ± 0.5       |
> | IID            | 47.1 ± 1.0       | 60.9 ± 1.0       | 70.1 ± 1.0       | 39.8 ± 1.0       | 47.2 ± 1.0       |
> | DSDM           | 46.5 ± 0.7       | 58.4 ± 0.7       | 67.0 ± 0.7       | 36.7 ± 0.7       | 45.4 ± 0.5       |
> | G-VBSM         | 46.5 ± 0.5       | 54.3 ± 0.5       | 60.4 ± 0.5       | 38.7 ± 0.7       | 45.7 ± 0.7       |
> | FreD           | **60.6 ± 0.8**   | 70.3 ± 0.3       | 75.8 ± 0.1       | 42.7 ± 0.2       | 47.8 ± 0.1       |
> | NCFM           | 49.5 ± 0.3       | **71.8 ± 0.3**   | 77.4 ± 0.3       | 48.7 ± 0.5       | **54.7 ± 0.2**   |
> | **CSDM (ours)**| 47.3 ± 0.3       | 71.2 ± 0.4       | **78.1 ± 0.4**   | **50.4 ± 0.3**   | 54.1 ± 0.2       |
> | Whole Dataset  | 84.8 ± 0.1       |                  |                  | 56.2 ± 0.3       |                  |

---

> ### Author Response · Authors · 2025-11-22
> **Q7: About why the baselines for long-tailed ImageNet are fewer than for long-tailed CIFAR.**
>
> Thanks for pointing this out. The difference is mainly driven by the current state of the literature and practical constraints in adapting baselines.
>
> **Long-tailed dataset distillation on ImageNet is less explored and requires heavy re-implementation.**
>
>    * Long-tailed dataset distillation is still a relatively new topic. Only a small number of works report results on **ImageNet-LT–style** settings.
>    * Creating long-tailed variants requires **re-sampling from the original dataset with class-dependent probabilities**. For each baseline method, this means modifying its codebase to adopt the new sampling scheme.
>    * On large-scale datasets like ImageNet, each additional baseline run is **computationally and engineering-wise expensive** (GPU hours, debugging time, etc.). Given time and resource limits, we prioritized:
>
>      * A **rich baseline set on CIFAR-10-LT and CIFAR-100-LT**, where running many methods is feasible.
>      * A **smaller but representative** baseline set on ImageNet-LT.
>
> ---
>
> Thanks again for all these detailed and technically oriented comments. They directly motivated us to add: (i) 12 pages of theoretical foundation (IPM/RKHS/universal kernels, moment expansions, SDD–CFD analysis, class-aware discriminativity); (ii) extensive balanced and long-tailed experiments; and (iii) explicit positioning of CSDM relative to high-order matching methods and frequency-parameterization methods.

---

> ### Author Response · Authors · 2025-11-26
> **In Appreciation of Your Review: Supplementary Materials and Clarifications Submitted**
>
> Dear Reviewer fbvC,
>
>
> Thank you once again for your thoughtful and detailed feedback on our submission. In response to your initial review—which raised several important concerns, some of which stemmed from **misunderstandings that significantly impacted the evaluation**—we have taken substantial steps to address them.
>
> Specifically, we have prepared a comprehensive **15-page appendix** containing rigorous theoretical proofs and additional experimental results to directly clarify the points in question. We believe these additions fully resolve the misconceptions and provide strong support for our claims.
>
> As the discussion phase is now drawing to a close on **December 3, AoE**, we kindly ask whether our responses and supplementary materials have adequately addressed your concerns (at least in part). If any questions remain or if further clarification would be helpful, please do not hesitate to let us know. Conversely, if you find our clarifications satisfactory, **we would be truly grateful if you could consider revising your evaluation scores accordingly—especially given the significance of the corrected misunderstandings to the overall assessment.**
>
> Given the timeline of the review process, we sincerely hope to receive your final feedback before the discussion window closes.
>
> Thank you very much for your time and consideration.

---

> ### Author Response · Authors · 2025-11-27
> **Follow-Up on Rebuttal Clarifications**
>
> Dear reviewer fbvC,
>
> Thank you very much for your valuable feedback on our paper.
>
> We wanted to follow up regarding the response we posted earlier, where we addressed all your comments, including **a clear clarification of the fundamental differences between our method and FreD**, the **theoretical and experimental comparisons with existing high-order matching approaches**, the **additional balanced CIFAR experiments**, and the **complexity, runtime, and memory analyses and corresponding experiments**, among others.
>
> If there are any remaining concerns, or if any of our clarifications (in the new Section 6, the extended appendix, or the official comments) would benefit from further explanation, please feel free to let us know — we would be glad to provide any additional details.
>
> If you feel that the revised manuscript and rebuttal have adequately addressed your main concerns, we would greatly appreciate it if you could reflect this in your final comments and score.
>
> Best,
> Authors

---

> ### Author Response · Authors · 2025-11-27
> **Supplementary Analysis on Frequency-Oriented Baselines**
>
> Dear Reviewer fbvC,
>
> We sincerely thank you for highlighting frequency-based methods, including **FreD and NSD**, in your review. We would like to point out that **we have now explicitly included and discussed these approaches** in both the Related Work section and throughout the rebuttal, providing **detailed comparisons and theoretical clarifications**.
>
> We kindly invite you to **re-examine the updated manuscript and supplemental materials**, as we believe these additions comprehensively address the concerns regarding frequency-based methods and clarify the distinctions and connections with our proposed CSDM framework.
>
> We greatly appreciate your time and careful consideration of the updated material.
>
> Sincerely,
>
> Authors

---

### Author Response · Authors · 2025-11-22
**(Part 1) General Response**

We sincerely thank the Area Chair and all reviewers for their valuable and constructive feedback. We are encouraged by the shared recognition of the importance of principled distribution matching and the strength of our long-tailed results. Across the reviews, several aspects of our work were consistently highlighted as strengths: the clear identification of linear-kernel MMD limitations, the simplicity and soundness of CSDM’s formulation, and its substantial performance gains on severe imbalance.

During the rebuttal period, we significantly strengthened the paper with **15 additional pages of theoretical analysis and new experiments**, addressing all concerns raised by the reviewers. Below we summarize the major updates and clarifications.

---

### Author Response · Authors · 2025-11-22
**(Part 2) General Response - New Experiments and Additional Empirical Analyses**

### **1. We substantially expanded experiments on balanced CIFAR-10/100 with 18 baselines, showing CSDM matches or surpasses SOTA at IPC ≥ 10.**

We now compare against a comprehensive suite of 18 baselines—including M3D, IID, DSDM, NCFM, FreD, NSD, MTT, FTD, TESLA, and others—on CIFAR-10 and CIFAR-100.
**CSDM achieves SOTA or competitive SOTA on both datasets** (Table 5, Page 29).
For example:

* **CIFAR-10 IPC 50:** CSDM = **78.1%**, **surpassing all listed methods** including NCFM (77.4%), IID(70.1%) and FreD (75.8%).
* **CIFAR-100 IPC 10:** CSDM = **50.4%**, higher than NCFM (48.7%), IID(39.8%) and FreD（42.7%）.

These results demonstrate that CSDM is not only effective for long-tailed distillation but is also a **general improvement** for balanced dataset distillation.

---

### **2. We added detailed runtime and memory comparisons, showing CSDM is dramatically more efficient than prior methods.**

We provide new runtime/memory experiments (Table 6, Page 30), showing:

* **CSDM is the only method that can distill CIFAR-10 IPC 50 on a single RTX 4090**, requiring only **≈3.5 GB** memory.
* DC, MTT, and DATM all **OOM at IPC 50**.
* **CSDM is at least 300× faster than DC** and substantially faster than MTT and DC-based approaches.

These results confirm that CSDM achieves excellent accuracy **while dramatically reducing computational cost**, addressing all efficiency concerns.

---

### **3. We evaluate more kernels and sampling strategies across balanced and imbalanced settings (Figure 5, Page 31).**

We added new ablations showing that kernels capable of capturing higher-order information—together with their corresponding spectral sampling—achieve significantly **stronger accuracy** and much **greater stability** than the **widely used linear MMD** baseline, particularly as the imbalance factor increases.

---

### **4. We visualize separately distilled amplitude-only and phase-only images (Figure 6, Page 31), supporting our diversity/realism interpretation.**
Figure 6 presents synthetic images distilled using only the amplitude term or only the phase term.The results show a striking contrast: amplitude-only distillation produces diverse, while phase-only distillation preserves realism but exhibits limited diversity.

These new visualizations empirically confirm that:

* **Amplitude** controls energy spread and sample diversity.
* **Phase** controls structural fidelity and realism.
  This supports the theoretical explanation behind class-aware weighting.

---

### Author Response · Authors · 2025-11-22
**(Part 3) General Response – Theoretical Clarifications and Major Additions**

### **5. Clarification of a major misunderstanding**
CSDM and FreD/NSD intervene at *different levels* of the distillation pipeline, targeting **fundamentally different** variables.

To clarify this key distinction, we explicitly added a unified view of dataset distillation using the standard formulation

$$\min\_{\theta}\mathcal{L}\_{\text{metric}}(\Phi_{\text{syn}}(\theta),\Phi_{\text{real}}(D)),$$

where **$Φ_{\text{syn}}$** specifies **how synthetic data are parameterized**, and **$\mathcal{L}_{\text{metric}}$** specifies **how distributions are compared**.

Under this formulation, the difference becomes unambiguous:

* **FreD and NSD modify the *data parameterization* $\Phi_{\text{syn}}(\theta)$** by representing synthetic samples directly in the frequency domain (FreD) or via spectral tensor decomposition (NSD). Their contribution lies in *how synthetic data are stored and generated*.

* **CSDM modifies the *distribution matching metric* $\mathcal{L}_{\text{metric}}$** by replacing linear-kernel MMD with a universal, spectral-domain discrepancy (SDD) and introducing a class-aware amplitude–phase weighting. Our contribution lies in *how real and synthetic distributions are aligned*, independent of how the data are parameterized.

Thus, **the two approaches operate on orthogonal components of the pipeline**—CSDM redesigns the *metric*, whereas FreD/NSD redesign the *representation*. This conceptual separation is now clearly documented in the revised Related Work, with a detailed analysis provided in **Appendix D**, **Page 28**.


---

### **6. We provide a unified theoretical analysis comparing five high-order DD methods (IID, DSDM, M3D, NCFM, WMDD).**

We added a full 5-method comparison (**6 DISCUSSION**, **Page 9**, **Appendix B**, **Page 23**):

* **IID & DSDM**: We show their covariance regularizers are equivalent to the **2nd-order Taylor term of the log characteristic function** (Theorem 6), hence approximating only limited high-order structure.
* **M3D**: While recognizing linear MMD’s limitations, it inconsistently uses non-universal kernels and reports results conflicting with its own theoretical claims. This highlights the need for a more principled kernel analysis.
* **NCFM**: The min–max sampling network does not guarantee alignment with downstream objectives. Our long-tailed results show the sampling network struggles under imbalance, validating our theoretical analysis and providing insights for future adaptive samplers.
* **Wasserstein-based DD**: We discuss its complementary nature and how our universal-kernel perspective can guide future unified metrics.

---

### **7. We provide a detailed explanation of amplitude and phase roles in diversity and realism.**

We expanded Section E with:

* A review of classical signal-processing literature (e.g., Oppenheim & Lim 1981).
* A numerical analysis showing why amplitude governs diversity and phase governs structural realism.
* A clear explanation of why long-tailed distillation specifically requires this decomposition.

---

### **8. We prove that class-aware SDD preserves discriminativity (Proposition).**

We added a complete proof showing that as long as $α(c) ∈ (0,1)$, the metric remains characteristic and satisfies $P = Q ⇔ d = 0$.
This resolves reviewers’ concerns about whether class-aware weighting preserves optimality.

---

### **9. We added theoretical foundation: MMD, universal kernels, moment expansions, and SDD–CFD relations.**

The appendix now includes:

* A clear review of IPM → RKHS → universal kernel MMD
* A moment expansion showing RBF MMD implicitly matches **all** orders of moments
* A precise analysis of SDD vs CFD, clarifying when they coincide and differ

This makes CSDM theoretically grounded in a way that has not existed previously in dataset distillation literature.

---

# **Closing**

We thank the AC and reviewers again for their careful reading and thoughtful comments. The expanded theoretical analysis, 18-baseline comparisons, balanced-dataset results, efficiency benchmarks, visualizations, and detailed clarifications comprehensively address all concerns.
CSDM is a principled, efficient, and robust distribution-matching framework that advances dataset distillation in both balanced and long-tailed settings.

---

### Meta-Review · Area_Chair_tfSS · 2026-01-06

**Summary:**

The reviewers raised several key concerns in the original reviews, including: (1) insufficient novelty and theoretical grounding of the Spectral Distribution Distance (SDD), which was seen as a known concept (Characteristic Function Distance); (2) lack of comparisons with prior higher-order distribution matching methods (e.g., IID, DSDM, M3D) and frequency-based approaches (FreD, NSD); (3) unclear or unjustified claims about the link between amplitude/phase and diversity/realism, and the heuristic choice of class-specific weights α(c); (4) missing complexity/runtime analysis and visualizations of distilled data.

The authors' rebuttal has addressed most of the concerns, there is still an important debate for the novelty of SDD versus prior CFD works. Given that all the reviewers gave negative scores in the original reviews, I recommend Reject.

**Reviewer Concerns:**

The authors' rebuttal has addressed most of the concerns. They added theoretical analysis, conducted additional experimental comparisons, and provided runtime/memory results. However, some reviewers may still question the novelty of SDD versus prior CFD works.

**Reviewer Scores:**

Although the authors' rebuttal has addressed most of the concerns, I guess most of the reviewers would still remain negative because of the novelty issue of SDD versus prior CFD works.

---

### Decision · Program_Chairs · 2026-01-26

Reject